# DR.MI-BENCH: A MODULAR-INTEGRATED BENCHMARK FOR SCIENTIFIC DEEP RESEARCH AGENT

## ABSTRACT

The explosive growth of academic literature drives the need for automated deep research (DR) agents, yet evaluating these systems remains a significant challenge. First, existing benchmarks often focus narrowly on retrieval while neglecting high-level planning and reasoning. Second, existing benchmarks favor general domains over the scientific domains that are the core application for DR agents. To address these gaps, we introduce **Dr.Mi-Bench**, a Modular-integrated Benchmark for scientific DR agents. Grounded in academic literature, our benchmark is a human-annotated dataset of 200 instances across 10 scientific domains, including both research and review papers. Furthermore, we propose a Modular-integrated Evaluation Paradigm for DR Agents (**Dr.Mi-Eval**), which leverages the rich structure of academic papers to assess the core capabilities of planning, retrieval, and reasoning. It employs two complementary modes: an **end-to-end** evaluation for DR agents and an **isolated** evaluation for foundational LLMs as potential backbones. Experimental results reveal an uneven performance landscape: while agents show specialized strengths, they share critical limitations, particularly in multi-source retrieval for review tasks and maintaining consistency across diverse scientific fields. Moreover, improving high-level planning capability is the crucial factor for unlocking the reasoning potential of foundational LLMs as backbones. By exposing these actionable failure modes, Dr.Mi-Bench provides a diagnostic tool to guide the development of more reliable academic research assistants.

## 1 INTRODUCTION

Since human society has entered an era of unprecedented information explosion, conducting deep, accurate, and reliable research requires intensive knowledge work. Therefore, it is increasingly impractical for human experts to handle this task with conventional manual methods (Zhang et al., 2025a). The scale and speed of academic publication have reached unprecedented levels. For instance, more than 64 million research papers have become accessible online since 1996 (WordsRated, 2025), and the pace of new contributions continues to accelerate. The trend increases the difficulty of completing a comprehensive report with conventional manual methods, thereby underscoring the urgent need for automated systems (Zheng et al., 2025). In response, a novel class of Large Language Model (LLM)-based DR agents has emerged to conduct this complex workflow and produce information-dense reports automatically (Li et al., 2025b).

*Deep research* agents, which autonomously conduct knowledge-intensive tasks by searching heterogeneous sources and synthesizing findings, are attracting significant attention in both academia (Zhang et al., 2025b; Xu & Peng, 2025) and industry (OpenAI, 2025; Google, b; PerplexityAI, a; xAI). For example, OpenAI characterizes the functionality of such agents as the ability to produce comprehensive, human-quality reports by automating the often hours-long process of searching, analyzing, and synthesizing evidence (OpenAI, 2025). However, research in this area remains at an early stage. Specifically, comprehensive evaluation is crucial yet currently limited. Thus, in this study, we investigate DR and decompose it into three key aspects: *Planning*, *Retrieval*, and *Reasoning*. First, planning module decomposes a complex query into a well-structured sequence of sub-tasks. Second, retrieval module searches multiple sources, then iteratively selects useful evidence within a fixed budget. Third, reasoning module reasons over the defined sub-tasks, calls external search tools, and synthesizes them into a comprehensive report. In summary, a comprehensive, modular-integrated evaluation

Table 1: Comparison of Dr.Mi-Bench with existing DR benchmarks.

| Benchmark | Domain | End-to-End | Planning | Retrieval | Reasoning | Efficiency |
|---|---|---|---|---|---|---|
| GAIA | General | ✗ | ✗ | ✓ | ✗ | ✗ |
| BrowseComp | General | ✗ | ✗ | ✓ | ✗ | ✗ |
| BrowseComp-ZH | General | ✗ | ✗ | ✓ | ✗ | ✗ |
| HLE | General | ✗ | ✗ | ✓ | ✗ | ✗ |
| Deep Research Bench | General | ✗ | ✗ | ✓ | ✗ | ✓ |
| **Dr.Mi-Bench (Ours)** | **Scientific** | ✓ | ✓ | ✓ | ✓ | ✓ |

paradigm can accelerate the development of stronger DR agents designed for knowledge-intensive tasks. However, current benchmarks are still at an early stage, with key bottlenecks outlined below.

Despite rapid progress, DR evaluation still exhibits two fundamental bottlenecks. **First, prevailing benchmarks reduce the task to a retrieval slice, with little systematic evaluation of high-level planning and reasoning** (Wei et al., 2025; Zhou et al., 2025; Phan et al., 2025; Mialon et al., 2023). Therefore, such evaluations do not faithfully reflect the capacity of an agent to solve complex problems. In particular, poor retrieval may be induced by an incorrect plan rather than shortcomings in the search process itself. Moreover, even when the correct evidence is retrieved, the model can still hallucinate, owing to limited semantic understanding and reasoning over long contexts, producing plausible yet incorrect conclusions. **Second, current evaluations demonstrate limited capability in scientific domains.** Existing DR benchmarks disproportionately emphasize general-purpose questions while neglecting domain-specific and professional workloads, which are central to real-world DR scenarios (FutureSearch et al., 2025; Du et al., 2025; Java et al., 2025).

To address the aforementioned limitations, we formalize the workflow of DR agents and propose **Dr.Mi-Bench**, a modular-integrated deep research benchmark for scientific deep research agent. Notably, **Dr.Mi-Eval** evaluates the top-down capability of deep-research models to solve open-domain scientific research questions (Section 3). Our methodology is grounded in academic papers, which naturally provide structured guidelines for core competencies of an DR agent: planning, retrieval, and reasoning. **For planning**, we compare decomposed sub-tasks provided by agents against research plans annotated by experts based on the structural outline of papers, to quantify coverage of requisite sub-tasks, redundancy, and correctness. **For retrieval**, the benchmark assesses the relevance, coverage, and provenance of retrieved evidence, using the citations of papers as a verifiable ground truth. **For reasoning**, to evaluate reasoning on open-ended research questions, we construct a suite of diagnostics comprising boolean statements and answers based on the content of academic papers, which is used to evaluate the accuracy, consistency, depth, and breadth against the key findings. The three stages can be evaluated independently and can also be integrated into a closed loop to measure the end-to-end performance of a deep-research system. Furthermore, we evaluate the efficiency of DR agents by analyzing the trade-offs between their performance and three key costs: processing time (latency), report length (token count), and price (average and total). A detailed comparison between Dr.Mi-Bench and other existing DR benchmarks appears in Table 1. In support of this evaluation paradigm, we curate a human-annotated dataset spanning 10 scientific domains with 200 instances that include both research papers and review papers (Section 4). To demonstrate the utility of our benchmark, we conduct a comprehensive evaluation of both state-of-the-art end-to-end DR systems (e.g., OpenAI, Google) in Section 5 and a diverse set of foundational LLMs (e.g., GPT-o3, Llama) assessed as potential backbones in Section 6. Our analysis reveals a fragmented performance landscape where agents exhibit specialized strengths: Grok in retrieval and Gemini in reasoning, yet all share common weaknesses, most notably in retrieving from complex review papers and maintaining consistent performance across diverse scientific domains. Dr.Mi-Bench reveals significant substantial capacity for reliable DR, enabling more trustworthy academic assistants.

## 2 FORMALIZATION OF DEEP RESEARCH AGENT WORKFLOW

In this section, we formalize the generalized workflow of a DR agent to establish a structured framework for our benchmark. Given a user query, the primary goal of such an agent is to produce a comprehensive, evidence-grounded report. As illustrated in Figure 1c, we decompose this workflow into three core modules: (i) Planning, the query understanding and task decomposition; (ii) Retrieval,

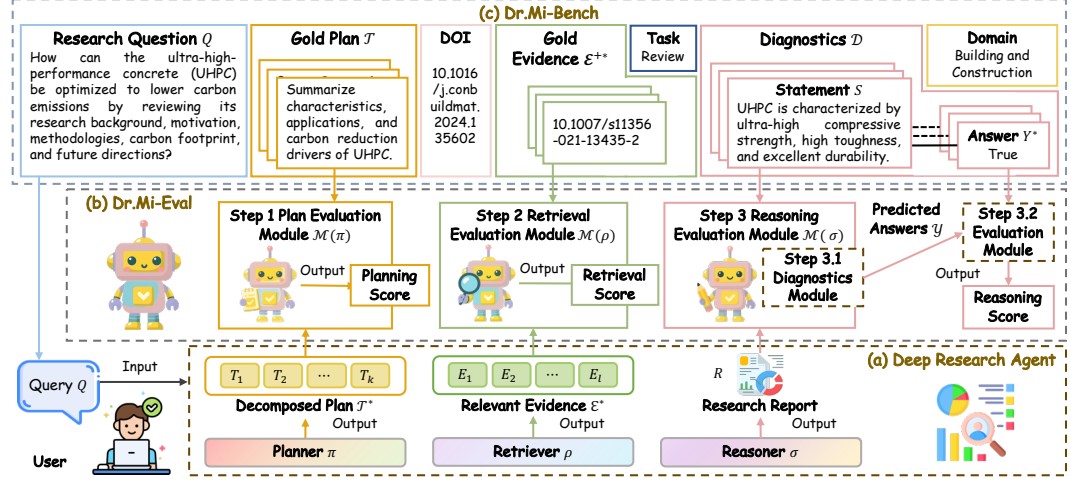

Figure 1: **The Dr.Mi-Bench and Dr.Mi-Eval Framework.** This figure illustrates how a DR agent workflow (a) is assessed by our modular-integrated evaluation paradigm (b) using our human-annotated benchmark (c). This approach enables a diagnostic analysis that pinpoints specific failures in the agent's planning, retrieval, and reasoning capabilities.

the acquisition of relevant documents; and (iii) Reasoning, the synthesis of evidence and generation of the final report.

**Definition 1 (Deep Research Agent)** *Let $Q$ denote a user query, $\mathcal{E}$ denotes the retrieval corpus (e.g., the open web or an offline collection), $\mathcal{H}$ a toolset (e.g., search APIs or local indexers), and $\mathcal{B}$ a budget (e.g., tool-call and time limits). A DR agent $\mathcal{A}$ is a mapping: $\mathcal{A}: (Q, \mathcal{E}, \mathcal{H}, \mathcal{B}) \mapsto (R, \mathcal{E}^+)$, where $R$ is a comprehensive report and $\mathcal{E}^+ = \{E_1^+, E_2^+, \ldots, E_l^+\} \subseteq \mathcal{E}$ is a set of retrieved evidence. that attributes each claim in $\mathcal{R}$ to cited evidence in $\mathcal{E}$. The report $R$ must be fully grounded in, and its claims attributable to, the evidence in $\mathcal{E}^+$.*

The DR agent $\mathcal{A}$ is a composition of three modules: $\mathcal{A} \equiv \sigma \circ \rho \circ \pi$, where $\pi$, $\rho$ and $\sigma$ denotes the planner, retriever and reasoner, respectively. Formally, the functionalization of the three modules can be denoted as:

$$\pi: Q \to \mathcal{T}, \quad \rho: (Q, \mathcal{T}, \mathcal{E}, \mathcal{H}, \mathcal{B}) \to \mathcal{E}^+, \quad \sigma: (Q, \mathcal{T}, \mathcal{E}^+) \to \mathcal{R}, \tag{1}$$

where $\mathcal{T} = \langle T_1, T_2, \ldots, T_k \rangle$ is a plan with ordered sub-tasks and $\mathcal{E}^+$ the curated set of predicted relevant evidence (citations). The detailed formalization and scoring for each component are provided in Appendix A.1 and Appendix A.15 .

## 3 DR.MI-BENCH EVALUATION PARADIGM

We introduce a novel **modular-integrated evaluation paradigm** for scientific DR agents, designed to assess their top-down research capabilities. Our evaluation is structured around three core modules: Planning, Retrieval, and Reasoning. It supports two complementary evaluation modes. In the **End-to-end mode**, modules are chained together for a holistic performance assessment, while in the **isolated mode**, each module is assessed independently with ground-truth inputs to measure its maximum capability. This dual-mode design combines the realism of a full-system evaluation with the precision of module-specific diagnostics, allowing us to attribute an agent's performance to the intrinsic capabilities of its foundational LLM. This paradigm is implemented in our new benchmark, **Dr.Mi-Bench**, which is built upon a human-annotated dataset of 200 instances from 10 scientific domains. For comprehensive evaluation, each instance is annotated with a user query, a ground-truth plan, a set of gold evidence, and a suite of diagnostics.

**Definition 2 (Modular-Integrated Evaluation Paradigm)** *Our paradigm provides a dual-level assessment. We evaluate end-to-end agents using a End-to-end Mode for holistic performance, and*

*foundational LLMs using an Isolated Mode for modular analysis of specific agentic capabilities. For each benchmark instance, we have a ground-truth tuple $(Q, \mathcal{T}^*, \mathcal{E}^{+*}, \mathcal{D})$ and corresponding metric suites $(\mathcal{M}_\pi, \mathcal{M}_\rho, \mathcal{M}_\sigma)$.*

$$\textbf{\textit{End-to-end (Agents):}} \quad \left. \begin{array}{l} \mathcal{T} \leftarrow \pi(Q) \\ \mathcal{E}^+ \leftarrow \rho(Q, \mathcal{T}, \mathcal{E}, \mathcal{H}, \mathcal{B}) \\ R \leftarrow \sigma(Q, \mathcal{T}, \mathcal{E}^+) \end{array} \right\} \implies \left\{ \begin{array}{l} \mathcal{M}_\pi(\mathcal{T}, \mathcal{T}^*) \\ \mathcal{M}_\rho(\mathcal{E}^+, \mathcal{E}^{+*}). \\ \mathcal{M}_\sigma(R, \mathcal{D}) \end{array} \right. \tag{2}$$

$$\textbf{\textit{Isolated (LLMs):}} \quad \left. \begin{array}{l} \mathcal{T} \leftarrow \pi(Q) \\ \mathcal{E}^+ \leftarrow \rho(Q, \mathcal{T}^*, \mathcal{E}, \mathcal{H}, \mathcal{B}) \\ R \leftarrow \sigma(Q, \mathcal{T}^*, \mathcal{E}^{+*}) \end{array} \right\} \implies \left\{ \begin{array}{l} \mathcal{M}_\pi(\mathcal{T}, \mathcal{T}^*) \\ \mathcal{M}_\rho(\mathcal{E}^+, \mathcal{E}^{+*}). \\ \mathcal{M}_\sigma(R, \mathcal{D}) \end{array} \right. \tag{3}$$

This dual-level protocol allows us to attribute an agent's end-to-end performance to the intrinsic capabilities of its foundational LLM backbone.

### 3.1 PLANNING MODULE EVALUATION

The evaluation of the Planning module is designed to assess the ability of agents to decompose a query $Q$ into a structured plan. A plan predicted by agent, denoted as $\mathcal{T} = \langle T_1, \ldots, T_k \rangle$, is scored against the expert-annotated ground-truth plan, $\mathcal{T}^* = \langle T_1^*, \ldots, T_m^* \rangle$. Our protocol employs an exhaustive **pairwise comparison** between the predicted and ground-truth sub-tasks. Specifically, an LLM-based evaluator semantically compares each predicted sub-task $T_i \in \mathcal{T}$ against every ground-truth sub-task $T_j^* \in \mathcal{T}^*$ to determine relevance. This matrix of pairwise judgments is then used by our metric suite, $\mathcal{M}_\pi$, to quantify the plan's overall coverage of essential sub-tasks, its structural correctness, and its redundancy. As planning is the initial stage of the pipeline, this evaluation process is identical for both the end-to-end and isolated modes.

### 3.2 RETRIEVAL MODULE EVALUATION

The evaluation of the retrieval module is designed to assess the consistency between the set of citations retrieved by the agent, $\mathcal{E}^+$, and the ground-truth citations annotated in our dataset, $\mathcal{E}^{+*}$. To quantify this alignment, our metric suite, $\mathcal{M}_\rho$, employs a **rule-based exact match** on canonical identifiers (DOIs) to calculate scores for relevance, coverage, and provenance (detailed in A.17) .

We assess this capability in our two distinct modes: In the **end-to-end Mode**, the module receives the agent's own predicted plan, $\mathcal{T}$, as input. This setup measures the retriever's practical, end-to-end performance and reveals its susceptibility to any errors propagated from the upstream Planning module. In the **isolated Mode**, the module is provided with the ground-truth plan, $\mathcal{T}^*$. This assesses the module's maximum intrinsic capability, assuming a perfect plan was formulated. By comparing results from these two modes, we can effectively disentangle retrieval failures from planning failures.

### 3.3 REASONING MODULE EVALUATION

The reasoning module is evaluated on its core capability to synthesize a factually correct, coherent, and evidence-grounded scientific report $R$. Since report generation is an open-ended task that admits various valid solutions, direct comparison against a gold report can be unreliable. Therefore, we transform the subjective assessment of report quality into an objective and reproducible evaluation. Our diagnostic framework, $\mathcal{M}_\sigma$, scores the report $R$ against a set of predefined, ground-truth diagnostic pairs $\mathcal{D}$.

In the **end-to-end Mode**, the module receives the predicted plan $\mathcal{T}$ and predicted evidence set $\mathcal{E}^+$ from the preceding stages. This evaluation measures the final, overall quality of the agent's generated report, capturing the cumulative impact of all potential upstream errors. In the **isolated Mode**, the module is given the perfect, ground-truth plan $\mathcal{T}^*$ and evidence set $\mathcal{E}^{+*}$. This setup isolates the foundational LLM from relying on its inherent reasoning capability, revealing its performance when all prerequisite information is correct and complete. This dual-mode analysis allows us to distinguish true reasoning failures from performance degradation caused by propagated errors from the planning and retrieval modules.

## 3.4 Efficiency Evaluation

Beyond correctness, a crucial aspect of any practical agent is its operational efficiency. Our evaluation protocol assesses this from two complementary perspectives: the trade-off between performance and processing time, and the relationship between the length of the final report and its generation time. First, to analyze the **equivalent performance-time efficiency**, we measure the end-to-end processing time (latency) in seconds for each task. This relationship is analyzed using **Pareto Frontier** analysis, which allows us to identify agents that offer an optimal balance between speed and accuracy and citation, distinguishing them from solutions that are merely fast but inaccurate, or accurate but slow. Second, we evaluate the **generative efficiency** of the agents: for each generated report $R$, we measure its total length in tokens. By correlating the token count with the processing time, we assess the agent's **generation throughput** (tokens per second). This metric provides practical insights into the verbosity, computational cost, and user perceived latency of the different systems.

## 4 DR.Mi-Bench Annotation

Our annotation methodology is fundamentally grounded in **top-notch academic papers**, which provide the implicit yet structured clues necessary to annotate **objective evaluation metrics** for core competencies of an agent: Planning, Retrieval, and Reasoning. Based on this principle, a team of trained domain experts (senior Ph.D. candidates) manually annotate a dataset of 200 instances across 10 diverse scientific disciplines. The overall framework is illustrated in Figure 1, while the appendices provide detailed annotation guidelines (Appendix A.2), quality management protocols (Appendix A.3), and complete examples for both a research paper and a review paper (Appendix A.4).

### 4.1 Data Sources

We ground our benchmark in peer-reviewed scientific papers, as their authenticity and quality provide an ideal source of test data for evaluating DR agents designed to emulate human experts.

**Paper Selection Criteria.** The curated papers were selected based on three primary criteria to ensure quality, diversity, and fairness. (i) Recency and Impact: All papers, including any pre-prints, were required to be publicly available after 2024 and have at least 10 citations. (ii) Task Diversity: The dataset contains a balanced distribution of both *research papers*, which test an agent's ability to conduct a deep analysis of a specific research question, and *review papers*, which assess its ability to perform a broad synthesis of an entire research area. (iii) Disciplinary Fairness: The dataset is equally distributed across 10 disciplines, including materials, finance, chemistry, computer science, medicine, biology, environmental science, energy, building and construction, and Earth science. (iv) Non-STEM and Applied Scope: Beyond traditional core sciences, the benchmark explicitly incorporates non-STEM and applied domains, specifically Finance, Medicine, and Building and Construction, to ensure broad generalization capabilities. (v) Cross-Disciplinary Nature: We prioritized domains that inherently bridge multiple specialties (e.g., Materials and Finance) to assess the agents' capacity for interdisciplinary knowledge synthesis.

**Paper Selection Process.** Our semi-automated curation process involved first using the Web of Science database to filter a candidate pool by discipline, date, and citations. Domain experts then manually selected the final papers from this pool, prioritizing quality, broad sub-field coverage, and the inclusion of landmark research from top-tier venues like Nature and ICLR.

### 4.2 Query Annotation

The first step in our annotation process is the creation of a realistic research query $Q$ for each source paper, formulated based on its abstract. Each query is designed to capture the paper's core scientific contribution while strictly omitting paper-specific identifiers (e.g., author names or novel method acronyms), ensuring it represents an open-ended question a practitioner might realistically ask.

### 4.3 Annotation of Planning Module

Given the finalized query $Q$ and the source paper, domain experts construct the corresponding gold-standard plan, $\mathcal{T}^* = \langle T_1^*, \ldots, T_k^* \rangle$. The objective of this plan is to provide a high-level roadmap that

Table 2: End-to-end performance across modules on research, review, and average tasks. All results are presented as percentages.

| Model | Research | | | | Review | | | | Avg. | | | |
|---|---|---|---|---|---|---|---|---|---|---|---|---|
| | Accuracy | Precision | Recall | F1 | Accuracy | Precision | Recall | F1 | Accuracy | Precision | Recall | F1 |
| **Planning** | | | | | | | | | | | | |
| Gemini | 23.50 | 23.50 | 23.02 | 22.95 | 27.64 | 27.64 | 29.01 | 27.72 | 25.57 | 25.57 | 26.02 | 25.33 |
| **Retrieval** | | | | | | | | | | | | |
| OpenAI | 62.00 | – | – | – | 0.69 | 1.73 | 5.27 | 1.36 | 31.35 | – | – | – |
| Gemini | 72.00 | – | – | – | 1.55 | 4.00 | 7.64 | 3.00 | 36.77 | – | – | – |
| Perleixity | 66.00 | – | – | – | **3.64** | **22.85** | 9.01 | 6.88 | 34.82 | – | – | – |
| Grok | **76.00** | – | – | – | 3.62 | 13.22 | **37.51** | 6.84 | **39.81** | – | – | – |
| **Reasoning** | | | | | | | | | | | | |
| OpenAI | 60.72 | **61.58** | 40.96 | **48.27** | 56.58 | **61.59** | 38.71 | 46.63 | 58.65 | **61.59** | 39.83 | 47.45 |
| Gemini | **61.31** | 60.54 | **41.20** | 48.05 | **58.21** | 60.81 | **39.75** | **47.27** | **59.76** | 60.67 | **40.48** | **47.66** |
| Perleixity | 44.57 | 60.51 | 29.80 | 38.96 | 44.39 | 61.27 | 30.22 | 39.30 | 44.48 | 60.89 | 30.01 | 39.13 |
| Grok | 31.93 | 54.03 | 21.48 | 29.76 | 30.90 | 58.07 | 21.56 | 30.44 | 31.42 | 56.05 | 21.52 | 30.10 |
| Search-r1 | 25.49 | 35.26 | 22.69 | 27.61 | 23.02 | 27.53 | 16.38 | 20.54 | 24.26 | 31.40 | 19.53 | 24.08 |

guides an agent to reconstruct the core findings of the source paper. To mirror a typical scientist's workflow, the plan's structure generally follows the logical progression of the paper, from motivation and methods to results and conclusions. Each plan consists of 5-10 sub-tasks, requiring that they are: (i) coarse-grained, complete sentences that avoid low-level details; and (ii) concisely phrased between 8 and 20 words. To ensure annotation quality, every plan was drafted by one expert and then independently reviewed and verified by a second expert.

### 4.4 ANNOTATION OF RETRIEVAL MODULE

The annotation of the gold-standard evidence set, $\mathcal{E}^{+*}$, is designed to be objective and tailored to the task's goal of evaluating either depth or breadth. Accordingly, for **research tasks**, the source paper itself is designated as the sole piece of gold evidence. For **review tasks**, the paper's complete bibliography constitutes the ground-truth evidence set. This list of evidence, comprising identifiers such as DOIs, arXiv IDs, book titles, and webpage links, was compiled via a semi-automated process using the CrossRef and Semantic Scholar APIs, supplemented with manual curation for any missing entries to ensure comprehensive and verifiable coverage.

### 4.5 ANNOTATION OF REASONING MODULE

For the Reasoning module, experts create a set of ground-truth diagnostic pairs, $\mathcal{D} = \{(S_1, Y_1^*), \ldots, (S_w, Y_w^*)\}$, for each paper. To ensure comprehensive and structured coverage, experts were instructed to author *15-20 diagnostic claims* ($S_i$) covering five core aspects of the paper: research background, research problem, methodology, experimental results, and future directions. Each claim is a simple, unambiguous declarative statement. For each claim, the expert provides a gold boolean label ($Y_i^* \in \{\text{TRUE}, \text{FALSE}\}$) indicating whether it is supported *solely by the content of the source paper*. This diagnostic framework enables a fine-grained, objective evaluation of a generated report's factual accuracy and faithfulness.

## 5 END-TO-END EXPERIMENTS

We assess state-of-the-art, end-to-end DR systems: OpenAI's o3-deep-research (OpenAI, 2025), Google's Gemini-2.5-pro Deep Research (Google, b), Perplexity Deep Research (PerplexityAI, a), Grok Deeper Search (xAI), and Search-r1 (Jin et al., 2025b). We employ GPT-4o (OpenAI, 2024) to automate the comparison between model outputs and our ground-truth data. To mitigate potential evaluation bias, we also performed a consistency check using a diverse set of LLMs (Appendix A.5).

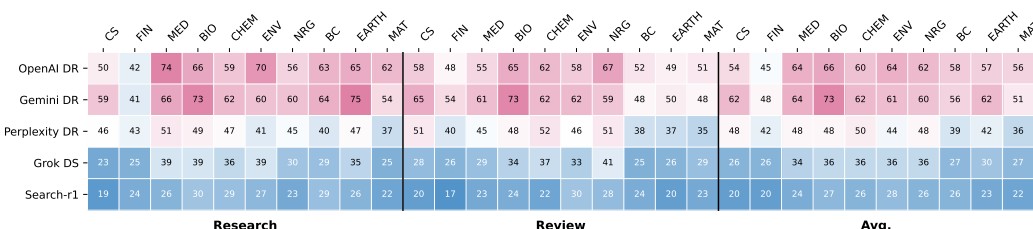

Figure 2: End-to-end DR agents performance across diverse domains: CS (Computer Science), FIN (Finance), MED (Medicine), BIO (Biology), CHEM (Chemistry), ENV (Environmental Science), NRG (Energy), BC (Building and Construction), EARTH (Earth Science), and MAT (Materials).All results are presented as percentages.

## 5.1 WHAT ARE THE MODULAR STRENGTHS AND WEAKNESSES OF END-TO-END DR AGENTS?

The main results, presented in Table 2 and Table 11 , reveal that end-to-end scientific DR remains a significant challenge for state-of-the-art agent. The different DR agents exhibit specialized strengths, with different agents excelling in different core modules, while overall performance indicates considerable opportunity for future advancements. We provide a qualitative analysis of high- and low-scoring cases in Appendix A.9 to offer deeper insights into specific model behaviors and failure modes.

**Planning.** The task of breaking down a research question into sub-tasks proves to be extremely difficult. Among the evaluated agents, Gemini is the only one with an explicit planning module that exposes its sub-tasks for evaluation. It achieves a modest total F1 score of only 25.33, highlighting the ability of scientific research question decomposition.

**Retrieval.** The Retrieval module reveals a dramatic performance gap between different task. While models achieve respectable Accuracy of 62-76% on focused research papers, their performance collapses on comprehensive review papers, with Accuracy plummeting to below 4%. Existing DR agents still exhibit significant limitations in searching comprehensive and precise coverage of representative literature for a sub-field. Among the models, Grok achieves the highest overall retrieval Accuracy (39.81), primarily driven by its strong performance on research paper task.

**Reasoning.** In the Reasoning module, models demonstrate relatively stronger capabilities, though scores are still moderate. A clear top tier emerges, with Gemini and OpenAI delivering the best performance. A nuanced trade-off is visible between them: Gemini consistently leads in overall Accuracy (59.76) and Recall (40.48), while OpenAI excels in Precision (61.59). In stark contrast, models like Grok and Perplexity lag significantly in this area, indicating that strong retrieval skills do not necessarily translate to effective reasoning and synthesis.

Our analysis highlights a complex performance landscape where none of single agent masters all modules: while Gemini stands out as the strongest overall due to its superior reasoning, its explicit planning capabilities are still nascent, and all agents, including the top retriever Grok, struggle significantly with retrieval on complex review papers.

## 5.2 HOW DOES AGENT PERFORMANCE DIFFER ACROSS SPECIFIC DOMAINS?

Our analysis of end-to-end DR agents reveals a clear performance hierarchy dependent on the scientific domain (Figure 2). We consistently observe the highest performance in fields like Medicine and Biology, which is likely attributable to the extensive representation of these domains in the models' training corpora. In contrast, agents struggle significantly more in specialized areas such as Finance and Materials, which are likely less prevalent in the training data. This performance gap indicates that the combined challenge of retrieval and reasoning varies considerably across different disciplines. This performance disparity reveals a significant bias towards data-rich fields and calls for a concerted research effort to bridge the capability gap for these underperforming but equally important scientific disciplines.

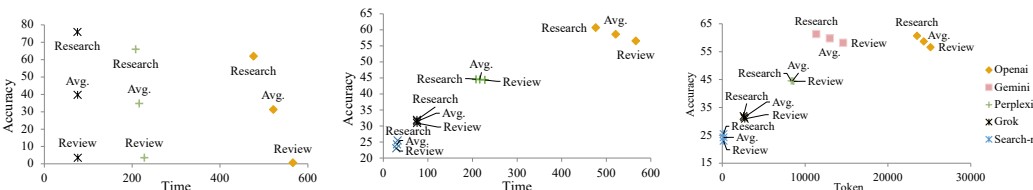

(a) Time-accuracy trade-off for the retrieval module.

(b) Time-accuracy trade-off for the reasoning module.

(c) Token-accuracy trade-off for the reasoning module.

Figure 3: Efficiency of retrieval and reasoning modules. All results are presented as percentages.

## 5.3 WHAT ARE THE PERFORMANCE-COST TRADE-OFFS IN DEEP RESEARCH AGENTS?

We analyze the efficiency of each end-to-end agent by examining its two core modules: retrieval and reasoning, aiming to maximize accuracy while minimizing latency and output tokens. This analysis, framed by the concept of the Pareto Frontier, reveals completely different efficiency landscapes within these critical stages of the DR process, as shown in Figure 3.

**Accuracy-Time.** The retrieval module's efficiency landscape does not present a balanced trade-off. Grok is the sole agent on the Pareto frontier, as it dominates other systems by being both significantly faster (~76s) and more accurate (39.81%) in retrieval, rendering competitors Pareto-inefficient (Figure 3a). In contrast, the reasoning module exhibits a classic speed-accuracy trade-off, with all evaluated agents lying on the Pareto Frontier (Figure 3b). This creates a balanced curve where users can choose between the highest-accuracy model, OpenAI (~59% Accuracy), and the fastest option, Search-r1 (~30s).

**Accuracy-Token Trade-off.** Our analysis of the relationship between reasoning accuracy and report length (token count) reveals a complex, non-linear trade-off. A strong positive correlation between verbosity and accuracy holds for most models, but the top performers show a key difference in token efficiency. As shown in Figure 3c, Gemini delivers the highest accuracy (~59.76%) from a concise ~13k token report, while OpenAI is nearly twice as verbose for no performance gain. This suggests that Gemini is significantly more token-efficient and that excessive verbosity does not guarantee superior performance. Below this top tier, other models like Perplexity and Grok exhibit a simpler positive correlation where shorter reports are progressively less accurate.

In summary, our efficiency analysis reveals distinct trade-offs for each module. For retrieval, Grok is the objectively optimal choice due to its superior speed and accuracy. For reasoning, however, the choice is subjective, requiring a balance along two clear frontiers: a speed-accuracy trade-off where slower models are more accurate, and an accuracy-verbosity trade-off where longer, more detailed reports achieve higher scores.

## 5.4 DOES OUR METRIC EFFECTIVELY ALIGN WITH HUMAN PREFERENCES?

To validate our automatic metrics against human judgment, we conducted a human evaluation study where domain experts performed pairwise comparisons on 100 blinded report pairs. Experts selected the superior report based on an overall assessment of its factuality, comprehensiveness, and coherence. Our primary finding is a high degree of alignment: the ranking produced by our automatic Reasoning F1 score matched the human expert's preference in 86.23% of the comparisons. This strong agreement validates our benchmark as a reliable proxy for what experts consider a high-quality research report.

## 5.5 DO AGENTS OVERESTIMATE THEIR CAPABILITIES THROUGH DATA LEAKAGE?

**Forward-citation Leakage.** *"Forward citations" (papers published after the source) could distort evaluation metrics by artificially lowering retrieval scores or inflating reasoning capabilities through information leakage.* To address this, we implement a temporal filtering experiment by explicitly adding publication date constraints to the agent prompts. As shown in Table 3 (detailed in Table 9), the overall performance shift is minimal (Retrieval Accuracy from 25.18% to 25.27%), confirming that forward citation leakage has a negligible impact and validating the robustness of our results.

Table 3: Impact of Forward-citation Leakage on OpenAI DR: Post-Cutoff vs. Full (Unrestricted). All results are presented as percentages.

| Model | Accuracy | Precision | Recall | F1 |
|---|---|---|---|---|
| **Retrieval** | | | | |
| Post-Cutoff | 25.27 | – | – | – |
| Full | 25.18 | – | – | – |
| **Reasoning** | | | | |
| Post-Cutoff | 59.41 | 60.18 | 39.34 | 46.66 |
| Full | 59.94 | 59.30 | 39.74 | 46.56 |

Table 4: Impact of Cut-off Dates Leakage in Gemini DR: Post-Cutoff vs. Full (Unrestricted). All results are presented as percentages.

| Model | Accuracy | Precision | Recall | F1 |
|---|---|---|---|---|
| **Retrieval** | | | | |
| Post-Cutoff | 37.15 | – | – | – |
| Full | 36.77 | – | – | – |
| **Reasoning** | | | | |
| Post-Cutoff | 63.97 | 60.97 | 43.33 | 49.91 |
| Full | 59.76 | 60.67 | 40.48 | 47.66 |

Table 5: Planning performance of foundational LLMs potential as DR backbones. Performance is shown across Research, Review, and Avg. subsets. All results are presented as percentages.

| Model | Research | | | | Review | | | | Avg. | | | |
|---|---|---|---|---|---|---|---|---|---|---|---|---|
| | Accuracy | Precision | Recall | F1 | Accuracy | Precision | Recall | F1 | Accuracy | Precision | Recall | F1 |
| GPT-o3 | 27.24 | 27.24 | 24.10 | 25.52 | 29.48 | 29.48 | 25.70 | 27.37 | 28.36 | 28.36 | 24.90 | 26.45 |
| Gemini-2.5-pro | **31.31** | **31.31** | **26.14** | **28.42** | **31.22** | **31.22** | **25.91** | **28.25** | **31.27** | **31.27** | **26.02** | **28.34** |
| Sonar-reasoning-pro | 29.41 | 29.41 | 26.13 | 27.58 | 26.93 | 26.93 | 23.56 | 25.08 | 28.17 | 28.17 | 24.85 | 26.33 |
| Grok-4 | 23.75 | 23.75 | 20.73 | 22.08 | 29.98 | 29.98 | 26.59 | 28.10 | 26.87 | 26.87 | 23.66 | 25.09 |
| Claude-3.7-sonnet | 24.12 | 24.12 | 21.83 | 22.86 | 26.15 | 26.15 | 24.60 | 25.29 | 25.14 | 25.14 | 23.21 | 24.07 |

**Cut-off Dates Leakage.** *We investigated the potential impact of data leakage relative to model knowledge cut-offs.* We explicitly validated the performance of Gemini DR on a strictly temporally constrained subset of papers published after March 2025, which post-dates the model's release. As presented in Table 4, performance on this unseen subset is comparable to, or even slightly higher than, the unrestricted baseline.

### 5.6 IS THE BENCHMARK SATURATED TO TEST-TIME COMPUTE?

*The benchmark is effectively unsaturated.* We conducted pass@k experiments (k=1, 3) on a 20-instance subset. As shown in Figure 4 (detailed in Table 10), a consistent performance gain from pass@1 to pass@3 across all modules. It indicates that the benchmark successfully measures the benefit of increased test-time compute and is not merely limited by rigid evaluation formats. Even with increased compute (pass@3), the highest average Reasoning score is 68.49% (OpenAI), and Planning remains around 40%. This substantial gap from 100% confirms that the benchmark is effectively unsaturated and remains a challenging standard for future, more capable agents.

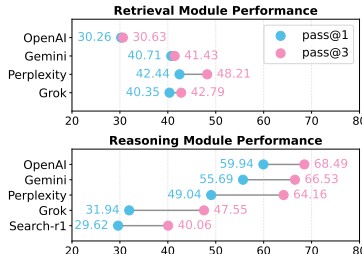

Figure 4: Performance scaling with test-time compute.

## 6 ISOLATED EXPERIMENTS

This section extends our analysis to the foundational LLMs, assessing their potential as the core backbones for DR agents. This includes prominent models such as GPT-o3 (OpenAI, a), Gemini-2.5-Pro (Google, a), Perplexity-sonar-pro (PerplexityAI, b), Grok-4 (OpenAI, b), Claude-3.7-sonnet (Claude), as well as models from the Llama (Grattafiori et al., 2024) (Llama-3.2-1B/3B-Instruct, Llama-3.1-8B-Instruct) and Qwen Instruct (Team, 2024) (Qwen2.5-1.5B/3B/7B-Instruct) series. To isolate their intrinsic capabilities, our analysis bypasses the retrieval module, often a separate, tool-based component.

### 6.1 IS HIGH-LEVEL PLANNING A KEY BOTTLENECK FOR FOUNDATIONAL LLMS?

*Our experimental results strongly suggest that high-level research planning currently represents a key bottleneck for even the most advanced foundational LLMs.* As shown in Table 5, no single model

Table 6: Reasoning performance of foundational LLMs potential as DR backbones. Performance is shown across Research, Review, and Avg. subsets. All results are presented as percentages.

| Model | Research | | | | Review | | | | Avg. | | | |
|---|---|---|---|---|---|---|---|---|---|---|---|---|
| | Accuracy | Precision | Recall | F1 | Accuracy | Precision | Recall | F1 | Accuracy | Precision | Recall | F1 |
| GPT-o3 | **85.14** | **63.70** | **56.91** | **59.70** | 59.72 | 60.11 | 40.79 | 47.78 | 72.43 | 61.91 | 48.85 | 53.74 |
| Gemini-2.5-pro | 69.09 | 61.69 | 46.88 | 52.79 | **87.17** | **63.02** | **57.90** | **59.99** | **78.13** | **62.36** | **52.39** | **56.39** |
| Sonar-reasoning-pro | 83.91 | 62.98 | 55.44 | 58.33 | 62.55 | 62.13 | 42.45 | 49.71 | 73.23 | 62.56 | 48.95 | 54.02 |
| Grok-4 | 82.89 | 63.50 | 55.04 | 58.38 | 54.43 | 60.88 | 37.16 | 45.17 | 68.66 | 62.19 | 46.10 | 51.77 |
| Claude-3.7-sonnet | 84.76 | 63.17 | 56.52 | 59.22 | 61.56 | 61.66 | 41.86 | 49.15 | 73.16 | 62.42 | 49.19 | 54.18 |

demonstrates a clear advantage. Instead, all evaluated systems are clustered in a narrow and low performance range, with the top model, Gemini-2.5-pro, achieving a modest average F1 score of only 28.34. The primary reason for this universal struggle is a consistent failure to achieve high Recall across all models. This shared bottleneck indicates that while current LLMs can generate a few relevant sub-tasks, they are not yet capable of producing the comprehensive, multi-step conceptual roadmaps required for complex research. This lack of reliable, high-level planning capability is therefore a critical limiting factor and a crucial area for future innovation.

## 6.2 WHAT IS THE PERFORMANCE HIERARCHY AMONG LLMS FOR REASONING MODULE?

*Our analysis of the reasoning module reveals a clear performance hierarchy, where Gemini-2.5-pro distinguishes itself from a pack of otherwise closely-performing models through its unique ability to handle complex, broad-scope documents.* As shown in Table 6, Gemini-2.5-pro leads with an average F1 score of 56.39. The key capability that separates the top performer from the rest is its performance on complex review papers. Gemini-2.5-pro is the only model whose F1 score significantly improves on these broader tasks (rising from 52.79 on research to 59.99 on review). In stark contrast, every other model in the evaluation experiences a notable performance degradation on review papers, with F1 scores dropping by 8-13 points. We provide a detailed comparison of open- and closed-source LLMs in Appendix A.7. This drop is primarily driven by a sharp fall in Recall, suggesting that while most models can reason effectively over a focused source, they struggle to synthesize the wide-ranging information presented in a literature review.

## 7 RELATED WORK

**Early work on search agents**, such as ReAct (Yao et al., 2023), established the paradigm of interleaving reasoning with tool use for multi-hop question answering (Li et al., 2025a; Jin et al., 2025a; Xiong et al., 2025; Wang et al., 2024; Xu et al., 2025; Song et al., 2025; Hao et al., 2025). While this line of work has been extended by commercial systems for more complex tasks (OpenAI, 2025; Google, b; PerplexityAI, a; xAI), they are either not designed for long-form synthesis (Zhang et al., 2025b) or their evaluations are opaque and non-diagnostic. Similarly, **existing benchmarks for DR** are also limited; early systems like GAIA and BrowseComp focus on isolated retrieval or short-answer generation (Mialon et al., 2023; Wei et al., 2025; Zhou et al., 2025; Phan et al., 2025; FutureSearch et al., 2025). Crucially, both lines of work fail to disentangle the distinct agent capabilities of planning, retrieval, and reasoning (see Appendix A.10 for more details).

## 8 CONCLUSION

To address critical gaps in the evaluation of DR agents, we introduce Dr.Mi-Bench, a human-annotated scientific benchmark, and Dr.Mi-Eval, a novel paradigm for the modular assessment of planning, retrieval, and reasoning. Our evaluation of state-of-the-art systems reveals a fragmented performance landscape where agents exhibit specialized strengths but also share critical weaknesses. By enabling the diagnostic analysis of such actionable failure modes, our work provides a crucial tool to guide the development of more reliable and trustworthy academic research assistants.

## 9 ETHICS STATEMENT

Our benchmark is constructed from publicly available, highly cited research papers across ten scientific disciplines. We ensure that all included materials are properly licensed or fall under fair use for academic research purposes. Nevertheless, several ethical considerations remain. First, as with any dataset derived from published literature, biases present in the source materials (e.g., disciplinary imbalance, geographic or linguistic concentration) may propagate into the benchmark. Second, while the benchmark focuses on scientific and technical domains, the reasoning outputs of DR agents could be misused to generate misleading or fabricated analyses if deployed irresponsibly. To mitigate these risks, we (i) release clear data statements describing the collection process, (ii) provide transparent evaluation protocols to discourage cherry-picking of results, and (iii) encourage responsible use of our benchmark for advancing trustworthy and reliable research assistants.

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

# A APPENDIX

## A.1 DEEP RESEARCH AGENT MODULE-WISE DEFINITIONS

**Definition 3 (Planner $\pi$)** *Given a query $Q$, the planning module $\pi$ yields a structured plan:*

$$\mathcal{T} = \pi(Q),$$

*where the plan $\mathcal{T}$ consists of an ordered list of sub-tasks $\mathcal{T} = \langle T_1, T_2, \ldots, T_k \rangle$. The objective is to decompose a potentially complex $Q$ into a covering and well-ordered set of sub-tasks.*

**Definition 4 (Retriever $\rho$)** *Given the query $Q$, plan $\mathcal{T}$, toolset $\mathcal{H}$, and budget $\mathcal{B}$, the retrieval module $\rho$ operates within a corpus $\mathcal{E}$ and returns an attributed evidence set:*

$$\mathcal{E}^+ = \rho(Q, \mathcal{T}, \mathcal{H}, \mathcal{B}) = \left\{ \left( T_i, \{e_{i1}, \ldots, e_{iN_i}\} \right) \right\}_{i=1}^k,$$

*where each evidence item $e_{ij} \in \mathcal{E}$ is a document or passage with verifiable provenance. The objective is to gather relevant, sufficient, and diversified evidence for each sub-task $T_i$ under the budget $\mathcal{B}$.*

**Definition 5 (Reasoner $\sigma$)** *Given the query $Q$, plan $\mathcal{T}$, and the attributed evidence set $\mathcal{E}^+$, the reasoning module $\sigma$ produces the final report and its justification:*

$$R = \sigma(Q, \mathcal{T}, \mathcal{E}^+),$$

*where $R$ is a structured report. The objective is to maximize the factual correctness, logical coherence, and faithfulness of the report $R$ to the provided evidence $\mathcal{E}^+$.*

## A.2 ANNOTATION GUIDELINE

---

**An Guideline of Annotation**

**Query Annotation Guideline**

Generate a professional meta-question that require detailed and comprehensive responses.

1. The query should only based on the content of given paper.

2. The query should not include the paper title or authors directly.

3. It should be an open-ended question that related to the paper.

4. The query should contain part or all aspects: background, research problem, methodology, experimental results, and contributions.

5. The query should be a user may asked before reading the given paper.

**Plan Annotation Guideline**

You are given a research question and an academic paper.
Your job is to list the high-level sub-tasks needed to answer the question based on the paper.
Requirements:

1. Identify 5-10 sub-tasks.

2. Coarse-grained only: write complete sentences but avoid low-level details such as specific numbers, datasets, instruments, figure/panel references, or citation markers.

3. Keep each sentence concise (about 820 words).

**Diagostics Annotation Guideline**

You are given a research question and an academic paper.
Your job is to list the diagnostic statements and gold answers which should cover the whole paper.
Requirements:

1. Identify 15-20 diagnostics.

2. Coverage: Statements should cover all key aspects of given paper.

3. Accuracy: Boolean answers corresponding to statements are used to verify the accuracy of model generated reports.

---

### A.3 ANNOTATION QUALITY MANAGEMENT

**Annotator Selection Criteria.** All annotators are PhD candidates from the relevant field. We recruited experts for all 10 domains, and each annotator exclusively annotated papers within their own domain of expertise, ensuring high-quality annotation. We verified expertise in two ways. First, all our annotators are PhD candidates from top universities who have published in top-tier venues within their specific fields. Second, all experts completed a rigorous training and calibration phase. This required them to analyze papers using a standardized rubric and conduct a cross-review, which aligned their understanding and ensured annotation consistency.

**Pre-annotation Training and Pilot Study.** To ensure high-quality and consistent annotations, all domain experts underwent a training and calibration phase. Annotators were first required to analyze a shared set of pilot papers and produce summaries based on a standardized rubric (Appendix A.4). This exercise, followed by a cross-review, served to align their understanding of what constitutes a paper's core claims, ensuring a shared basis for all annotation tasks.

**In-process Dual Annotation and Adjudication.** Our core quality control mechanism is a rigorous protocol of dual annotation and adjudication, which was applied to all four annotation stages (Query, Plan, Evidence, and Diagnostics). For every data instance, two domain experts independently produced an annotation. A third, senior annotator then compared the two versions and served as the final adjudicator, resolving any disagreements through structured discussion to produce a single consensus artifact. We conducted an inter-annotator agreement assessment where two annotators

independently labeled the diagnostic answers (True/False) based on given papers. The agreement rate is 96.05%, with a Cohens $\kappa$ of 0.89, indicating strong consistency.

**Post-annotation Validation and Maintenance.** Following the consensus annotation, we performed a final validation pass to filter for any potentially offensive or biased content and to verify the fidelity of all provenance links and citations. The benchmark is versioned, and we have scheduled periodic maintenance to reflect stable corrigenda in the source papers, ensuring the long-term integrity and utility of the dataset. The frozen evaluation corpus and gold artifacts remain versioned; updates are documented to preserve comparability across releases.

## A.4 EXAMPLES

**An example of the summary writing.**

---

**A Summary Example**

**DOI**

10.1016/j.conbuildmat.2024.138507

**Summary**

This paper addresses the dual challenge of high embodied carbon in concrete and shortages of natural aggregates by exploring biochar as a carbon-negative resource for construction. It targets the problem that direct addition of biochar often degrades concrete mechanics, proposing instead a cold-bonded coreshell aggregate (biochar core; OPC/GGBS cementitious shell) and evaluating it via physical tests (loose bulk density, crushing strength, water absorption), hydration calorimetry, XRD/TGA, SEM/BSE, X-CT, and structural concrete trials with carbon-footprint accounting. The MG80 variant achieved a loose bulk density of $789kg/m^3$, crushing strength of $6.84MPa$, water absorption of $19.4\%$, and superior strength efficiency ($8669Pam^3/kg$); concrete with $90\%$ replacement by MG80 maintained structural-grade performance (42.2 MPa, $1866kg/m^3$). Carbon analysis shows the aggregate can be carbon-negative (e.g., MG80 $\approx -69kgCO_2$ per ton), outperforming commercial sintered lightweight aggregates. The work contributes a scalable route to high-volume biochar utilization that preserves structural performance, reduces lifecycle emissions, and alleviates reliance on mined aggregates, advancing sustainable concrete and the building sectors decarbonization.

---

**An example of the research task annotation.**

---

**An Example of The Research Task Annotation**

**DOI**

10.1016/j.conbuildmat.2024.138507

**Domain**

Building and Construction

**Task**

Research

**Research Question**

How can biochar-enabled carbon-negative core-shell aggregates contribute to sustainability and carbon neutrality in the construction industry in terms of background, research gaps, motivation, methods, experiments, performance, and carbon footprint?

**Plan**

**Sub-task 1:** Summarize the high carbon emissions and natural resource shortages in the construction industry.

---

**Sub-task 2:** Explain why biochar is a promising solution and outline current challenges.

**Sub-task 3:** Describe the coreshell aggregate design method and its application to biochar encapsulation.

**Sub-task 4:** Provide an overview of material selection and mix design, including biochar aggregates and concrete.

**Sub-task 5:** Summarize measurements of aggregate properties to show lightweight density, adequate strength, and controlled water absorption.

**Sub-task 6:** Present hydration and microstructural evidence supporting optimized shell design and coreshell integrity.

**Sub-task 7:** Assess concrete performance when replacing natural aggregates with biochar-CSA across different substitution levels.

**Sub-task 8:** Detail the carbon footprint assessment scope, assumptions, and resulting emissions for biochar-CSA production.

**Sub-task 9:** Compare strength efficiency and emissions of biochar-CSA against conventional lightweight aggregates.

**Sub-task 10:** Discuss broader sustainability benefits and contributions to carbon neutrality in construction.

**Evidence**

10.1016/j.conbuildmat.2024.138507

**Diagnostics**

**Statement 1:** The construction industry accounts for more than 30% of global production-related $CO_2$ emissions.
**Answer:** True

**Statement 2:** There is a limit to the direct use of biochar in concrete because large amounts of biochar can negatively affect the mechanical properties of concrete.
**Answer:** True

**Statement 3:** The cold-bonding method used in this study does not require high-temperature processing.
**Answer:** False

**Statement 4:** The core-shell structure of biochar-CSA is designed to overcome the high water absorption issue of biochar.
**Answer:** True

**Statement 5:** This study evaluates the physical properties of biochar-CSA, such as crushing strength and water absorption.
**Answer:** True

**Statement 6:** Using ground blast furnace slag (GGBS) in the shell material reduces the carbon footprint of biochar-CSA.
**Answer:** True

**Statement 7:** The biochar-CSA achieves a crushing strength higher than 6 MPa after 28 days of curing.
**Answer:** True

**Statement 8:** The biochar-CSA does not have a water absorption rate exceeding 50% after 28 days of curing.
**Answer:** False

**Statement 9:** The use of biochar-CSA in concrete reduces its density and compressive strength compared to natural aggregate concrete.
**Answer:** True

**Statement 10:** Replacing OPC with GGBS can reduce the carbon emissions of biochar-CSA production to negative values.

**Answer:** True

**Statement 11:** Using biochar-CSA in concrete contributes to energy-efficient building construction.

**Answer:** True

**Statement 12:** The cold-bonding method used in this study is not more carbon-intensive than conventional sintering methods for artificial lightweight aggregate production.

**Answer:** False

**Statement 13:** The study proposes a high-volume utilization solution for biochar in concrete while maintaining concretes structural performance.

**Answer:** True

**Statement 14:** The construction industry facing a shortage of natural resources (such as natural river sand).

**Answer:** True

**Statement 15:** The biochar concrete developed using biochar-CSA in this study have wide application potential and help achieve carbon neutrality of concrete.

**Answer:** True

**An example of the review task annotation.**

### An Annotation Example of The Review Task

**DOI**

10.1016/j.conbuildmat.2024.135602

**Domain**

Building and Construction

**Task**

Review

**Research Question**

How can the ultra-high-performance concrete (UHPC) be optimized to lower carbon emissions by reviewing its research background, motivation, challenges, methodologies, carbon footprint, and future directions?

**Plan**

**Sub-task 1:** Summarize the characteristics, applications, and carbon reduction drivers of UHPC.

**Sub-task 2:** Define carbon footprint assessment framework using LCA boundaries and indicators of embodied carbon emissions (EC), embodied energy consumption (EE), and embodied CO2 index (CI).

**Sub-task 3:** Identify environmental hotspots in UHPC mixes, highlighting binders, aggregates, and steel fibers.

**Sub-task 4:** Summary of cementitious material alternatives for reducing the environmental impact of UHPC.

**Sub-task 5:** Evaluate recycled aggregates and powders for environmental benefits and performance trade-offs.

**Sub-task 6:** Assess the impact on emissions and mechanics of replacing conventional steel fibers with scrap fibers.

**Sub-task 7:** Establish decision rules for ranking strategies by EC, EE, and CI reductions.

**Sub-task 8:** Discuss challenges: data standardization, durability, waste variability, and processing burdens.

**Sub-task 9:** Propose future directions and implementation roadmap for scalable, durable, low-carbon UHPC.

### Evidence

1. 10.1016/j.conbuildmat.2024.135602
2. 10.1007/s11356-021-13435-2
3. 10.1007/s11367-013-0614-0
4. . . .

### Diagnostics

**Statement 1:** UHPC is characterized by ultra-high compressive strength, high toughness, and excellent durability.
**Answer:** True

**Statement 2:** Due to optimized material usage, UHPC has a lower carbon footprint than conventional concrete (NC).
**Answer:** False

**Statement 3:** UHPC is widely used in marine environments due to its low permeability and high durability.
**Answer:** True

**Statement 4:** The main challenge in developing low-carbon UHPC is to reduce carbon emissions without compromising compressive strength.
**Answer:** True

**Statement 5:** Replacing steel fibers with waste fibers is the most effective way to reduce the carbon footprint of UHPC.
**Answer:** False

**Statement 6:** The use of waste aggregates greatly reduces the environmental impact of UHPC.
**Answer:** False

**Statement 7:** Life Cycle Assessment (LCA) can be used to evaluate the environmental impacts of UHPC.
**Answer:** True

**Statement 8:** The LCA analysis in the paper considers both material-level and structural-level impacts.
**Answer:** False

**Statement 9:** The CI value is calculated as the total embodied $CO_2$ emissions divided by the compressive strength of the UHPC mixture.
**Answer:** True

**Statement 10:** Geopolymer-based UHPC (UHPGC) has the lowest environmental impact among all UHPC types.
**Answer:** True

**Statement 11:** Cement replacement with supplementary cementitious materials (SCMs) is less effective than aggregate replacement in reducing carbon emissions.
**Answer:** False

**Statement 12:** Utilizing waste fibers, such as waste steel fiber (WSF), can significantly reduce the energy consumption of UHPC.
**Answer:** True

Table 7: Full planning performance of foundational LLMs potential as DR backbones, including open- and closed-source LLMs. Performance is shown across Research, Review, and Avg. subsets.

| Model | Research | | | | Review | | | | Avg. | | | |
|---|---|---|---|---|---|---|---|---|---|---|---|---|
| | Accuracy | Precision | Recall | F1 | Accuracy | Precision | Recall | F1 | Accuracy | Precision | Recall | F1 |
| **Closed-source LLMs** | | | | | | | | | | | | |
| GPT-o3 | 27.24 | 27.24 | 24.10 | 25.52 | 29.48 | 29.48 | 25.70 | 27.37 | 28.36 | 28.36 | 24.90 | 26.45 |
| Gemini-2.5-pro | **31.31** | **31.31** | **26.14** | **28.42** | **31.22** | **31.22** | 25.91 | **28.25** | **31.27** | **31.27** | **26.02** | **28.34** |
| Sonar-reasoning-pro | 29.41 | 29.41 | 26.13 | 27.58 | 26.93 | 26.93 | 23.56 | 25.08 | 28.17 | 28.17 | 24.85 | 26.33 |
| Grok-4 | 23.75 | 23.75 | 20.73 | 22.08 | 29.98 | 29.98 | **26.59** | 28.10 | 26.87 | 26.87 | 23.66 | 25.09 |
| Claude-3.7-sonnet | 24.12 | 24.12 | 21.83 | 22.86 | 26.15 | 26.15 | 24.60 | 25.29 | 25.14 | 25.14 | 23.21 | 24.07 |
| **Open-source LLMs** | | | | | | | | | | | | |
| Qwen2.5-1.5B | 17.08 | 17.08 | 12.71 | 6.98 | 19.75 | 19.75 | 14.03 | 10.71 | 18.42 | 18.42 | 13.37 | 8.85 |
| Qwen2.5-3B | 13.60 | 13.60 | 13.82 | 9.47 | 18.18 | 18.18 | 18.80 | 16.18 | 15.89 | 15.89 | 16.31 | 12.83 |
| Qwen2.5-7B | **23.73** | **23.73** | 16.58 | 13.56 | 22.16 | 22.16 | 16.14 | 13.17 | **22.95** | **22.95** | 16.36 | 13.37 |
| Llama-3.2-1B | 13.39 | 13.39 | 11.47 | 6.70 | 18.67 | 18.67 | 17.83 | 15.00 | 16.03 | 16.03 | 14.65 | 10.85 |
| Llama-3.2-3B | 18.00 | 18.00 | 17.79 | 14.05 | 17.70 | 17.70 | 17.78 | 13.48 | 17.85 | 17.85 | 17.79 | 13.77 |
| Llama-3.1-8B | 20.90 | 20.90 | **20.88** | **16.29** | 24.40 | 24.40 | 24.82 | 21.39 | 22.65 | 22.65 | **22.85** | **18.84** |

> **Statement 13:** Future research should focus on incorporating durability properties into LCA of UHPC.
> **Answer:** True
>
> **Statement 14:** UHPC developed using limestone calcined clay cement (LC3) achieves both low carbon emissions and high compressive strength.
> **Answer:** False
>
> **Statement 15:** The combined use of carbon emissions, energy consumption, and compressive strength is sufficient to assess the environmental impact of UHPC.
> **Answer:** False
>
> **Statement 16:** The availability of waste fibers is a limiting factor for their large-scale application in UHPC.
> **Answer:** True
>
> **Statement 17:** The use of nanomaterials, such as graphene oxide, is a cost-effective way to enhance the mechanical properties of UHPC.
> **Answer:** False

## A.5 How Consistent Are Different LLMs When Used as Evaluators?

To ensure that our automated evaluation is robust and not biased by the choice of a single evaluator model, we conducted a consistency analysis between two powerful, independent LLMs. We used both GPT-4o and Gemini-2.5-pro as separate, blind judges to perform pairwise comparisons on 100 randomly sampled pairs of agent-generated reports. For each pair, both LLM judges were tasked with selecting the superior report based on our established criteria of factuality, comprehensiveness, and coherence. The results show a very high level of inter-model agreement, with the two evaluators making the same choice (i.e., preferring the same report) in an average of 93.00% (standard deviation = 3.42%) of the cases across these runs. This high consistency validates top-tier LLMs as reliable proxies for objective evaluation, confirming the robustness of our proposed Dr.Mi-Eval paradigm.

## A.6 How Do Closed- and Open-Source LLMs Compare as Planning Backbones?

*Our evaluation of foundational LLMs as planning backbones reveals two key findings: (1) a significant performance gap exists between closed- and open-source models, and (2) high-level research planning represents a profound challenge for all current models, regardless of their origin or scale.*

**Performance of Closed-Source LLMs.** While outperforming their open-source counterparts, the closed-source models are all clustered in a narrow and low performance range, indicating a universal struggle with the task. As shown in Table 7, Gemini-2.5-pro is the marginal leader with an average F1

Table 8: Full reasoning performance of foundational LLMs potential as DR backbones, including open- and closed- source LLMs. Performance is shown across Research, Review, and Avg. subsets.

| Model | Research | | | | Review | | | | Avg. | | | |
|---|---|---|---|---|---|---|---|---|---|---|---|---|
| | Accuracy | Precision | Recall | F1 | Accuracy | Precision | Recall | F1 | Accuracy | Precision | Recall | F1 |
| Closed-source LLMs | | | | | | | | | | | | |
| GPT-o3 | **85.14** | **63.70** | **56.91** | **59.70** | 59.72 | 60.11 | 40.79 | 47.78 | 72.43 | 61.91 | 48.85 | 53.74 |
| Gemini-2.5-pro | 69.09 | 61.69 | 46.88 | 52.79 | **87.17** | **63.02** | **57.90** | **59.99** | **78.13** | **62.36** | **52.39** | **56.39** |
| Sonar-reasoning-pro | 83.91 | 62.98 | 55.44 | 58.33 | 62.55 | 62.13 | 42.45 | 49.71 | 73.23 | 62.56 | 48.95 | 54.02 |
| Grok-4 | 82.89 | 63.50 | 55.04 | 58.38 | 54.43 | 60.88 | 37.16 | 45.17 | 68.66 | 62.19 | 46.10 | 51.77 |
| Claude-3.7-sonnet | 84.76 | 63.17 | 56.52 | 59.22 | 61.56 | 61.66 | 41.86 | 49.15 | 73.16 | 62.42 | 49.19 | 54.18 |
| Open-source LLMs | | | | | | | | | | | | |
| Qwen2.5-1.5B | 27.74 | 54.52 | 19.07 | 27.26 | 18.02 | 45.20 | 12.99 | 19.32 | 22.88 | 49.86 | 16.03 | 23.29 |
| Qwen2.5-3B | 48.81 | 60.94 | 33.12 | 42.13 | 27.70 | 55.90 | 19.60 | 28.12 | 38.26 | 58.42 | 26.36 | 35.13 |
| Qwen2.5-7B | **56.64** | **62.74** | **38.15** | **46.36** | **32.82** | **57.06** | **23.32** | **32.17** | **44.73** | **59.90** | **30.73** | **39.27** |
| Llama-3.2-1B | 20.16 | 50.48 | 13.93 | 21.13 | 12.31 | 38.50 | 8.88 | 13.86 | 16.24 | 44.49 | 11.41 | 17.49 |
| Llama-3.2-3B | 32.59 | 57.10 | 22.42 | 31.06 | 17.61 | 45.92 | 12.66 | 19.01 | 25.10 | 51.51 | 17.54 | 25.04 |
| Llama-3.1-8B | 35.36 | 58.86 | 24.12 | 33.15 | 19.12 | 51.42 | 13.86 | 21.01 | 27.24 | 55.14 | 18.99 | 27.08 |

score of only 28.34. The other models, including GPT-o3 (26.45) and Sonar-reasoning-pro (26.33), perform at a very similar level. A consistent pattern across this group is that their Precision scores are significantly higher than their Recall scores. This points to a shared failure mode: the models generate a small number of correct sub-tasks but fail to produce a comprehensive plan that covers all necessary steps.

**Performance of Open-Source LLMs.** The open-source models operate at a considerably lower performance level, with the top performer, Llama-3.1-8B, achieving an F1 score of just 18.84. Within this category, clear scaling laws are evident, as larger models in both the Llama and Qwen families consistently outperform their smaller counterparts. The primary bottleneck for these models is a catastrophic failure in comprehensiveness. The massive gap between their Precision scores (e.g., 22.65 for Llama-3.1-8B) and their F1 scores (18.84) is driven by extremely low Recall. This demonstrates that while they may generate a few relevant plan steps, they are largely incapable of creating a sufficiently complete roadmap for a complex research query.

A.7    HOW DO CLOSED- AND OPEN-SOURCE LLMs COMPARE AS REASONING BACKBONES?

*Our evaluation of foundational LLMs in an isolated reasoning setting reveals a stark performance gap between the top-tier closed-source models and their current open-source counterparts.* As shown in Table 8, while a competitive hierarchy exists among the proprietary models, even the best open-source LLMs operate in a significantly lower performance bracket, primarily due to a shared difficulty with recall and handling complex, broad-scope synthesis tasks.

**Performance of Closed-Source LLMs.** Among the closed-source models, a clear performance hierarchy emerges. Gemini-2.5-pro establishes itself as the state-of-the-art, achieving the highest average F1 score (56.39). It is followed by a pack of strong contenders including Claude-3.7-sonnet (54.18), sonar-reasoning-pro (54.02), and GPT-o3 (53.74). The key capability that separates Gemini-2.5-pro from the rest is its exceptional performance on complex Review papers, where its F1 score increases to 59.99. In stark contrast, every other closed-source model experiences a significant performance degradation on Review tasks. This suggests that while most top models are adept at reasoning over focused content, the ability to synthesize broad, multi-document information is a rare and differentiating capability.

**Performance of Open-Source LLMs.** The open-source models, while promising, perform at a considerably lower level. Within this group, performance scales predictably with model size, demonstrating clear scaling laws. The Qwen2.5-7B (39.27 F1) is the top performer, significantly outpacing smaller models in its family and the Llama series. However, all open-source models share a critical weakness: a severe drop in performance on Review tasks, which is even more pronounced than that of the closed-source models. The primary reason for this is extremely low Recall. This

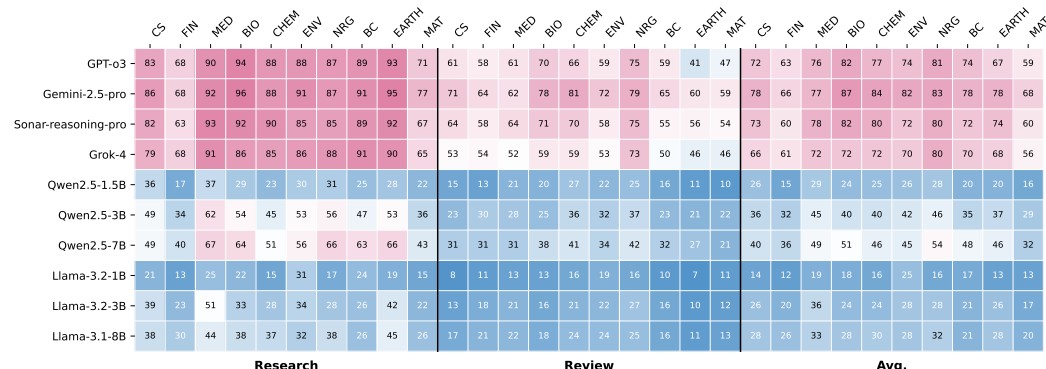

Figure 5: Foundational LLMs performance across diverse domains. The heatmap visualizes the accuracy scores of various models (rows) across 10 scientific and technical domains (columns), where color intensity corresponds to performance. The domains are: **CS** (Computer Science), **FIN** (Finance), **MED** (Medicine), **BIO** (Biology), **CHEM** (Chemistry), **ENV** (Environmental Science), **NRG** (Energy), **BC** (Building and Construction), **EARTH** (Earth Science), and **MAT** (Materials).

reveals a systemic failure to synthesize sufficient evidence, undermining the comprehensiveness essential for high-quality scientific reasoning.

## A.8 HOW DOES THE INTRINSIC REASONING POTENTIAL OF AN LLM BACKBONE VARY ACROSS DIFFERENT DOMAINS?

*Our analysis of foundational LLMs in a reasoning-only setting reveals that the domain performance hierarchy persists even when retrieval is bypassed, with models like o3 scoring highest in Computer Science and lowest in Earth Science , proving that reasoning complexity is domain-intrinsic.* As shown in Figure 5, even when supplied with ground-truth evidence, LLMs exhibit the same performance patterns across domains observed in the end-to-end tests. Computer Science remains the domain where models achieve the highest reasoning scores, followed by fields like Medicine and Biology. The crucial insight comes from confirming that this hierarchy persists when the retrieval challenge is removed. For example, the top-performing o3 scores highest in Computer Science (83%) and among its lowest in Earth Science (67%). This strongly implies that the performance variation across domains stems not only from differing retrieval difficulties but also from the intrinsic complexity of reasoning and synthesis required within each scientific field.

## A.9 CASE STUDY

**A case study of a high-score report.**

### A Case Study of A High-score Report

**DOI**

10.1038/s41467-024-45147-9

**Domain**

Biology

**Performance**

**Module 1:** Planning Acc 42.86%

**Module 2:** Retrieval Acc 100.00%

**Module 3:** Reasoning Acc 94.11%

**Task**

Research

**Research Question**

How can smart bioelectronic materials and systems address the challenges in regenerative medicine and tissue engineering, and what are their potential applications across various tissue types?

**Gold Plan**

**Subtask 1:** Identify shortcomings of current tissue adhesives in versatility, manufacturability, and customization.

**Subtask 2:** Summarize the graft interpenetrating polyurethanepoly(acrylic acid) network enabling robust, stretchable adhesion.

**Subtask 3:** Explain how NHS ester functionalization creates covalent bonds with tissue amines under wet conditions.

**Subtask 4:** Describe the benign-solvent ink formulation and rheological tuning that enable direct-ink-write printing.

**Subtask 5:** Outline how printed mesh architectures provide conformability, porosity, and programmable mechanical anisotropy.

**Subtask 6:** Evaluate cytocompatibility and hydrolytic degradability using in vitro assays and subcutaneous implantation.

**Subtask 7:** Demonstrate ex vivo sealing performance and in vivo sutureless repair in tracheal and colonic models.

**Subtask 8:** Analyze the design and efficacy of liquid-infused patches for blood-resistant tissue sealing.

**Subtask 9:** Illustrate integration into bioelectronic and drug-delivery devices enabled by multimaterial 3D printing.

**Subtask 10:** Assess potential clinical translation, customization benefits, and broader healthcare impact of the platform.

**Predicted Plan**

**Sub-task 1:** Conduct a systematic search across specified academic databases (PubMed, arXiv, Semantic Scholar, Science, Nature, IEEE Xplore) for peer-reviewed literature on "3D printable tissue adhesives," "bio-adhesive inks," and "direct-ink-write bioprinting for tissue repair."

**Sub-task 2:** From the retrieved papers, identify and synthesize the stated challenges of conventional tissue adhesives, such as cytotoxicity, weak adhesion on wet surfaces, and mechanical mismatch with biological tissues.

**Sub-task 3:** Analyze the innovative material design of 3D printable adhesives. Detail the chemical composition, including base polymers, cross-linking agents, and functional additives that enable both adhesion and printability.

**Sub-task 4:** Investigate the principles of the direct-ink-write (DIW) 3D printing process as applied to these biomaterials. Explain how this technology facilitates the creation of customized, patient-specific adhesive structures.

**Sub-task 5:** Find and extract performance data from in vitro and ex vivo experimental models presented in the literature. Focus on key metrics like adhesive strength (lap shear, burst pressure), biocompatibility (cell viability), and degradation kinetics.

**Sub-task 6:** Locate studies demonstrating the efficacy of these adhesives in in vivo animal models. Summarize the outcomes related to wound closure, tissue regeneration, and host immune response.

**Sub-task 7:** Evaluate the broader contributions of this technology by synthesizing information on its potential to advance tissue repair, create novel biomedical devices, and address unmet clinical and societal healthcare needs.

**Sub-task 8:** Consolidate all findings into a comprehensive report, ensuring every factual claim is directly supported by a citation from a relevant academic paper, including its DOI or arXiv ID."

**Predicted Evidence**

1. 10.1038/s41467-024-45147-9
2. 10.1002/SMMD.20220033
3. 10.1016/j.nantod.2014.09.006
4. . . .

**Gold Evidence**

1. 10.1038/s41467-024-45147-9

**Diagnostics**

**Statement 1:** The construction industry accounts for more than 30% of global production-related $CO_2$ emissions.
**Answer:** True

**Statement 2:** There is a limit to the direct use of biochar in concrete because large amounts of biochar can negatively affect the mechanical properties of concrete.
**Answer:** True

**Statement 3:** The cold-bonding method used in this study does not require high-temperature processing.
**Answer:** False

**Statement 4:** The core-shell structure of biochar-CSA is designed to overcome the high water absorption issue of biochar.
**Answer:** True

**Statement 5:** This study evaluates the physical properties of biochar-CSA, such as crushing strength and water absorption.
**Answer:** True

**Statement 6:** Using ground blast furnace slag (GGBS) in the shell material reduces the carbon footprint of biochar-CSA.
**Answer:** True

**Statement 7:** The biochar-CSA achieves a crushing strength higher than 6 MPa after 28 days of curing.
**Answer:** True

**Statement 8:** The biochar-CSA does not have a water absorption rate exceeding 50% after 28 days of curing.
**Answer:** False

**Statement 9:** The use of biochar-CSA in concrete reduces its density and compressive strength compared to natural aggregate concrete.
**Answer:** True

**Statement 10:** Replacing OPC with GGBS can reduce the carbon emissions of biochar-CSA production to negative values.
**Answer:** True

**Statement 11:** Using biochar-CSA in concrete contributes to energy-efficient building construction.
**Answer:** True

**Statement 12:** The cold-bonding method used in this study is not more carbon-intensive than conventional sintering methods for artificial lightweight aggregate production.

**Answer:** False

**Statement 13:** The study proposes a high-volume utilization solution for biochar in concrete while maintaining concretes structural performance.

**Answer:** True

**Statement 14:** The construction industry facing a shortage of natural resources (such as natural river sand).

**Answer:** True

**Statement 15:** The biochar concrete developed using biochar-CSA in this study have wide application potential and help achieve carbon neutrality of concrete.

**Answer:** True

**Report**

---

The Convergence of Material Science and Additive Manufacturing: A Comprehensive Analysis of 3D Printable Tissue Adhesives for Advanced Biomedical Applications

Section 1: The Clinical Imperative for Advanced Tissue Adhesives

The closure of surgical incisions and the repair of damaged tissues are fundamental challenges in medicine, with over 300 million surgical procedures performed globally each year.1 For centuries, mechanical methods such as sutures and staples have been the standard of care. While effective in providing high tensile strength, these traditional techniques are increasingly recognized as suboptimal, particularly in the context of modern minimally invasive surgery and delicate tissue repair.2 This has driven the development of tissue adhesives and sealants as promising alternatives. However, conventional adhesives have been beset by a persistent set of limitations related to biocompatibility, performance in wet environments, and mechanical compatibility with living tissues.3 The shortcomings of these first- and second-generation technologies have created a clear and urgent clinical imperative for a new class of tissue repair materials that can overcome these fundamental challenges through innovations in both material chemistry and manufacturing.

1.1 Limitations of Traditional Mechanical Closure Methods (Sutures and Staples)

Mechanical closure methods, despite their ubiquity, are inherently traumatic. The process of passing a needle and thread through tissue causes secondary damage, which can lead to heightened inflammation, an increased risk of postoperative infection, potential nerve damage, and the formation of scar tissue.2 The application of sutures is a time-consuming process that demands a high level of surgical skill and precision, especially in anatomically complex or difficult-to-access regions of the body.2 These challenges are magnified in sensitive procedures such as ocular surgeries, vascular anastomosis, or nerve repair, where the physical stress of suturing can compromise surgical outcomes.3 Surgical staples were introduced as a faster alternative, particularly for skin lacerations, and have been associated with lower infection rates and shorter healing times compared to sutures.6 However, they are not without significant drawbacks. The high tensile strength of metal staples can cause patients considerable pain during their eventual removal, and meticulous application is required to avoid improper wound closure and subsequent scarring.6 For both sutures and staples, their application in minimally invasive surgeries is exceptionally challenging, limiting the advancement of less traumatic surgical techniques.3 The collective disadvantages of these mechanical methodstissue damage, procedural difficulty, and risk of complicationsunderscore the need for less invasive, easier-to-use, and more biocompatible alternatives for joining and sealing tissues.2

1.2 Deficiencies of Conventional Tissue Adhesives and Sealants

The development of tissue adhesives represents a significant step toward overcoming the limitations of mechanical closures. However, existing adhesives, which are typically formulated as amorphous glues or hydrogels, have introduced a new set of challenges that have limited their clinical versatility and efficacy.1 These deficiencies can be categorized into three primary areas: poor performance in physiological environments, biocompatibility concerns, and a fundamental mismatch with the properties of biological tissues. A primary failure

mode for many conventional adhesives is their inability to form strong, durable bonds in the presence of water and bodily fluids. This wet adhesion challenge is a critical barrier to internal applications.[3] Naturally derived adhesives, such as fibrin-based glues, are known for their poor adhesion to wet tissues and low mechanical strength, rendering them unsuitable for many load-bearing surgical procedures.[3] Even synthetic adhesives with higher intrinsic strength see their performance degrade significantly in wet conditions.[5] Furthermore, biocompatibility and cytotoxicity remain persistent concerns. Early synthetic adhesives, most notably cyanoacrylates ("super glues"), were found to elicit severe inflammatory responses and cytotoxicity, which has largely restricted their use to external, topical applications.[3] To improve strength, composite glues were developed that use crosslinking agents like glutaraldehyde; however, the presence of unreacted aldehydes can cause localized toxicity.[3] Even naturally derived materials are not without issue, as they can carry the risk of immunogenicity, pro-inflammatory potential, and, in the case of human-derived products like fibrin glue, disease transmission.[3] Perhaps the most significant limitation is the mechanical mismatch between conventional adhesives and the soft, dynamic tissues of the human body. Most biological tissues are soft, elastic, and anisotropic, whereas many high-strength adhesives are rigid and brittle.[3] This disparity in mechanical properties prevents the adhesive from deforming with the tissue, leading to high stress concentrations at the bond interface, eventual delamination, and failure of the repair. This issue is compounded by the "one-size-fits-all" nature of existing products. Conventional glues and hydrogels are supplied with a fixed form factor and a static set of properties, offering no capacity for customization to match the unique geometry of a specific wound or the distinct mechanical behavior of a particular tissue type.[4] The persistent challenges facing conventional adhesives reveal a fundamental engineering conflict. Attempts to maximize one property, such as adhesive strength through highly reactive chemistry, often come at the cost of another, such as biocompatibility. For example, strong cyanoacrylates are cytotoxic, while biocompatible fibrin glues are mechanically weak.[3] This suggests that the ideal solution cannot be a simple material but must be an integrated system. The consistent failure of uniform, amorphous materials highlights that the physical form and architecture of the adhesive are as critical as its chemical composition. This realization points toward the necessity of a new technological paradigm that combines advanced material science with a manufacturing process capable of precise architectural control, thereby addressing the multifaceted requirements for successful tissue repair.

Technology Mechanism of Action Key Advantages Key Limitations Sutures & Staples Mechanical approximation of tissue edges using foreign materials (thread, metal). High tensile strength; Established standard of care. - Tissue Trauma: Causes secondary injury, inflammation, scarring.[2]

- Wet Adhesion: Not applicable (mechanical). - Mechanical Match: Creates high stress points; poor match for soft tissue compliance. - Biocompatibility: Foreign body reaction; risk of infection.[2]

- Versatility: Difficult to use in delicate tissues and minimally invasive surgery.[3] Conventional Adhesives Chemical bonding (covalent, ionic) or physical entanglement at the tissue interface. Easy and fast to apply; Less traumatic than sutures.[2] - Tissue Trauma: Generally low, but some components can be cytotoxic.[6]

- Wet Adhesion: Often poor; strength degrades significantly in physiological fluids.[3]
- Mechanical Match: Typically poor; rigid adhesives mismatch with soft, dynamic tissues.[3]
- Biocompatibility: Risk of cytotoxicity, inflammation, and immunogenicity.[3]
- Versatility: "One-size-fits-all" form factor; cannot be customized for specific wound geometries or functionalities.[1] 3D Printable Adhesives Multi-modal: rapid physical bonding followed by robust covalent bonding. Customizable architecture; Tunable mechanical properties; Strong wet adhesion. - Tissue Trauma: Minimal, designed for biocompatibility.[4]
- Wet Adhesion: Specifically designed for rapid and robust adhesion to wet tissues.[1]
- Mechanical Match: Can be programmed via 3D printed architecture to match target tissue properties.[1]
- Biocompatibility: Formulated with biocompatible polymers; demonstrated low inflammation in vivo.[4]
- Versatility: Enables fabrication of patient-specific, application-specific devices with complex geometries.[1]

Table 1: Comparative Analysis of Tissue Closure Technologies. This table summarizes the advantages and limitations of traditional methods versus the novel approach of 3D printable adhesives, highlighting how the latter is designed to overcome the key deficiencies of its predecessors.

Section 2: Engineering a Printable Bioadhesive: Innovations in Material Chemistry and Rheology

The development of a 3D printable tissue adhesive requires a sophisticated approach to material design, one that can simultaneously satisfy the demanding and often contradictory requirements of high-performance biological adhesion and the specific rheological properties necessary for additive manufacturing. The solution lies in a multi-component polymer system where each constituent is selected to perform a distinct function—adhesion, mechanical toughness, or printability—and is integrated through a carefully controlled synthesis process. This approach effectively decouples the material's properties in its printable "ink" state from its final "functional" state, overcoming the limitations that have constrained previous generations of adhesives.

2.1 Core Polymer Network Design

The foundation of the innovative bio-adhesive is a graft interpenetrating polymer network (IPN) meticulously composed of two primary polymers: a hydrophilic polyurethane (PU) and poly(acrylic acid) (PAA).4 The synthesis is achieved through a one-pot photoinitiated polymerization process. A precursor solution containing acrylic acid (AA) monomer, hydrophilic PU, and photoinitiators is exposed to ultraviolet (UV) light.1 Two initiators, benzophenone and -ketoglutaric acid, play crucial roles. Under UV irradiation, benzophenone, a Type II photoinitiator, generates radical sites along the PU backbone. These sites initiate the polymerization of AA, resulting in PAA chains that are covalently grafted to the PU polymer. Simultaneously, -ketoglutaric acid initiates the polymerization of free AA monomers in the solution, forming a separate PAA network that becomes physically entangled with the PU-g-PAA structure, thus creating the graft IPN.4 Following polymerization and purification via dialysis to remove unreacted reagents, the PU-PAA network is functionalized to impart its adhesive properties. This is achieved by reacting the carboxylic acid groups on the PAA component with N-hydroxysuccinimide (NHS) using 1-Ethyl-3-(3-dimethylaminopropyl)carbodiimide (EDC) as a coupling agent. This step converts a fraction of the carboxylic acid groups into highly reactive NHS esters, yielding the final PU-PAA-NHS polymer.1

2.2 The Multi-Modal Adhesion Mechanism

The robust performance of the adhesive stems from a multi-modal mechanism that combines rapid physical bonding, strong covalent chemical bonding, and high bulk mechanical toughness to resist failure. This strategy is biomimetic, mirroring natural adhesion systems that often employ a combination of fast-acting, reversible interactions and slower-forming, permanent bonds. Rapid Physical Adhesion: The key to the adhesive's ability to bond to wet tissue within seconds lies in the hydrophilic nature of the PAA component. When the dry, 3D printed adhesive patch makes contact with a hydrated tissue surface, the high density of charged carboxylic acid groups on the PAA chains drives rapid absorption of interfacial water.4 This process is further accelerated by the capillary action of the porous mesh structure created during 3D printing. The swift removal of the boundary water layer allows the adhesive to achieve intimate contact and consolidate with the tissue surface, forming strong intermolecular bonds such as hydrogen bonds almost instantaneously.1 Robust Chemical Adhesion: While physical bonds provide initial tack, long-term, durable adhesion is achieved through the formation of stable covalent bonds. The reactive NHS ester groups, which were grafted onto the PAA chains, readily react with primary amine groups (RNH2) that are abundant on tissue surfaces, primarily from the lysine residues of proteins in the extracellular matrix. This reaction forms a strong amide bond, covalently linking the adhesive patch to the tissue.1 This chemical cross-linking at the interface ensures the adhesion is robust and can withstand physiological forces over extended periods. Bulk Mechanical Toughness: Strong interfacial adhesion is meaningless if the adhesive material itself fails cohesively under stress. The mechanical integrity, stretchability, and toughness of the adhesive are primarily provided by the elastomeric PU network. The hard segments within the PU polymer chains interact via dynamic, reversible hydrogen bonds. When the material is subjected to mechanical strain, these bonds can break and reform, providing an efficient mechanism for dissipating energy.4

This prevents crack propagation within the adhesive, allowing it to stretch and deform with the underlying tissue without catastrophic failure.

2.3 Engineering Rheology for Direct-Ink-Write (DIW) Printing

A central innovation of this technology is the formulation of the adhesive polymer into a viscoelastic ink suitable for extrusion-based 3D printing. This presents a significant challenge, as the properties that make a good adhesive (e.g., high viscosity, tackiness) are generally antithetical to those required for a printable ink (e.g., controlled flow).4 This challenge is overcome by creating a temporary, printable state through careful formulation of the ink's rheology. The PU-PAA-NHS polymer is dissolved in an aqueous ethanol solution (typically 70Shear-Thinning Behavior: The ink is a non-Newtonian fluid whose viscosity decreases under applied shear stress. Inside the narrow printing nozzle, the ink is subjected to high shear rates, causing its viscosity to drop and allowing it to be extruded smoothly and continuously.9 This behavior arises from the alignment of polymer chains and the temporary disruption of their entangled network under flow. Yield Stress: The ink possesses a well-defined yield stress, meaning it behaves as a solid-like gel below a certain stress threshold.4 Once the ink exits the nozzle, the shear stress is removed, and its viscosity rapidly recovers. The material "sets" almost immediately, retaining the shape of the printed filament. This high yield stress is crucial for maintaining structural integrity and enabling the layer-by-layer stacking required to build complex 3D architectures without the structure collapsing under its own weight.4 This intelligent material design, which separates the requirements for printing from the final application, is the key to the technology's success. The solvent creates a transient, printable ink state. The DIW process then imparts the desired architecture. Finally, a simple drying step removes the solvent, converting the printed object into its final, solid, functional form as a dry adhesive patch, ready for clinical application where its adhesive properties are activated by contact with wet tissue.

Component Chemical Type / Structure Primary Function(s) Polyurethane (PU) Hydrophilic, thermoplastic elastomer. - Mechanical: Provides bulk toughness, stretchability, and elasticity. Dissipates energy via dynamic hydrogen bonds to prevent cohesive failure.4

- Biocompatibility: Serves as a biocompatible backbone for the polymer network.4 Poly(acrylic acid) (PAA) Hydrophilic polymer with dense carboxylic acid groups. - Adhesion (Physical): Drives rapid interfacial water absorption from wet tissue, enabling fast consolidation and formation of intermolecular bonds.4 - Adhesion (Chemical): Serves as the scaffold for grafting reactive NHS ester groups. N-Hydroxysuccinimide (NHS) Ester Reactive functional group grafted onto PAA chains. - Adhesion (Chemical): Forms strong, stable covalent amide bonds with primary amines on tissue surfaces, ensuring long-term adhesion.1 Ethanol/Water Solvent Benign solvent mixture (e.g., 70- Printability: Dissolves the PU-PAA-NHS polymer to create a viscoelastic ink with the appropriate rheological properties (shear-thinning, yield stress) for DIW printing.4

Table 2: Functional Components of the 3D Printable Bio-adhesive Ink. This table deconstructs the bio-ink to illustrate how each chemical component contributes a specific function, highlighting the rational design that enables both printability and high-performance adhesion.

Section 3: Direct-Ink-Write (DIW) 3D Printing: A Platform for Customized Bio-architectures

While the innovative chemistry of the bio-adhesive ink provides the potential for superior performance, it is the application of Direct-Ink-Write (DIW) 3D printing that unlocks this potential, transforming the material into functional, customized medical devices. DIW serves as the critical enabling technology that bridges the gap between advanced material science and personalized medicine. It addresses the fundamental limitations of conventional adhesivesnamely, their lack of versatility and mechanical mismatch with tissueby providing unparalleled control over the device's macro- and micro-architecture.

3.1 Principles of DIW for Biomaterials

DIW is an extrusion-based additive manufacturing technique in which a computer-controlled motion stage guides a micro-nozzle to deposit a viscoelastic ink layer-by-layer, meticulously constructing a three-dimensional object from a digital blueprint.9 Its primary advantages in the biomedical field are its material versatility, low cost, and high resolution. Unlike other 3D printing methods that are often restricted to specific classes of materials (e.g., photopolymers for stereolithography), DIW can be adapted to print a vast range of custom-formulated inks, including polymer solutions, hydrogels, and even cell-laden bio-inks for tissue engineering.9

This flexibility makes it an ideal platform for processing novel biomaterials like the PU-PAA-NHS adhesive ink.

3.2 From CAD Model to Patient-Specific Device

The DIW process fundamentally shifts the paradigm of medical device manufacturing from mass production to on-demand personalization. The fabrication workflow begins not with a physical mold, but with a digital file, typically a computer-aided design (CAD) model.14 This digital-first approach has profound implications for clinical practice. Patient-specific anatomical data, acquired from standard medical imaging modalities such as computed tomography (CT) or magnetic resonance imaging (MRI), can be converted into a 3D digital model of a wound or tissue defect.14 This model then serves as the template for designing a bespoke adhesive patch or implant that is perfectly contoured to the individual's unique anatomy.18 This capability directly confronts the "one-size-fits-all" limitation of conventional adhesives.4 Instead of a surgeon having to manually trim a generic patch to approximate the shape of a complex liver laceration, for example, a perfectly fitting device can be fabricated pre-operatively. This not only promises a better seal and improved function but also has the potential to reduce surgical time and improve overall outcomes.18 Furthermore, DIW provides precise control over the device's internal architecture. The diameter of the printed filaments, the size and shape of the pores within a mesh, and the orientation of successive layers can all be programmed into the printing path.4 This architectural control is not merely for shaping the device but is a powerful tool for engineering its functional properties.

3.3 Programming Mechanical Properties through Structural Design

The most transformative aspect of using DIW for this application is the ability to program the bulk mechanical properties of the adhesive patch through its geometric design, independent of the underlying material chemistry. This allows the creation of "meta-materials," where the overall behavior of the structure is dictated by its architecture. This capability provides a direct and elegant solution to the critical problem of mechanical mismatch that plagues conventional rigid adhesives.3 For instance, by printing the adhesive ink in a mesh-like pattern, the stress-strain response of the final patch can be precisely tuned by simply altering the angle () between the intersecting filaments in the CAD model.1 A patch printed with a smaller angle will exhibit greater stretchability at lower strains, while a patch with a larger angle will be stiffer. This means that a surgeon could request an adhesive patch for lung repair that needs to be highly compliant to accommodate breathing, or a patch for tendon repair that requires high stiffness and strength. Both devices could be printed from the exact same bio-ink, with their profoundly different mechanical profiles achieved solely by modifying the digital design file. This uncouples the material's chemistry from the device's mechanical function, allowing each to be optimized independently. The resulting patches are not only mechanically tuned but are also highly flexible and conformable, ensuring they can maintain intimate contact with the complex, dynamic surfaces of biological tissues.4 The convergence of patient-specific imaging, digital design, and rapid DIW fabrication creates a powerful new workflow for personalized medicine at the point of care. A patient's medical scan can be used to design a custom-fit, mechanically-tuned adhesive device that can be manufactured on-demand in a hospital's fabrication facility and sterilized for use in surgery, potentially within hours.22 This represents a fundamental shift in surgical practice, moving away from generic implants and toward truly individualized therapeutic solutions.

Section 4: Performance Characterization in Preclinical Models

The translation of a novel biomaterial from concept to clinical potential requires rigorous validation through a hierarchy of experimental models. The 3D printable tissue adhesive has been subjected to a comprehensive suite of tests, ranging from ex vivo benchtop characterization of its mechanical and adhesive properties to in vivo studies in animal models to assess its real-world efficacy, biocompatibility, and degradation profile. The collective data from these preclinical evaluations demonstrates that the adhesive not only meets but often exceeds the performance of existing technologies, validating the innovative design principles behind its material chemistry and 3D printed architecture.

4.1 Ex Vivo Mechanical and Adhesion Profiling

To quantify the adhesive's performance under controlled conditions, a series of standardized mechanical tests were conducted on excised porcine tissues, which serve as a reliable surrogate for human tissues.4 These tests are crucial for establishing baseline performance metrics and

for direct comparison against commercial products. Adhesion Strength: The adhesive's ability to bond to various tissues was measured using lap-shear and burst pressure tests.7 In lap-shear tests, two pieces of tissue are overlapped and bonded with the adhesive, and the force required to shear them apart is measured. Burst pressure tests are used to evaluate the sealing capability of the adhesive, where a patch is used to seal a defect in a tissue (like an intestine or artery), and the internal pressure is increased until the seal fails or leaks. The 3D printed adhesives have demonstrated exceptionally robust adhesion to a wide range of soft and hard tissues, including skin, heart, liver, kidney, and bone.23 In many cases, the measured adhesive strength has been shown to be superior to that of commercially available clinical adhesives.23 Mechanical Properties: Uniaxial tensile testing confirmed that the printed patches are highly stretchable and conformable, consistent with the properties of their elastomeric PU base.4 Crucially, these tests also validated the principle of architectural control over mechanical properties. Patches printed with different geometric patterns exhibited distinct and predictable stress-strain curves, confirming that the material's mechanical response can be tuned to match the requirements of different biological tissues.1 Performance in Challenging Conditions: A key differentiator for this technology is its performance in clinically realistic, challenging environments. The adhesive maintains strong adhesion even when applied underwater, simulating the fluid-filled surgical field.24 A particularly significant innovation is the development of a liquid-infused patch designed for hemostasis on actively bleeding tissuesa scenario where most adhesives fail catastrophically.3 By infusing the 3D printed porous mesh with a biocompatible, hydrophobic liquid like oleic acid, the patch can effectively repel blood from the tissue surface. When gentle pressure is applied, the hydrophobic liquid is displaced through the pores, clearing a path for the adhesive to make direct contact with the underlying tissue and form a rapid, robust seal.1 The porous architecture is critical to this function; it allows for higher burst pressures to be achieved at lower application pressures compared to a simple oil-coated, non-porous surface, demonstrating a true synergy between the material's chemistry and its printed form.1

4.2 In Vivo Efficacy and Biocompatibility in Animal Models

Following successful ex vivo characterization, the adhesive was evaluated in living animal models to assess its real-world performance, safety, and long-term fate. These studies, conducted in rats under approved animal care protocols, provided compelling evidence of the technology's clinical potential.4 Fluid-Tight Tissue Sealing: The adhesive patches were tested in several challenging surgical defect models: Trachea Repair: In a model of tracheal injury, a 2 mm oval defect was created in the rat trachea, causing an air leak. The 3D printed patch was applied and, within 10 seconds, formed a complete, air-tight seal, restoring normal ventilation. The animals were monitored for 4 weeks, after which histological analysis confirmed that the defect had repaired successfully under the patch with no signs of leakage or significant tracheal narrowing.4 Colon Repair: A 5 mm full-thickness incision was made in the rat colon, a high-risk site for leakage and infection. The patch conformed to the curved surface of the colon and provided a fluid-tight seal, preventing any leakage of bowel contents. After 4 weeks, the patch remained adhered to the repair site, and the colonic wall had healed without any signs of abscesses or other complications.4 Hemostasis: The efficacy of the liquid-infused patch was confirmed in vivo. It was used to seal actively bleeding wounds, including a deep incision in the liver and a snip injury to the femoral artery. In both cases, the patch achieved complete hemostasis within seconds of application, demonstrating its potential for use in trauma and hemorrhagic situations.4 Biocompatibility and Host Response: The body's reaction to an implanted material is a critical determinant of its clinical viability. To assess this, the 3D printed adhesive material was implanted subcutaneously in rats. Histological evaluation at 2 and 4 weeks post-implantation revealed only a mild inflammatory response, which was comparable to that elicited by a commercially available and clinically used hemostatic patch (Tachosil) that served as a control.4 This finding indicates that the material possesses favorable biocompatibility, a prerequisite for any implantable medical device.26 Biodegradability: An ideal tissue engineering construct should provide temporary support and then gradually degrade as the body replaces it with new, native tissue.29 The in vivo studies showed that the 3D printed adhesive patch undergoes slow, gradual hydrolytic degradation over time. At the 4-week time point in both the subcutaneous and organ defect models, the implanted patch was still present but was partially degraded, with a noticeable decrease in

volume.4 This controlled degradation profile is a desirable feature, suggesting that the material can provide sufficient support during the critical healing phase before being safely resorbed by the body. The successful repair observed in the trachea and colon models indicates that the adhesive functions not just as a passive sealant but as a temporary, biocompatible scaffold that creates a protected environment conducive to the body's natural healing processes, bridging the gap between simple wound closure and regenerative medicine.31

Performance Metric Test Model Key Result Commercial Comparator Citation(s) Lap-Shear Adhesion Ex vivo porcine skin Robust shear strength, tunable by composition. Outperforms many commercial bioadhesives. 23 Burst Pressure Ex vivo porcine skin Significantly higher burst pressure than control materials. Higher than normal arterial blood pressure. 23 Hemostasis In vivo rat liver & femoral artery Achieved complete sealing of actively bleeding injuries within 10 seconds. N/A 4 Air-Tight Sealing In vivo rat trachea defect Formed an air-tight seal within 10 seconds; maintained for >4 weeks. N/A 4 Fluid-Tight Sealing In vivo rat colon defect Formed a fluid-tight seal within 10 seconds; maintained for >4 weeks. N/A 4 Biocompatibility In vivo rat subcutaneous implant Mild inflammatory response at 2 and 4 weeks. Comparable to commercial Tachosil patch. 4 Degradation In vivo rat subcutaneous implant Gradual hydrolytic degradation with decreased volume observed at 4 weeks. Slower degradation than commercial Tachosil patch. 4

Table 3: Summary of Preclinical Performance Metrics. This table consolidates the key quantitative and qualitative performance data from ex vivo and in vivo studies, providing a clear, evidence-based summary of the 3D printable adhesive's efficacy and safety profile.

Section 5: Expanding the Clinical Horizon: Applications Beyond Wound Sealing

The development of a 3D printable, biocompatible, and strongly adherent material represents more than just an incremental improvement in wound closure technology. It constitutes the creation of a versatile platform technology with the potential to address a fundamental and pervasive challenge across numerous fields of biomedical engineering: the stable, non-traumatic interfacing of synthetic devices with soft, wet, and dynamic biological tissues. By solving this core problem, the 3D printable adhesive can serve as an enabling foundation for a new generation of bio-integrated devices, from advanced diagnostics to targeted therapeutics.

5.1 A Robust Interface for Bio-integrated Devices

Many advanced biomedical technologies, including flexible bioelectronics, implantable sensors, and localized drug delivery systems, have been limited in their clinical translation by the lack of a reliable method to attach them to the target tissue. Sutures are too damaging for delicate electronics and can create stress points that lead to device failure. Conventional adhesives often lack the required biocompatibility, mechanical compliance, or long-term adhesion in a physiological environment. The 3D printable adhesive platform directly addresses these limitations, offering a solution that is simultaneously strong, flexible, biocompatible, and customizable.1 It can function as a "universal adapter" or "interfacing layer," providing a stable mechanical and electrical connection between rigid or flexible synthetic systems and the soft tissues of the body. This capability facilitates a paradigm shift in device design, moving from simple implantationwhere a device is merely placed within the bodyto true bio-integration, where the device is mechanically and functionally coupled with the body. Traditional rigid implants are often recognized as foreign objects and are encapsulated by a layer of fibrous scar tissue, effectively isolating them from the surrounding biological environment.27 In contrast, a device fabricated with or attached via the 3D printable adhesive, with its programmed mechanical compliance and porous architecture, is designed to move with the native tissue, minimizing chronic inflammation and mechanical stress at the interface. The potential for cellular infiltration into the porous structure further blurs the line between the synthetic construct and the host tissue, paving the way for true hybrid bionic systems.20

5.2 Proof-of-Concept Applications

The versatility of the DIW platform has been demonstrated through the fabrication of several proof-of-concept devices that highlight its broad applicability beyond simple wound sealing. Bioelectronic Patches: By integrating multiple printing steps, it is possible to fabricate flexible, multi-layered bioelectronic devices. For example, a base layer of the adhesive material can be printed, followed by the deposition of a conductive ink (e.g., silver-based) to create intricate circuits, electrodes, or sensor arrays.4 An insulating layer can then be printed on top to protect the electronics. This approach has been used to create adhesive patches with embedded

LEDs that can be applied to an ex vivo porcine heart. The adhesive ensures stable, conformal contact, allowing for reliable electrical stimulation or recording of physiological signals like an electrocardiogram (ECG) directly from the surface of the beating heart.4 Drug Delivery Devices: The inherent porosity of the 3D printed mesh structures makes them ideal candidates for use as localized drug delivery depots. The pores can be loaded with therapeutic agents, such as analgesics for post-operative pain management, antibiotics to prevent infection, or growth factors to accelerate tissue regeneration. The architecture of the printed patchincluding its porosity, filament thickness, and overall geometrycan be precisely designed to control the release kinetics of the loaded drug, enabling sustained, targeted delivery directly to the site of injury or disease.4 This approach maximizes therapeutic efficacy while minimizing systemic side effects. Customized Wound Dressings and Scaffolds: The technology provides a powerful tool for creating advanced wound dressings that are tailored to the specific needs of a patient and a particular wound type.36 For chronic wounds, such as diabetic foot ulcers, a dressing can be printed to perfectly match the wound's complex geometry based on a 3D scan.16 The dressing's porous architecture can be optimized to manage wound exudate, while the bio-ink can be formulated with antimicrobial agents to combat infection or other bioactive molecules to promote healing.22 This level of personalization and multi-functionality is unattainable with conventional wound care products. These examples illustrate that the 3D printable adhesive is not merely a single product but a foundational technology. Its most significant contribution may be its role as a platform that solves the universal challenge of soft tissue interfacing, thereby unlocking the potential of countless other biomedical innovations that have been stalled by this critical bottleneck.

Section 6: Future Perspectives and Societal Impact

The development of 3D printable tissue adhesives represents a significant milestone in biomedical engineering, establishing a powerful synergy between material science and additive manufacturing. Looking forward, this technology is poised to evolve beyond its current applications, driving further innovations in surgical practice and personalized medicine. However, the path to widespread clinical adoption involves surmounting significant technical, regulatory, and logistical challenges. The successful navigation of this path promises to have a profound societal and economic impact on healthcare delivery, improving patient outcomes and enabling a new era of data-driven, individualized treatment.

6.1 Emerging Frontiers: In Situ Bioprinting

The next logical and most exciting frontier for this technology is in situ bioprintingthe direct printing of adhesive materials, scaffolds, or even cell-laden bio-inks directly into a patient's body during a surgical procedure.29 This concept envisions the use of handheld or robot-assisted bioprinters in the operating room. A surgeon could use such a device to precisely fill a complex bone or tissue defect in real-time, fabricating a perfectly matched, customized implant on the spot.43 This approach would offer transformative advantages. It would eliminate the need for pre-operative imaging and the time-consuming, costly off-site manufacturing of patient-specific implants.29 The ability to make real-time adjustments during the printing process would ensure a perfect fit, and the direct application of the material would enhance its adhesion and integration with the surrounding native tissue.29 This would represent the ultimate form of personalized, on-demand manufacturing, collapsing a complex, multi-day supply chain into a single intraoperative step and potentially democratizing access to custom medical devices. Such technology could be invaluable in trauma surgery, military medicine, and remote or resource-limited healthcare settings.45

6.2 Challenges for Clinical Translation

Despite its immense promise, the journey of 3D printable adhesives from the research laboratory to the clinic is fraught with challenges that must be systematically addressed. Scalability and Manufacturing: Transitioning the synthesis of the complex bio-ink from small, laboratory-scale batches to large-scale, consistent production under Good Manufacturing Practice (GMP) standards is a major hurdle. Ensuring batch-to-batch reproducibility of the polymer's molecular weight, degree of functionalization, and rheological properties is critical for safety and efficacy. Sterilization: The bio-ink contains reactive chemical groups (NHS esters) that are essential for its adhesive function. Standard sterilization methods, such as gamma irradiation or high heat, could potentially degrade these groups or alter the polymer's structure, compromising its performance. Developing and validating a sterilization process

(e.g., ethylene oxide or sterile filtration) that maintains the ink's integrity is a critical and non-trivial step. Regulatory Pathways: As a novel combination product (a medical device made from a new material), 3D printable adhesives will face a rigorous regulatory approval process from bodies like the FDA. This will require extensive preclinical data on long-term safety, biocompatibility, degradation kinetics, and the fate of degradation byproducts. Comprehensive, long-term animal studies and, eventually, well-designed human clinical trials will be necessary to establish its safety and efficacy. Long-Term Performance: While short-term in vivo studies of up to 4 weeks have shown promising results 4, the long-term behavior of the adhesive within the body is not yet fully understood. Studies extending over many months or years are needed to characterize the complete degradation profile and to assess the chronic host tissue response to the implant and its byproducts.

6.3 Broader Societal and Economic Impact

Should these challenges be overcome, the widespread adoption of 3D printable tissue adhesives and related technologies could have a far-reaching impact on society and the economics of healthcare. Personalized Medicine and Improved Outcomes: This technology is a quintessential example of personalized medicine, enabling the creation of treatments and devices that are tailored to the unique anatomy and physiology of each individual patient.13 By ensuring a better fit and matching the mechanical properties of the host tissue, these customized devices promise to improve surgical outcomes, reduce complication rates (e.g., leaks, infections, device migration), shorten patient recovery times, and enable new surgical repairs that are currently infeasible.18 Healthcare Economics: The economic implications are significant. The market for 3D printed medical devices is already substantial and projected to grow to over 5.89 billion USD by 2030.47 By facilitating minimally invasive procedures, reducing time in the operating room, and lowering the risk of costly complications and revision surgeries, this technology has the potential to generate considerable cost savings for the healthcare system.18 On-demand, point-of-care manufacturing could also reduce inventory costs and logistical overheads associated with stocking a vast range of standard implant sizes. A Catalyst for Data-Driven Surgery: The integration of real-time imaging, computational modeling, and robotic in situ printing points toward a future of data-driven, precision surgery. One can envision a workflow where a surgeon scans a defect, software uses finite element analysis to automatically design an architecturally optimized and mechanically tuned scaffold 21, and a robotic arm prints the multi-material construct with superhuman precision. The 3D printable adhesive serves as the key effector material in this system, transforming surgery from a practice based primarily on manual dexterity to a high-technology engineering discipline.

Section 7: Conclusion

The development of 3D printable tissue adhesives marks a pivotal advancement in the field of biomedical materials and regenerative medicine. By systematically addressing the fundamental limitations of traditional sutures, staples, and conventional adhesives, this technology offers a versatile and powerful platform for tissue repair and bio-integration. The core innovation lies in the intelligent co-design of material chemistry and manufacturing methodology. A multi-component polymer network provides a multi-modal adhesion mechanism, combining rapid physical bonding for immediate sealing with robust covalent bonding for long-term stability, all while maintaining the bulk mechanical toughness required to function in dynamic physiological environments. Crucially, this advanced material is formulated into a viscoelastic ink with precisely engineered rheological propertiesnamely, shear-thinning behavior and a high yield stressthat make it compatible with high-resolution Direct-Ink-Write (DIW) 3D printing. This convergence of material and method is transformative. It liberates medical device design from the constraints of the "one-size-fits-all" paradigm, enabling the fabrication of patient-specific adhesive patches and implants that are customized to the unique anatomical geometry of a defect. Furthermore, it allows for the creation of architecturally defined "meta-materials" whose mechanical properties can be programmed to match those of the target tissue, solving the critical challenge of mechanical mismatch. Comprehensive preclinical validation in both ex vivo and in vivo models has provided compelling evidence of the technology's efficacy and safety. The 3D printed adhesives have demonstrated superior adhesion strength, the ability to form rapid, fluid-tight seals in challenging organ systems, and a remarkable capacity for achieving hemostasis on actively bleeding tissues. These

functional benefits are coupled with favorable biocompatibility and a controlled degradation profile, positioning the material not just as a sealant but as a temporary regenerative scaffold that facilitates the body's own healing processes. Beyond wound closure, the 3D printable adhesive serves as a foundational platform technology, solving the universal challenge of interfacing synthetic devices with soft biological tissues. This capability opens new frontiers for the development of integrated bioelectronics, localized drug delivery systems, and advanced tissue engineering constructs. While significant challenges related to manufacturing scale-up, sterilization, and regulatory approval must be overcome, the future trajectory points toward even more advanced applications such as real-time, intraoperative in situ bioprinting. Ultimately, 3D printable tissue adhesives represent a cornerstone technology for the future of personalized medicine, promising to enhance surgical outcomes, reduce healthcare costs, and catalyze a new era of data-driven, individualized patient care.

**A case study of a low-score report.**

## A Case Study of A Low-score Report

### DOI

10.1016/j.conbuildmat.2024.135602

### Domain

Building and Construction

### Performance

**Module 1:** Planning Acc 10.00%

**Module 2:** Retrieval Acc 0.76%

**Module 3:** Reasoning Acc 41.18%

### Task

Review

### Research Question

How can the development of ultra-high-performance concrete (UHPC) with lower carbon emissions be optimized by addressing its research background, challenges, methodologies, environmental impacts, and future directions?

### Gold Plan

**Subtask 1:** Summarize UHPCs properties, applications, and drivers for carbon reduction efforts.

**Subtask 2:** Define assessment framework using LCA boundaries and EC, EE, CI indicators.

**Subtask 3:** Identify environmental hotspots in UHPC mixes, highlighting cement and steel fibers.

**Subtask 4:** Compare binder reduction strategies, including SCMs and geopolymers, against strength requirements.

**Subtask 5:** Evaluate recycled aggregates and powders for environmental benefits and performance trade-offs.

**Subtask 6:** Assess replacing conventional steel fibers with waste fibers on emissions and mechanics.

**Subtask 7:** Integrate particle packing and statistical mixture design with LCA-driven optimization.

**Subtask 8:** Establish decision rules ranking strategies by EC, EE, and CI reductions.

**Subtask 9:** Discuss challenges: data standardization, durability integration, waste variability, and processing burdens.

**Subtask 10:** Propose future directions and implementation roadmap for scalable, durable, low-carbon UHPC.

**Predicted Plan**

**Sub-task 1:** Search specified academic databases for review articles to establish the research background of Ultra-High-Performance Concrete (UHPC), focusing on its fundamental properties, conventional composition, and the primary sources of its carbon footprint, particularly the high cement content.

**Sub-task 2:** Identify and synthesize information from research papers detailing the main technical challenges in reducing the carbon emissions of UHPC, such as maintaining mechanical performance, durability, and workability when substituting traditional cement.

**Sub-task 3:** Investigate and categorize the primary methodologies for developing low-carbon UHPC by searching for papers on: (a) The use of supplementary cementitious materials (SCMs) like fly ash, slag, silica fume, and calcined clays. (b) The formulation and performance of alternative binder systems, including geopolymers and alkali-activated materials. (c) The application of computational models and machine learning for mix design optimization.

**Sub-task 4:** Find and analyze Life Cycle Assessment (LCA) studies from peer-reviewed journals that compare the environmental impacts, especially the global warming potential, of conventional UHPC versus various low-carbon UHPC formulations.

**Sub-task 5:** Explore emerging trends and future research directions by finding recent academic papers on innovations such as the integration of nanomaterials, bio-inspired binders, and carbon capture, utilization, and storage (CCUS) technologies within the UHPC matrix.

**Sub-task 6:** Synthesize the gathered findings into a structured report covering the requested sections: research background, challenges, methodologies, environmental impacts, and future directions.

**Sub-task 7:** For every factual claim and data point presented in the final synthesis, ensure a direct citation to the source academic paper is included, providing its DOI or arXiv ID.

**Predicted Evidence**

1. 10.56748/ejse.23426
2. 10.1007/s11356-021-13435-2
3. 10.3390/buildings14020382
4. . . .

**Evidence**

1. 10.1016/j.conbuildmat.2024.135602
2. 10.1007/s11356-021-13435-2
3. 10.1007/s11367-013-0614-0
4. . . .

**Diagnostics**

**Statement 1:** UHPC is characterized by ultra-high compressive strength, high toughness, and excellent durability.
**Answer:** True

**Statement 2:** Due to optimized material usage, UHPC has a lower carbon footprint than conventional concrete (NC).
**Answer:** False

**Statement 3:** UHPC is widely used in marine environments due to its low permeability and high durability.

**Answer:** True

**Statement 4:** The main challenge in developing low-carbon UHPC is to reduce carbon emissions without compromising compressive strength.

**Answer:** True

**Statement 5:** Replacing steel fibers with waste fibers is the most effective way to reduce the carbon footprint of UHPC.

**Answer:** False

**Statement 6:** The use of waste aggregates greatly reduces the environmental impact of UHPC.

**Answer:** False

**Statement 7:** Life Cycle Assessment (LCA) can be used to evaluate the environmental impacts of UHPC.

**Answer:** True

**Statement 8:** The LCA analysis in the paper considers both material-level and structural-level impacts.

**Answer:** False

**Statement 9:** The CI value is calculated as the total embodied $CO_2$ emissions divided by the compressive strength of the UHPC mixture.

**Answer:** True

**Statement 10:** Geopolymer-based UHPC (UHPGC) has the lowest environmental impact among all UHPC types.

**Answer:** True

**Statement 11:** Cement replacement with supplementary cementitious materials (SCMs) is less effective than aggregate replacement in reducing carbon emissions.

**Answer:** False

**Statement 12:** Utilizing waste fibers, such as waste steel fiber (WSF), can significantly reduce the energy consumption of UHPC.

**Answer:** True

**Statement 13:** Future research should focus on incorporating durability properties into LCA of UHPC.

**Answer:** True

**Statement 14:** UHPC developed using limestone calcined clay cement (LC3) achieves both low carbon emissions and high compressive strength.

**Answer:** False

**Statement 15:** The combined use of carbon emissions, energy consumption, and compressive strength is sufficient to assess the environmental impact of UHPC.

**Answer:** False

**Statement 16:** The availability of waste fibers is a limiting factor for their large-scale application in UHPC.

**Answer:** True

**Statement 17:** The use of nanomaterials, such as graphene oxide, is a cost-effective way to enhance the mechanical properties of UHPC.

**Answer:** False

### Report

Optimizing the Next Generation of Ultra-High-Performance Concrete: Pathways to Decarbonization and Enhanced Performance

1.0 The Dichotomy of Ultra-High-Performance Concrete: Unparalleled Performance and Environmental Burden

Ultra-High-Performance Concrete (UHPC) represents a significant leap in cementitious materials technology, offering a combination of strength, ductility, and durability that far surpasses conventional concrete.1 It is not merely an incremental improvement but a material class engineered from fundamental principles to achieve a near-perfect microstructure.3 This exceptional performance, however, is achieved through a composition that carries a substantial environmental burden, primarily a high carbon footprint. This creates a central dichotomy: a material that enables more sustainable, long-lasting infrastructure is, on a per-volume basis, one of the most carbon-intensive forms of concrete. This report provides an exhaustive analysis of the strategies being developed to resolve this conflict, exploring pathways to optimize UHPC for lower carbon emissions without compromising its signature performance. It synthesizes the academic literature on the material's background, its environmental impact, the technical challenges of decarbonization, and the methodologiesfrom established practices to frontier technologiesthat are paving the way for a new generation of sustainable, ultra-high-performance construction materials.

1.1 Defining UHPC: Compositional Philosophy and Material Properties

The development of UHPC is guided by a distinct compositional philosophy aimed at maximizing particle packing density to create an extremely dense and homogenous microstructure with minimal porosity.3 This is not achieved by simply adding more cement, but through a systematic, multi-scale optimization of the granular mixture. Four core tenets underpin this design philosophy: 1. Extremely Low Water-to-Binder Ratio (w/b): UHPC formulations employ a very low w/b ratio, typically ranging from 0.14 to 0.25.2 This is a fundamental departure from conventional concrete and is the primary mechanism for minimizing the volume of capillary pores, which are the main source of weakness and permeability in hardened cement paste.8 The reduction in water ensures that a dense, robust structure is formed through hydration.2 2. Optimized Particle Packing Density: The principle of dense particle packing is central to UHPC design.2 The mix is engineered with a carefully graded distribution of fine materials, including Portland cement, fine aggregates like quartz sand, and very fine powders such as silica fume.2 The goal is for smaller particles to progressively fill the voids between larger particles, minimizing the interstitial space that would otherwise be filled with water, thereby reducing the overall water demand of the mixture.2 This dense matrix is the source of the material's exceptional durability.3 3. High Dosage of Superplasticizers: The extremely low w/b ratio would render the concrete unworkably stiff. To counteract this, a high dosage of high-range water reducers (superplasticizers) is essential.2 These chemical admixtures adsorb onto the surface of cement particles, imparting a negative charge that causes them to repel each other, thus fluidifying the mixture and ensuring adequate workability for placement and consolidation.2 4. Inclusion of Discontinuous Fibers: To overcome the inherent brittleness of high-strength cementitious matrices, UHPC incorporates a high volume of discontinuous fibers, most commonly high-strength steel fibers.2 These fibers bridge microcracks as they form, providing significant post-cracking tensile strength, ductility, and toughness.6 This transforms the material's failure mode from a sudden, brittle fracture to a gradual, ductile one, a critical property for structural applications.6 This meticulously engineered composition yields a suite of mechanical and durability properties that are an order of magnitude better than those of conventional concrete. Mechanically, UHPC is defined by compressive strengths that routinely exceed 120 MPa and can surpass 150 MPa.2 Critically, it also exhibits a sustained post-cracking tensile strength, typically in the range of 5 to 8 MPa or higher, a direct result of the fiber reinforcement.2 This combination of high compressive and tensile strength, along with enhanced ductility and energy absorption capacity, makes UHPC an ideal material for structures subjected to extreme loads, such as seismic events, blasts, or impacts.1 Perhaps the most significant benefit of UHPC is its exceptional durability.3 The highly dense, low-porosity microstructure provides a formidable barrier to the ingress of aggressive agents. This results in superior resistance to moisture permeability, chloride ion penetration, chemical attacks, and freeze-thaw degradation.1 For example, the chloride ion penetration value for UHPC can be as low as 22 coulombs, compared to over 1700 coulombs for normal strength concrete.8 This remarkable durability translates directly into a longer

service life for structures, reducing the need for costly maintenance and repairs over their lifespan.2

1.2 The Carbon Footprint Imperative: Quantifying the Environmental Impact of Conventional UHPC

The superior performance of UHPC is inextricably linked to its carbon-intensive composition. The primary driver of its environmental footprint is the extremely high content of Ordinary Portland Cement (OPC), which can range from 800 to 1300 kg/mş.7 This is often double the amount used in normal concrete.14 Given that cement production is responsible for 5-8% of global anthropogenic CO2 emissions, with approximately 0.8 tons of CO2 released for every ton of Portland cement produced, the high cement dosage in UHPC makes it a material with a very high embodied carbon content.9 When compared on a direct, cradle-to-gate (covering raw material extraction, transport, and production, known as life cycle stages A1-A3) and per-cubic-meter basis, UHPC consistently demonstrates a significantly higher carbon footprint than conventional concrete (CC).15 For both UHPC and CC, these A1-A3 stages are dominant, accounting for over 90% of the total carbon footprint of the material itself.15 However, the sources of these emissions differ in their distribution between the two material types. For conventional concrete, cement production is overwhelmingly the largest contributor, responsible for over 80% of its greenhouse gas (GHG) emissions.15 In UHPC, the emissions are more distributed among its high-value components. While the high cement content remains a major factor, the production of the high-strength steel fibers required for ductility can be responsible for over 40% of the total carbon footprint of the UHPC mixture.15 Additional contributions come from the production of other energy-intensive components, such as finely ground quartz sand and chemical admixtures like superplasticizers.5 This high per-volume carbon footprint creates a critical paradox when the material's functional benefits are considered. The exceptional strength and stiffness of UHPC allow for the design of structures that are significantly more slender and lightweight than their conventional concrete counterparts.1 This "dematerialization" can lead to a significant reduction in the total volume of concrete required for a given structural function, such as a bridge girder or a building column.19 For instance, a well-designed reinforced UHPC (R/UHPC) beam can reduce the required cross-section size by 50% and steel bar usage by 3053% compared to a conventional concrete beam with the same flexural load capacity.19 This reduction in material volume, coupled with the extended service life and reduced maintenance needs stemming from UHPC's superior durability, means that a structure built with a high embodied carbon material can ultimately have a lower overall life-cycle carbon footprint.15 This functional paradox is central to understanding the sustainability case for UHPC and underscores the importance of life cycle assessment (LCA) over simplistic per-volume comparisons. The very design philosophy that grants UHPC its unparalleled performancethe maximization of particle packing density to create a virtually flawless microstructureis the direct cause of its primary environmental liability. This dense packing has historically relied on filling every available void with fine, energy-intensive powders, chief among them being Portland cement, used in quantities far exceeding what is necessary for chemical hydration alone, with the unhydrated particles acting as expensive, high-carbon filler.7 This establishes a fundamental tension: the engineering mechanism for success is also the source of the environmental problem. Consequently, the challenge of creating low-carbon UHPC is not a simple exercise in material substitution. It is a fundamental re-engineering of the entire granular and binder system to achieve the same microstructural density and performance with a palette of materials that carries a much lower carbon burden. This reframes the task from merely "replacing cement" to holistically "re-designing the particle system for sustainability."

2.0 Foundational Strategies for Decarbonization: The Role of Supplementary Cementitious Materials (SCMs)

The most established and widely implemented strategy for reducing the carbon footprint of concrete, including UHPC, is the partial replacement of Portland cement with supplementary cementitious materials (SCMs).21 These materials are often industrial byproducts or processed natural materials that exhibit pozzolanic or latent hydraulic properties, allowing them to contribute to the strength and durability of the concrete. By substituting a portion of the energy-intensive cement clinker, SCMs directly lower the embodied carbon of the binder system.

2.1 Leveraging Industrial Byproducts: Fly Ash, Ground Granulated Blast-Furnace Slag (GGBFS), and Silica Fume

Three SCMs have historically dominated the field of high-performance concrete due to their availability and proven performance: Silica Fume (SF): While technically a byproduct of silicon and ferrosilicon alloy production, silica fume is a critical, high-value component in nearly all conventional UHPC formulations.6 It consists of extremely fine (sub-micron), amorphous, spherical particles of silicon dioxide. Its role in UHPC is twofold. Physically, its minute particles are highly effective at filling the microscopic voids between cement grains, dramatically improving particle packing density.6 Chemically, it is a highly reactive pozzolan that reacts with the calcium hydroxide (CH), a byproduct of cement hydration, to form additional calcium-silicate-hydrate (C-S-H) gelthe primary binding phase in concrete.6 This pozzolanic reaction refines the pore structure and strengthens the interfacial transition zone (ITZ) between aggregates and paste, leading to significant gains in strength and durability.8 Typical dosages in UHPC range from 20 Fly Ash (FA): A byproduct of coal-fired power plants, fly ash is composed of fine, spherical glassy particles. It is widely used as a cement replacement to improve the workability and long-term properties of concrete.22 The spherical shape of FA particles creates a "ball-bearing" effect, reducing inter-particle friction and improving the flowability of the fresh mix.24 Chemically, it is a pozzolan that reacts more slowly than silica fume, contributing to strength gain over longer periods (beyond 28 days) and enhancing ultimate strength and durability.24 However, this slow reactivity can be a disadvantage in applications requiring high early strength, such as in the precast industry.24 It is often used in binary (OPC-FA) or more complex ternary blends with other SCMs to balance performance.22 Ground Granulated Blast-Furnace Slag (GGBFS): A byproduct of the iron manufacturing process, GGBFS is a latent hydraulic material. This means that, unlike pozzolans, it does not require calcium hydroxide to react but can hydrate directly in the presence of water when activated by the alkaline environment created by Portland cement hydration.24 It contributes significantly to long-term strength and produces a highly refined pore structure, leading to excellent durability and resistance to chemical attack.25 GGBFS can also improve the workability of concrete.24 Similar to fly ash, its reaction can be slower than that of cement, potentially reducing early-age strength. To counteract this, it is frequently used in ternary systems, often combined with silica fume, which provides the needed early-age strength boost.24

2.2 Emerging Pozzolans: Metakaolin and Other Alternatives

As the global supply of traditional, high-quality SCMs like fly ash becomes increasingly constrained, the search for viable alternatives has intensified. Metakaolin (MK): Metakaolin is a highly reactive pozzolanic material produced through the controlled thermal activation (calcination) of kaolinitic clay. It is not a byproduct but a manufactured SCM. Its high reactivity can lead to rapid strength development, making it an effective partial replacement for cement or a high-performance alternative to silica fume.23 Studies have shown that its inclusion in concrete can significantly enhance strength, reduce permeability, and improve durability, including mitigating the deleterious alkali-silica reaction (ASR).2 Other Alternatives: The uncertainty surrounding the future availability of fly ash, driven by the global shift away from coal-based power generation, has created an urgent need to identify and qualify new sources of SCMs.23 Research is actively exploring a range of materials, including natural pozzolans like pumicite (volcanic ash) and manufactured materials like calcined clays.23 Experimental work has demonstrated that materials such as pumicite and metakaolin can successfully replace a majority, or even all, of the fly ash content in UHPC mixtures while still producing acceptable mechanical and durability properties.23 Other agro-industrial wastes with pozzolanic potential, such as Rice Husk Ash (RHA), are also being investigated as sustainable components for UHPC.7

2.3 Performance Trade-offs: Balancing Strength, Durability, and Workability

Incorporating high volumes of SCMs into UHPC is not a straightforward substitution; it is a complex balancing act that requires careful consideration of the trade-offs between various properties.14 Each SCM imparts a unique signature on the fresh and hardened characteristics of the concrete. Workability and Rheology: The fresh properties of the mix are highly sensitive to the type and amount of SCM used. The spherical morphology of fly ash and the properties of GGBFS generally improve flowability.24 In contrast, the extreme fineness and

high surface area of silica fume increase inter-particle forces and water demand, which can significantly reduce workability and require higher dosages of superplasticizer to achieve the target flow.24 Strength Development Profile: The rate of strength gain is a critical parameter, particularly for commercial applications. Silica fume contributes significantly to both early-age (1-7 days) and long-term strength due to its high reactivity.25 Fly ash and GGBFS, reacting more slowly, primarily contribute to strength gain at later ages (28 days and beyond).24 This can be a drawback for applications that require rapid formwork removal or early loading. To manage this, designers often resort to ternary blends, such as OPC-GGBFS-SF, which combine the benefits of each material to achieve a more balanced strength development curve.22 Durability: The use of SCMs is almost universally beneficial for the long-term durability of UHPC. By consuming the more soluble calcium hydroxide and producing additional, denser C-S-H, SCMs refine the capillary pore structure of the cement paste.10 This makes the concrete matrix less permeable and more resistant to the ingress of chlorides, sulfates, and other aggressive chemicals, thereby enhancing the service life of the structure.25 The strategy of decarbonizing UHPC through SCMs is facing a significant and growing challenge: the vulnerability of its supply chain. The primary method for reducing UHPC's carbon footprint relies on replacing cement with industrial byproducts like fly ash and GGBFS.21 However, the research explicitly acknowledges that the future availability of high-quality fly ash is in jeopardy due to the global phasing out of coal-fired power plants.23 Similarly, GGBFS is a finite byproduct of the steel industry. This reality implies that a decarbonization strategy that depends solely on these traditional waste streams is not sustainable in the long term. This situation creates a pressing need to accelerate research and development into a new generation of SCMs. As suggested by studies exploring materials like calcined clays and pumicite 23, the future lies in "manufactured" SCMs that can be produced at scale from abundant raw materials (like clays), with consistent quality and a low carbon footprint. This represents a major shift from passively using industrial waste to actively creating a new, engineered, and sustainable raw material stream for the global concrete industry, a transition that will require significant investment in resource geology, processing technology, and international standardization.

3.0 Paradigm Shift in Binder Technology: Geopolymers and Alkali-Activated Systems

While SCMs offer a pathway to reduce the carbon footprint of UHPC, a more radical approach involves the complete elimination of Portland cement. This paradigm shift is being led by the development of alternative binder systems based on alkali-activation chemistry. These materials, broadly known as alkali-activated materials (AAMs) or geopolymers, offer the potential for dramatic carbon reductions by creating high-performance binders without the need for the energy- and emissions-intensive process of cement clinker production. The application of this technology to create Ultra-High-Performance Geopolymer Concrete (UHPGC) represents a frontier in sustainable construction materials.

3.1 Principles of Alkali-Activation for Cement-Free UHPC

The formation of a geopolymer binder is fundamentally different from the hydration of Portland cement. It is a chemical process of polycondensation involving aluminosilicate materials in a highly alkaline environment. Geopolymer Chemistry: Geopolymers are inorganic, amorphous to semi-crystalline aluminosilicate polymers.30 They are formed through the reaction of a solid aluminosilicate source, known as a precursor, with a concentrated alkaline activator solution.30 The process begins with the dissolution of the precursor in the alkaline solution, releasing silicate and aluminate monomers into the liquid phase. These monomers then undergo a process of polycondensation to form a rigid, three-dimensional network of Si-O-Al bonds, which constitutes the hardened binder.32 The critical advantage of this process is that it occurs at or near ambient temperatures, completely avoiding the high-temperature (around 1450řC) calcination of limestone required for cement production, which is the primary source of process-related $CO_2$ emissions in the cement industry.9

Precursor Materials: A wide range of natural and industrial materials can serve as precursors for geopolymerization, provided they are rich in reactive alumina and silica. The most commonly used precursors are industrial byproducts, which enhances the sustainability profile of the technology. These include Ground Granulated Blast-Furnace Slag (GGBFS), fly ash (particularly Class F), and metakaolin.30 The chemical and mineralogical composition of the precursor, especially the molar ratio of silicon to aluminum ( $SiO_2/Al_2O_3$), is a critical

parameter that governs the reaction kinetics and the properties of the resulting geopolymer binder.32 Alkaline Activators: The activation of the precursor is achieved using a highly alkaline solution. These activators are typically a combination of an alkali hydroxide, such as sodium hydroxide (NaOH) or potassium hydroxide (KOH), and an alkali silicate, such as sodium silicate ($Na_2SiO_3$) or potassium silicate (often referred to as waterglass).9 The choice of activator, its concentration, and the ratio of silicate to hydroxide are crucial variables that significantly influence the workability of the fresh mix, the rate of strength development, and the final microstructure of the hardened UHPGC.34

3.2 Performance Characteristics of Ultra-High-Performance Geopolymer Concrete (UHPGC) Research into UHPGC has demonstrated its potential to match or even exceed the performance of conventional, cement-based UHPC. Mechanical Properties: By applying the same principles of optimized particle packing and low liquid-to-binder ratios, researchers have successfully developed UHPGC formulations with exceptional mechanical properties. Compressive strengths well over 100 MPa have been reported, with some studies achieving strengths approaching 123 MPa.33 As with conventional UHPC, the inclusion of reinforcing fibers (such as steel or PVA fibers) is essential to impart ductility, toughness, and strain-hardening behavior in tension and flexure, transforming the material into a viable structural composite.35 Durability and Microstructure: The hardened binder in UHPGC consists of a dense, amorphous aluminosilicate network, which can result in very low porosity and permeability.32 This microstructure can provide excellent durability, including superior resistance to certain forms of chemical attack, such as acids and sulfates, when compared to traditional OPC-based concrete.31 Carbon Reduction Potential: The primary driver for the development of UHPGC is its transformative potential for carbon reduction. By completely eliminating Portland cement from the formulation, the significant process emissions associated with clinker production are avoided. Life cycle assessment studies have quantified this benefit, indicating that the cradle-to-gate carbon emissions of UHPGC can be 32

3.3 A Holistic Environmental Assessment: Beyond Carbon to Energy and Water Footprints While the reduction in direct $CO_2$ emissions is substantial, a comprehensive environmental assessment of UHPGC reveals a more nuanced picture involving significant trade-offs in other impact categories. The Activator Trade-off: The sustainability of UHPGC is heavily influenced by the environmental footprint of its chemical activators. The industrial production of alkali silicates (waterglass) and hydroxides is an energy-intensive chemical process that also consumes significant amounts of water.9 This means that while the carbon burden is removed from the cement plant, a new environmental burden is created at the chemical manufacturing plant. LCA Comparative Analysis: A detailed cradle-to-gate Life Cycle Assessment (LCA) comparing alkali-activated material (AAM) based UHPC with Portland cement (PC) based UHPC illustrates this trade-off clearly. The analysis shows that while AAM-UHPC can achieve a 32-45% lower climate footprint (in kg CO2-eq.) and a 19-33% lower material footprint, it can simultaneously exhibit a 44-83% higher energy footprint and a 75-146% higher water footprint.9 The vast majority of these increased energy and water impacts are directly attributable to the production of the alkaline activators, particularly waterglass.9 Contingent Sustainability: This analysis reveals that the overall environmental superiority of UHPGC is not absolute but is contingent upon external factors. The net environmental benefit is highly dependent on the carbon intensity of the electrical grid that powers the chemical plants producing the activators.9 In a region with a predominantly fossil-fuel-based grid, the high energy demand for activators could offset a significant portion of the carbon savings from eliminating cement. Conversely, in a region with abundant renewable energy, the case for UHPGC becomes much stronger. This makes regional context and the decarbonization of the broader industrial ecosystem critical factors in the sustainable deployment of geopolymer technology. The promise of geopolymers as a mainstream solution for low-carbon UHPC extends beyond materials science into the realms of industrial ecology and chemical engineering. The high energy and water footprint associated with the production of conventional alkaline activators creates a scenario of "problem-shifting," where the environmental burden is effectively transferred from the cement kiln (process $CO_2$ emissions) to the chemical plant (energy consumption for activator synthesis). This means that a simple switch to a geopolymer binder does not automatically guarantee a superior overall environmental outcome. The true sustainability of the technology is contingent

on factors external to the concrete itself. A deeper analysis suggests that the strategic deployment of UHPGC should be prioritized in industrial ecosystems where sources of aluminosilicate precursors (like steel plants producing slag or power plants producing fly ash) are co-located with sources of low-carbon energy that can be used to produce the activators more sustainably. This points to a future where integrated industrial parks, designed around principles of circular economy, become the hubs for producing next-generation construction materials. Furthermore, this challenge highlights a critical frontier for chemical research: the development of novel, low-impact activation methods, such as solid, "one-part" activators that are pre-blended with the precursor and only require the addition of water, thereby simplifying logistics and potentially reducing the overall environmental footprint.30 Table 1: Comparative Properties of UHPC Binder Systems

Property Conventional OPC-UHPC SCM-Modified UHPC (>50Typical Binder Composition Portland Cement (OPC), Silica Fume (SF) 10 OPC, High-volume GGBFS, Fly Ash (FA), and/or Metakaolin (MK) 23 FA, GGBFS, MK (precursors); Alkali Activators (e.g., Sodium Silicate, NaOH) 30 Compressive Strength Range (MPa) 150 - 200+ 3 120 - 180 (highly dependent on SCM type and curing) 27 100 - 165 33 Key Durability Indicator Very Low Chloride Permeability (<100 Coulombs) 8 Very Low to Extremely Low (often improved over OPC-UHPC) 23 Excellent (often superior acid/sulfate resistance) 31 Relative Workability/Admixture Demand High superplasticizer demand due to SF and low w/b ratio 2 Variable; FA/GGBFS can improve flow, but overall demand remains high 24 Challenging; high alkalinity can affect superplasticizer performance, requiring specialized admixtures 14 Relative Cradle-to-Gate CO2-eq/mş High (Baseline) 14 Moderate to Low (up to 40-60% reduction vs. baseline) 18 Low to Very Low (up to 60% reduction vs. baseline, but with higher energy/water footprint) 9

4.0 Overcoming Technical Hurdles in Low-Carbon UHPC Formulation

The transition from conventional, high-cement UHPC to low-carbon alternatives is fraught with technical challenges that extend beyond simple material substitution. Achieving the required levels of performance, ensuring economic viability, and enabling widespread adoption requires overcoming significant hurdles in material science, engineering, and logistics. These challenges are often interconnected, creating a complex optimization problem for researchers and practitioners.

4.1 The Challenge of Maintaining Rheological Control and Workability

The fresh-state properties, or rheology, of UHPC are critical for its successful application. The material must be fluid enough to be placed in complex formwork and to properly encapsulate the dense network of reinforcing fibers, yet cohesive enough to prevent segregation. The introduction of high volumes of SCMs or entirely new alkali-activated binder systems can profoundly alter this delicate balance. Many alternative precursors and SCMs have particle shapes and surface textures that are different from Portland cement. For example, materials with more angular particles or a higher specific surface area can increase inter-particle friction and the demand for water, leading to a reduction in flowability and workability.14 This often necessitates an increase in the dosage of expensive superplasticizers to maintain the target slump flow. Furthermore, the chemical environment of geopolymer systems presents a unique challenge. The highly alkaline nature of the activator solutions can be incompatible with conventional polycarboxylate-based superplasticizers, reducing their effectiveness or causing rapid slump loss. This has spurred research into the development of new generations of chemical admixtures specifically designed to function effectively in these high-pH, noncementitious environments.

4.2 Ensuring Long-Term Durability and Mechanical Integrity

While many low-carbon UHPC formulations demonstrate promising mechanical properties at standard testing ages (e.g., 28 days), ensuring their long-term performance and durability is paramount for structural applications where service lives of 100 years or more are expected. This involves a deep understanding of their long-term behavior under sustained loads and environmental exposure. A key concern for all UHPC types is volumetric stability, particularly autogenous shrinkage. This is the internal volume reduction that occurs due to the chemical reactions of the binder. It is particularly pronounced in systems with very low water-to-binder ratios and highly reactive binders, leading to internal stresses that can cause microcracking if not properly managed.2 While conventional UHPC is known to have high

autogenous shrinkage, the behavior of novel binder systems must be thoroughly characterized to prevent premature deterioration.7 Similarly, long-term creepthe gradual deformation of the material under sustained loadmust be quantified for these new formulations to ensure the long-term serviceability of structures. For geopolymers and other alkali-activated systems, the mechanisms of aging and interaction with the environment are fundamentally different from those in Portland cement-based systems. While they may offer superior resistance to certain chemical attacks, their long-term performance with respect to other phenomena, such as carbonation (reaction with atmospheric CO2) and chloride-induced reinforcement corrosion, requires extensive and prolonged investigation to build the same level of confidence that exists for conventional concrete.14

4.3 Economic Viability and Scalability of Alternative Binders

For any new technology to be adopted by the construction industry, it must be both economically viable and scalable. UHPC already faces a significant barrier in its high initial material cost compared to normal concrete.2 Developing low-carbon versions must not exacerbate this issue. Cost: While replacing a large portion of expensive Portland cement with what are often considered lower-cost industrial byproducts seems economically advantageous, the overall cost equation is more complex. High-performance SCMs like silica fume and metakaolin can be as expensive as, or more expensive than, cement.23 For geopolymer systems, the cost of the chemical activators, particularly sodium or potassium silicate, can be a major contributor to the final price, potentially offsetting the savings from eliminating cement.30 The need for higher dosages of superplasticizers or special curing regimes can also add to the total cost. Scalability and Standardization: The widespread, commercial production of low-carbon UHPC depends on access to a reliable and consistent supply of alternative raw materials. The inherent variability in the chemical and physical properties of industrial byproducts, which can differ based on their source and processing, presents a significant challenge for quality control and consistent performance.14 Overcoming this requires robust quality assurance protocols and mix designs that are resilient to minor variations in raw materials. Furthermore, the construction industry relies heavily on established standards and design codes, which provide engineers with the confidence to specify materials. The lack of such standards for UHPC with very high SCM contents or for UHPGC is a major impediment to their broader acceptance and use in critical structural applications.37 The technical hurdles in formulating low-carbon UHPC are not isolated issues but are deeply interconnected, forming what can be described as a "trilemma" between performance, cost, and sustainability. An attempt to optimize one parameter can often have unintended negative consequences on the others. For example, a primary goal is to replace expensive, high-carbon cement with a cheaper, low-carbon SCM.14 However, this SCM may have lower reactivity, leading to unacceptably slow strength development for many applications.24 A potential solution is to apply thermal curing to accelerate the chemical reactions and achieve the required early strength.14 This, however, introduces a new problem: heat curing consumes energy, which adds to the operational cost and generates CO2 emissions, thereby partially negating the initial environmental benefit of using the SCM.14 Concurrently, the new SCM might also negatively affect the workability of the fresh mix.14 This could be addressed by increasing the dosage of superplasticizer, but high-performance superplasticizers are themselves expensive and have an embodied carbon footprint, again adding to the cost and environmental impact.38 This cascade of interconnected challenges demonstrates that a successful low-carbon UHPC formulation cannot be achieved through a simple, linear substitution approach. It requires a holistic, systems-level optimization that simultaneously considers the complex chemical and physical interactions between all components, the influence of curing conditions, and their associated economic and environmental costs. The sheer complexity of this multi-parameter problem is precisely why computational modeling and data-driven optimization have become indispensable tools for advancing the field.

5.0 Computational and Data-Driven Optimization of UHPC Mix Designs

Navigating the complex, multi-variable landscape of low-carbon UHPC formulation requires tools that can go beyond the traditional, iterative, trial-and-error approach of laboratory experiments. The intricate and non-linear relationships between mix composition, processing, and final performance make it an ideal domain for the application of advanced computational and data-driven methods. These techniques, particularly machine learning (ML), are transforming

the process of materials development from an empirical art to a predictive science, enabling the rapid and rational design of optimized, eco-efficient UHPC.

5.1 Predictive Modeling of Material Properties with Machine Learning

The foundational step in computational mix design is the development of accurate models that can predict the performance of a given UHPC formulation without the need for physical testing. The Need for Predictive Models: The traditional method of developing new concrete mixes involves casting and testing numerous trial batches, a process that is time-consuming, labor-intensive, and expensive.40 Furthermore, the performance of UHPC is sensitive to small changes in the proportions of its many components, making it difficult to intuitively understand the complex, multi-dimensional relationships at play.42 Machine Learning Approaches: To overcome these limitations, researchers are increasingly leveraging machine learning algorithms to build robust predictive models. By training on large, curated datasets of experimental results from published literature, these models can learn the underlying patterns connecting mix proportions to performance outcomes.38 A variety of algorithms have been successfully employed, including Artificial Neural Networks (ANNs), which excel at modeling complex non-linear relationships; ensemble methods like Extreme Gradient Boosting (XGBoost), which combine multiple "weak" learners to create a powerful predictive model; and Gaussian Process Regression (GPR), a probabilistic approach that can also provide a measure of uncertainty in its predictions.42 The inputs to these models are typically the dosages of the various mix components (e.g., cement, fly ash, silica fume, water, superplasticizer, fibers), while the outputs are key performance indicators such as 28-day compressive strength and slump flow.38 High Predictive Accuracy: These data-driven models have demonstrated remarkable accuracy in predicting UHPC properties. Studies consistently report high coefficients of determination ($R^2$ values of 0.90 or higher) and prediction errors of less than 10% when compared against unseen experimental data.42 This high level of accuracy provides a powerful tool for virtual screening and rapid prototyping, allowing researchers to evaluate thousands of potential mix designs computationally before committing to costly and time-consuming laboratory work.

5.2 Multi-Objective Optimization for Low-Carbon, High-Performance Formulations

The true transformative power of these computational methods is realized when the predictive ML models are integrated into a formal optimization framework. This allows for the automated search for novel mix designs that satisfy multiple, often competing, objectives. Optimization Framework: The process involves coupling the trained ML prediction model with a metaheuristic optimization algorithm, such as a Genetic Algorithm (GA) or Particle Swarm Optimization (PSO).40 The ML model acts as a "fitness function," rapidly evaluating the performance of candidate mix designs proposed by the optimization algorithm, which then intelligently guides the search towards better solutions in the vast design space. Defining Objectives and Constraints: This framework is exceptionally well-suited for the multi-objective problem of designing sustainable UHPC. The optimization can be formulated to simultaneously achieve several goals, such as minimizing the embodied carbon emissions (calculated from the mix proportions), minimizing the material cost, and maximizing key performance metrics like compressive strength and workability (slump flow).39 This search is conducted within a set of realistic constraints that define a feasible solution. These constraints can include required minimum performance levels (e.g., compressive strength > 120 MPa), practical ranges for the dosage of each component, established rules of concrete mix design (e.g., limits on the water-to-binder ratio), and the physical requirement that the absolute volume of all components sums to one cubic meter.42 Proven Success and Impact: This integrated ML-optimization approach has been successfully applied to design eco-efficient UHPC. In one notable study, an ANN model for strength and flow was combined with a GA to minimize carbon emissions. The optimization process identified a novel mix proportion that successfully reduced the calculated carbon emissions to 688 kg/m³, a significant improvement, while still satisfying the performance constraints of achieving over 120 MPa in compressive strength and 600 mm in slump flow. Crucially, this computationally derived mix was then produced and tested in the laboratory, and the experimental results validated the model's predictions with high accuracy.42 Such results demonstrate that this data-driven methodology is not just a theoretical exercise but a practical and powerful tool for accelerating the development of verifiably sustainable, high-performance materials.39 The adoption

of machine learning in UHPC development signifies a fundamental evolution in materials science, marking a transition from a traditional, empirical, materials-testing paradigm to a modern, computational, materials-design paradigm. The conventional research process, characterized by a slow, linear path of formulating, mixing, and physically testing individual batches, is being replaced by a rapid, parallel, and comprehensive exploration of a vast design space. The first-order benefit of this shift is a dramatic increase in efficiency, saving immense time, resources, and cost.42 The second-order benefit is the ability to perform true multi-objective optimization. By coupling predictive models with search algorithms, the system can identify not just a single "good" mix, but a whole frontier of "Pareto-optimal" solutionsa suite of designs where no single objective (like lowering CO2) can be improved without compromising another (like reducing strength). This provides engineers with a range of optimal trade-off options to choose from based on specific project requirements. The third-order implication of this technological shift is the potential to democratize advanced material design. Once robust, validated computational models are developed and made accessible, smaller research labs or companies lacking extensive physical testing facilities could leverage these powerful tools to design bespoke, high-performance, low-carbon materials for their specific applications, fostering innovation across the industry. Table 3: Overview of Machine Learning Applications in UHPC Mix Design

Study Citation Predictive Model Used Optimization Algorithm Input Variables Optimization Objectives Key Outcome / Validated Reduction 42 Artificial Neural Network (ANN) Genetic Algorithm (GA) Cement, FA, GGBS, SF, Fine Aggregate, Steel Fiber, Superplasticizer, Water Minimize Carbon Emissions; Maximize Compressive Strength & Slump Flow Carbon emissions reduced to $688 kg/m^3$ while maintaining compressive strength > 120 MPa and slump flow > 600 mm. Experimentally validated. 38 Gaussian Process Regression (GPR), ANN, Ensemble Techniques Genetic Algorithm (GA) Cement, Water, Superplasticizer, FA, GGBFS, Coarse & Fine Aggregates Minimize Carbon Footprint; Maximize Compressive Strength GPR model showed excellent accuracy (R2=0.90). GA identified mix configurations with high mechanical performance and reduced carbon emissions. 44 XGBoost Particle Swarm Optimization (PSO) Cement, Admixtures, etc. Minimize Cost; Minimize Carbon Emissions Model achieved high accuracy (R2=0.9043). PSO results showed potential for 7-10% carbon emission reduction and 1-3% cost reduction. 39 Machine Learning (various models compared) Analytic Hierarchy Process (AHP) for ranking Cement, SF, FA, Slag, Aggregates, Water, Steel Fiber, Superplasticizer Balance Compressive Strength, Flexural Strength, Fluidity, Shrinkage, Cost, and Carbon Emissions Developed a framework to derive optimal mix proportions oriented towards either low-cost or low-carbon emissions based on a comprehensive performance index. 40 XGBoost Regressor NSGA-II (a type of GA) Various concrete components Maximize Compressive Strength; Minimize Cost; Minimize CO2 Emissions Model achieved 98.5% accuracy. Optimization identified feasible high-strength concrete mixes (70-110 MPa) with optimized cost and emissions.

6.0 The Frontier of Sustainable UHPC: Advanced Materials and Carbon Sequestration

Beyond the optimization of existing material systems, the next frontier in the development of sustainable UHPC lies in the integration of advanced materials and novel technologies that can fundamentally alter the performance-to-impact ratio. These innovations include the use of nanomaterials to enhance efficiency at the molecular level, the application of carbon capture technologies to turn the concrete into a carbon sink, and the valorization of new, unconventional material streams.

6.1 Nanomaterial Integration for Enhanced Performance and Reduced Binder Content

Nanotechnology offers a powerful, bottom-up approach to improving the properties of cementitious materials. By engineering the material at the nano-scale, it is possible to achieve significant enhancements in performance with very small additions of nanomaterials. Mechanism of Action: Nanomaterials such as nano-silica (NS), nano-calcium carbonate (NC), nano-alumina, and carbon nanotubes (CNTs) influence the properties of UHPC through several mechanisms, owing to their extremely high surface area-to-volume ratio and unique chemical reactivity.47 They can act as nucleation sites, accelerating the precipitation of hydration products and leading to a more refined and homogenous microstructure.49 Their minute size allows them to fill the nano-scale pores within the C-S-H gel structure that are inaccessible to larger particles like silica fume, further densifying the matrix.48 Certain

nanomaterials, like CNTs and graphene, can also provide reinforcement at the molecular level, bridging nano-cracks and strengthening the C-S-H gel itself.48 Performance Enhancements: The incorporation of small dosages of these materials has been shown to yield significant improvements in both mechanical properties and durability. Studies have reported increases in compressive and flexural strength, enhanced bonding at the interfacial transition zone, and improved resistance to degradation mechanisms.47 For example, the synergistic use of carbon microfibers and carbon nanotubes in a hybrid reinforcement strategy has been shown to increase tensile strength by over 300% compared to a reference mixture.48 CNTs, in particular, can form cross-linked networks within the binder matrix that effectively arrest the propagation of microcracks at their inception.48 Potential for Dematerialization and Carbon Reduction: The primary sustainability benefit of nanomaterials in this context is their potential to make the cementitious binder more efficient. By significantly boosting the performance of the binder system, they may allow engineers to achieve the target UHPC performance levels (e.g., 150 MPa compressive strength) with a lower total volume of binder.52 This reduction in the required amount of cement and SCMs would lead to an indirect but significant reduction in the material's overall carbon footprint. Table 2: Impact of Nanomaterials on UHPC Properties Nanomaterial Primary Mechanism of Action Typical Dosage (% of binder) Observed Effect on Compressive Strength Observed Effect on Flexural/Tensile Strength Key Challenges Nano-Silica (NS) High pozzolanic activity; pore filling; nucleation site for C-S-H 49 1 - 3% Significant increase, especially at early ages 48 Moderate increase 47 Cost; potential for agglomeration; increased water demand 49 Nano-Calcium Carbonate (NC) Nucleation site for C-S-H, accelerating hydration 49 1 - 5% Increase, particularly at early ages 49 Minor increase 52 Lower cost than NS, but less pozzolanic effect 49 Carbon Nanotubes (CNTs) Nano-reinforcement (crack bridging); nucleation site; network formation 48 0.05 - 0.5% Moderate to significant increase 47 Significant increase due to crack bridging 48 Dispersion is extremely difficult; high cost 47 Nano-Alumina (Al2O3) Accelerates C3S hydration; pore filling 50 1 - 3% Increase in early and late strength 52 Moderate increase 50 Cost; influence on rheology Nano-Titanium Dioxide (TiO2) Fills pores; photocatalytic effect (self-cleaning) 47 1 - 5% Moderate increase 47 Minor increase 47 High cost; primarily used for functional properties Graphene / Graphene Oxide (GO) Nano-reinforcement; nucleation; barrier properties 50 0.02 - 0.1% Significant increase 52 Significant increase 52 Very high cost; dispersion challenges; potential interaction with admixtures 14

6.2 Innovations in Carbon Capture, Utilization, and Storage (CCUS) within the UHPC Matrix A truly transformative approach to decarbonization involves technologies that not only reduce emissions but actively capture and sequester CO2. Concrete, with its high content of calcium-bearing compounds, is an ideal medium for permanently storing CO2 through mineral carbonation. Concept of Mineral Carbonation: Carbon Capture, Utilization, and Storage (CCUS) in concrete is a process where captured CO2 gas is reacted with calcium (and magnesium) ions present in the cementitious materials to form thermodynamically stable carbonate minerals, such as calcium carbonate (CaCO3).53 This process effectively locks the CO2 into the solid matrix of the concrete, providing a durable and permanent form of sequestration.36 Pressurized CO2 Curing: One of the most promising methods for implementing this in UHPC is through accelerated carbonation curing. In this process, the freshly cast UHPC elements are placed in a sealed chamber which is then filled with pressurized, high-concentration CO2 gas.36 The pressure and concentration gradients drive the CO2 into the pore structure of the fresh concrete, where it dissolves and reacts with calcium-bearing phases like calcium hydroxide and unhydrated cement particles to form carbonates.36 Effectiveness and Performance Trade-offs: This technique has been shown to be highly effective at sequestering significant quantities of CO2. Studies have demonstrated carbon uptakes as high as 80 kg of CO2 per cubic meter of UHPC, particularly when the technology is combined with mixes that already have a reduced cement content (e.g., using 30% GGBFS as a cement replacement).36 This effectively turns the UHPC component into a carbon sink. However, the process is not without its challenges. The formation of carbonates can alter the microstructure and the progress of normal hydration reactions, which can lead to a slight decrease in the ultimate compressive strength compared to conventionally cured samples.36 Therefore, careful optimization of the mix design, curing pressure, and duration

is required to maximize carbon uptake while minimizing any negative impact on mechanical performance.

6.3 Novel Material Streams: From Recycled Tailings to Bio-Inspired Binders

Innovation in sustainable UHPC also involves looking beyond conventional construction materials and byproducts to valorize large-volume waste streams and draw inspiration from nature. Industrial Waste Valorization: A significant area of research focuses on the utilization of high-volume industrial wastes that are currently underutilized or sent to landfills. A prime example is the use of mine tailings, such as those from gold mining operations, as a comprehensive replacement for conventional materials in UHPC.56 Because these tailings often have a wide particle size distribution, from fine powders to sandy particles, they can be used in a "full scale recycling" approach to replace both the fine aggregates (like quartz sand) and a portion of the cementitious powders simultaneously.56 This simplifies processing by eliminating the need for sieving and maximizes the recycling rate. Research has shown that this approach can produce UHPC with high compressive strength, excellent durability, and a significantly reduced carbon footprint, with one study reporting a 25% decrease in global warming potential (GWP).56 Bio-Inspired Design: While still in its early stages for UHPC, the field of bio-inspired materials design offers intriguing future pathways. This involves learning from the efficient and high-performance structures found in nature. For example, researchers are mimicking the "brick-and-mortar" layered structure of nacre (mother-of-pearl) to create composite materials with exceptional fracture toughness and resilience.57 In the context of reinforced concrete, this could involve creating bio-inspired coatings or impregnations for reinforcing fibers to tailor the bond behavior and improve the overall composite performance.57 Other research is exploring the use of bio-based materials, such as wood aggregates, to create lightweight "bio-concretes".58 While not yet achieving the performance of UHPC, these concepts point towards a future where high performance is achieved not just through material strength, but through intelligent, hierarchical structural design inspired by biological systems. The convergence of these frontier technologies is fundamentally shifting the long-term goal for sustainable UHPC. The objective is moving beyond simple "carbon reduction" and towards the credible possibility of "carbon negativity." A synergistic, multi-pronged strategy can create a clear pathway to this goal. The first step is to drastically lower the baseline embodied emissions of the binder by replacing the majority of Portland cement with low-carbon SCMs or by adopting a cement-free geopolymer system, which can cut the initial carbon footprint by more than half.9 The second step is to apply a CCU technology, such as pressurized $CO_2$ curing, to this already low-carbon mix. This actively removes and permanently sequesters additional $CO_2$ from the atmosphere, with demonstrated uptakes of up to 80 kg per cubic meter.36 The third step involves integrating nanomaterials to enhance the chemical efficiency of the now-reduced binder content, potentially allowing for a further reduction in the total amount of binder required to achieve the target performance.48 The combined effect of these layered strategies can result in a final product with a net carbon footprint that is drastically lower than what could be achieved by any single approach alone, and potentially negative. This could transform high-performance infrastructure projects like bridges, wind turbine towers, and coastal defenses from being major sources of carbon emissions into durable, long-term carbon repositories, fundamentally altering the role of the construction industry in climate change mitigation.

7.0 A Life Cycle Perspective: Comparative Assessment of Low-Carbon UHPC Systems

To accurately evaluate the environmental credentials of any construction material, it is essential to move beyond single metrics and adopt a holistic life cycle perspective. A Life Cycle Assessment (LCA) provides a comprehensive framework for quantifying the environmental impacts of a product or system from raw material extraction through to end-of-life. For a high-performance material like UHPC, where the initial impacts are high but the in-service benefits are substantial, this life cycle approach is not just beneficialit is essential for a fair and meaningful comparison.

7.1 Cradle-to-Gate Analysis of Alternative Formulations

The most common starting point for an LCA is a "cradle-to-gate" analysis, which assesses the environmental impacts associated with the production of the material itself. This includes the extraction and processing of raw materials (A1), their transportation to the manufacturing

facility (A2), and the manufacturing process itself (A3).15 This type of analysis is crucial for comparing the embodied carbon of different UHPC formulations. As established, conventional OPC-based UHPC has a very high cradle-to-gate carbon footprint due to its high content of cement and steel fibers.14 However, alternative formulations can offer dramatic reductions. For example, replacing a high-clinker cement (CEM I) with a blended cement containing a high volume of GGBFS (such as CEM III, with up to 70% slag) can reduce the $CO_2$-equivalent emissions of the binder phase by up to 60%, and the emissions of the full UHPC mix (including fibers) by as much as 40%.18 Similarly, cement-free, AAM-based UHPC can achieve comparable or even greater reductions in carbon footprint, although this often comes with a trade-off in the form of higher impacts in other environmental categories, such as non-renewable energy use and water consumption, due to the production of chemical activators.9 This highlights the necessity of a multi-criteria assessment even within the cradle-to-gate boundary.

7.2 The Functional Unit Advantage: Service Life Extension and Reduced Material Volume While cradle-to-gate analysis is useful for comparing materials on a like-for-like mass or volume basis, it presents a fundamentally flawed and misleading picture for advanced materials like UHPC, whose primary value lies in their superior performance.20 A more accurate and meaningful assessment must be based on a "functional unit," which defines the function the material is intended to perform. For a structural material, this could be a bridge beam designed to carry a specific load over a given span for a service life of 100 years.15 When viewed through this lens, the environmental case for UHPC becomes much more compelling. Material Reduction and Dematerialization: The ultra-high strength of UHPC allows structural engineers to design elements with significantly smaller cross-sections compared to those made with conventional concrete to achieve the same load-bearing capacity.1 This directly translates to a reduction in the total volume of concrete needed for the structure. Furthermore, the enhanced mechanical properties can lead to a reduction in the amount of traditional steel reinforcement required. Studies have shown that a well-designed UHPC beam can achieve the same performance as a conventional beam with up to 50% less cross-sectional area and 30-53% less reinforcing steel.19 This "dematerialization" is a powerful lever for reducing the overall environmental impact of a construction project. Extended Service Life and Reduced Maintenance: Perhaps the most significant long-term benefit of UHPC is its exceptional durability. Structures built with UHPC are far more resistant to deterioration from environmental factors like chloride ingress and freeze-thaw cycles.1 This translates into a significantly longer service life with substantially lower requirements for maintenance, repair, and eventual replacement compared to structures built with conventional concrete.2 When these use-phase benefits are incorporated into a full "cradle-to-grave" LCA, the initial high embodied carbon of the UHPC can be more than offset by the avoidance of impacts associated with future repair and reconstruction activities. A comprehensive analysis of a bridge, for example, can show that the UHPC design provides a net reduction of 14% in carbon footprint, 27% in material footprint, and 43% in water footprint over its entire life cycle compared to a conventional concrete design.15 A properly designed R/UHPC beam can even match the carbon footprint of a conventional beam at the component level while offering dramatic reductions in material volume.19 The "functional unit advantage" is the most robust and powerful argument for the adoption of UHPC as a key material in sustainable construction. However, the realization of this advantage in practice is not automatic; it depends on a concurrent evolution in structural design, engineering practice, and procurement policy. An LCA clearly shows that on a per-cubic-meter basis, UHPC has a high environmental impact.15 Yet, on a functional, whole-of-life basis for a structure like a bridge, it can be far superior due to material efficiency and longevity.17 This benefit only materializes if engineers and architects actively design structures differently to leverage the unique properties of the material. If UHPC is simply substituted into a design that was originally conceived for conventional concrete, its potential for dematerialization is lost, and the result is an over-designed structure with a needlessly high environmental impact. Furthermore, common procurement processes in both the public and private sectors often prioritize the lowest initial construction cost, which inherently penalizes a premium material like UHPC, even if its total life-cycle cost is significantly lower due to reduced maintenance.2 Therefore, the successful large-scale deployment of sustainable UHPC is as much a challenge of education, policy

reform, and cultural change as it is a materials science problem. Without updates to design codes that encourage more efficient designs, without the training of engineers to think beyond conventional structural forms, and without procurement policies that value and reward whole-life carbon and cost performance, the full sustainability potential of this remarkable material will remain largely unrealized.

8.0 Synthesis and Strategic Directions for Future Research and Development

The development of low-carbon Ultra-High-Performance Concrete represents a critical intersection of materials science, environmental engineering, and computational design. The research synthesized in this report demonstrates a clear and accelerating progression from incremental improvements to transformative innovations aimed at resolving the inherent conflict between UHPC's exceptional performance and its conventional carbon intensity. A multi-faceted strategy, leveraging a portfolio of technologies, is emerging as the most effective path forward.

8.1 Key Optimization Pathways for Low-Carbon UHPC

The evidence points to three primary pathways for the development and deployment of sustainable UHPC, each suited to a different timescale and technological maturity level: 1. Near-Term: Optimized Hybrid Binder Systems: The most pragmatic and immediately deployable solution involves the aggressive substitution of Portland cement with locally available Supplementary Cementitious Materials. The focus should be on optimizing ternary (OPC-SCM1-SCM2) or even more complex quaternary blended cements with total SCM replacement levels exceeding 50%. The key to success in this pathway is the use of computational modeling and multi-objective optimization to rapidly identify mix designs that balance the competing demands of mechanical performance, workability, durability, cost, and carbon footprint. This data-driven approach can de-risk the use of high SCM volumes and accelerate their adoption in commercial applications. 2. Medium-Term: Regionally-Adapted Geopolymer Systems: For the medium term, the complete elimination of Portland cement through alkali-activated or geopolymer binders (UHPGC) offers the greatest potential for deep decarbonization. However, to be truly sustainable, this pathway must address the high energy and water footprint of chemical activators. The strategic focus should be on developing UHPGC systems that are tailored to regional industrial ecosystems, valorizing locally abundant aluminosilicate precursors (such as specific types of industrial slag, fly ash, or calcined clays) and pairing their production with low-carbon energy sources for activator synthesis. Further research into lower-impact activators and "one-part" systems will be crucial for improving their economic and environmental viability. 3. Long-Term: Carbon-Negative Formulations: The long-term vision, representing a paradigm shift in material design, is the development of UHPC systems that function as net carbon sinks. This will be achieved through the synergistic combination of the most advanced technologies. The foundation will be an ultra-low-carbon binder system (either a high-SCM or geopolymer base). This will be enhanced with nanomaterials to maximize the efficiency of the binder, allowing for dematerialization. Finally, Carbon Capture and Utilization (CCUS) technologies, such as pressurized $CO_2$ curing, will be integrated into the production process to permanently sequester atmospheric $CO_2$ within the material's matrix. Achieving this vision will transform critical infrastructure from a source of emissions into a durable, long-term carbon repository.

8.2 Critical Research Gaps and Recommendations for Industry Adoption

To translate these scientific advancements into widespread industrial practice, several critical gaps must be addressed through targeted research and strategic action: Generation of Long-Term Durability Data: While short-term performance data is abundant, there is a pressing need for long-term (10+ years) field and laboratory data on the durability of low-carbon UHPC formulations, particularly for geopolymer systems. This data is essential for validating performance under real-world service conditions and building the confidence of asset owners and engineers. Development of Standards and Design Codes: The lack of internationally recognized standards and building codes for UHPC containing very high volumes of SCMs, and especially for UHPGC, is a major barrier to adoption. A concerted international effort is needed to develop performance-based standards that can accommodate these novel materials, providing a clear pathway for their specification and use in structural applications. Harmonization of Life Cycle Assessment (LCA) Methodologies: To enable fair and transparent comparisons between different material solutions, a standardized methodology

for conducting and reporting LCAs for UHPC is required. This should include clear guidelines on defining the functional unit, system boundaries, and allocation procedures for byproducts, ensuring that sustainability claims are robust and comparable. Investment in High-Profile Demonstration Projects: Overcoming the inherent conservatism of the construction industry requires tangible proof of concept. Funding and supporting high-profile demonstration projectssuch as bridges, slender high-rise elements, or coastal protection structuresbuilt with low-carbon UHPC would be invaluable. These projects serve not only to validate the technology at scale but also to showcase its aesthetic, economic, and environmental benefits to designers, contractors, and the public, thereby accelerating its acceptance and adoption.

### A.10  FURTHER RELATED WORK

**Reasoning-Guided Search Agents**  Early work on LLM-based search agents established the foundation for combining reasoning with tool use. ReAct (Yao et al., 2023) demonstrated how models can interleave reasoning traces with external search actions, while subsequent systems such as Search-o1 (Li et al., 2025a), Search-R1 (Jin et al., 2025a), RAG-Gym (Xiong et al., 2025), ReSearch(Wang et al., 2024), Collab-RAG (Xu et al., 2025), R1-Searcher (Song et al., 2025), and DynaSearcher (Hao et al., 2025) expanded this paradigm with reinforcement learning, collaborative retrieval, dynamic search policies, and evaluation frameworks. These methods are effective for multi-hop question answering, but they remain limited to relatively short contexts and cannot analyze or synthesize insights from large collections of research papers or long-form articles (Zhang et al., 2025b). To address more complex research-oriented scenarios, several commercial systems have recently emerged, including Gemini DR (Google, b), OpenAI DR (OpenAI, 2025), Perplexity DR (PerplexityAI, a), and Grok (xAI), each positioned as an end-to-end deep-research assistant. In practice, these systems follow a similar operational paradigm: they first decompose user queries into sub-goals, then invoke search engines or RAG pipelines to collect relevant evidence, and finally synthesize an answer with a textual explanation. Despite their utility, such systems remain largely opaque, and their evaluation typically reports only aggregate success without disentangling planning, retrieval, and reasoning capabilities.

**Deep Research Evaluation**  Several recent benchmarks have attempted to measure aspects of DR, but they remain partial in scope. GAIA (Mialon et al., 2023) focuses primarily on short-form answer generation, providing little insight into planning or evidence attribution. BrowseComp (Wei et al., 2025) and its multilingual variant BrowseComp-ZH (Zhou et al., 2025), together with Humanitys Last Exam (HLE) (Phan et al., 2025), emphasize the retrieval stage, evaluating how well models can locate information in web documents, but they do not capture the full cycle of task decomposition, retrieval, and reasoning. More recently, the Deep Research Benchmark (FutureSearch et al., 2025) has been introduced to provide a broader evaluation of research agents. While this benchmark expands the task set and evaluation dimensions compared to earlier efforts, it still operates largely at the end-to-end level and mainly tests search capability, leaving open the question of how to disentangle planning, retrieval, and reasoning performance in a transparent manner.

### A.11  LLM USAGE

We acknowledge the use of a large language model (LLM) to assist in the editing and polishing of this manuscript. The LLM was utilized to refine grammar, improve sentence structure, and ensure the clarity and readability of the text. All core ideas, analyses, and conclusions presented in this paper were conceived by the human authors, who retain full responsibility for the final content of this work.

## A.12 PROMPTS

> **Evaluation Prompt**
>
> You are a professional researcher preparing a structured, data-driven report on behalf of a global research team. Your task is to analyze the research question the user poses.
>
> Do:
> - Focus on data-rich insights: include specific figures, trends, statistics, and measurable outcomes.
> - When appropriate, summarize data in a way that could be turned into charts or tables, and call this out in the response.
> - Prioritize reliable, up-to-date sources: peer-reviewed research, academic institutions, regulatory agencies, or authoritative reports.
> - Include inline citations and return all source metadata.
>
> Be analytical, avoid generalities, and ensure that each section supports data-backed reasoning that could inform research or policy decisions.
>
> Question: {QUESTION}

## A.13 POTENTIAL PAPER SELECTION BIAS

We acknowledge that our citation filter (10 citations in 2 years) represents a deliberate trade-off, prioritizing signal-to-noise ratio over the inclusion of the "long tail." Given the explosive growth in publications, this criterion acts as a vital heuristic to ensure the benchmark consists of high-impact, peer-validated work. Our goal was to first establish robust agent performance on research that reflects clear methodologies and community consensus, rather than risk noise from unvetted papers. We agree this is a limitation and identify generalization to the "long tail" as a key direction for future work.

## A.14 TOOLKIT SETTING

For DR agent, we describe toolkit usage in this section. First, we use web_search_preview in OpenAI. Second, we do not use extra tools in Perplexity. Third, we use auto mode search parameters and set return_citations as True.

## A.15 METRICS DEFINITION

We define our evaluation metrics based on the comparison between the set of predicted items ($P$), the set of gold (ground-truth) items ($G$) and the universal set ($U$). We first define the standard components based on this comparison:

**Definition 6 (True Positives)** *True Positives ($TP$): The number of items correctly identified that are in the gold set.*

$$TP = |P \cap G|. \tag{4}$$

**Definition 7 (True Negatives)** *True Negatives ($TN$): The number of items correctly identified that are not in the gold set.*

$$TN = |(U \setminus G) \cap (U \setminus P)|. \tag{5}$$

**Definition 8 (False Positives)** *False Positives ($FP$): The number of items incorrectly identified that are not in the gold set.*

$$FP = |P \setminus G| = |P| - TP. \tag{6}$$

**Definition 9 (False Negatives)** *False Negatives ($FN$): The number of items in the gold set that are not identified.*

$$FN = |G \setminus P| = |G| - TP. \tag{7}$$

From these components, our reasoning metrics are formalized as follows:

$$\text{Accuracy} = \frac{TP + TN}{TP + TN + FP + FN}. \tag{8}$$

$$\text{Precision} = \frac{TP}{TP + FP} = \frac{|P \cap G|}{|P|}. \tag{9}$$

$$\text{Recall} = \frac{TP}{TP + FN} = \frac{|P \cap G|}{|G|}. \tag{10}$$

$$\text{F1-Score} = 2 \cdot \frac{\text{Precision} \cdot \text{Recall}}{\text{Precision} + \text{Recall}} = \frac{2 \cdot TP}{2 \cdot TP + FP + FN}. \tag{11}$$

In planning and retrieval module, for the metric we label Accuracy in this paper, we use the Jaccard Index (IoU), which is the standard metric for set similarity. We use this, and not standard Accuracy, because True Negatives ($TN$) are undefined for this retrieval task.

$$\text{Accuracy (Jaccard Index)} = \frac{TP}{TP + FP + FN} = \frac{|P \cap G|}{|P \cup G|}. \tag{12}$$

To interpret these metrics, we map them to intuitive concepts. First, Accuracy measures total judgment correctness of the model across all items, which penalizes both missing ($FN$) and extra steps ($FP$). Second, Recall measures Coverage, quantifying how many of the necessary ground-truth items ($G$) the model successfully identified (penalizing False Negatives, $FN$). Third, Precision measures Redundancy, which penalizes incorrect or redundant steps ($FP$). Finally, the F1-Score provides a balanced measure of overall quality, representing the mean of Coverage and Correctness.

## A.16 LLM-AS-A-JUDGE PROMPTS

We show the prompts used in LLM-as-a-Judge.

---

**Prompt for LLM-as-a-Judge**

Evaluate the quality of the given scientific report to the question.

Rate the response on three dimensions: accuracy, coverage, and redundancy. Use a scale from 1 (worst) to 10 (best).

- Accuracy: Assess how well the response addresses the question and provides correct information.
- Coverage: Evaluate whether the key points are covered in the response.
- Redundancy: Check for duplicate information.

Please be strict in your evaluation, ensuring that scores reflect subtle differences in quality. Aim to distribute scores more evenly across the scale, avoiding clusters at the high end. Consider the following criteria for scoring:

- A score of 1-3 indicates major flaws in multiple dimensions.
- A score of 4-6 indicates moderate issues or inconsistencies.
- A score of 7-8 reflects generally good quality with minor flaws.
- A score of 9-10 is reserved for exemplary responses that excel in all dimensions.

Question: {RESEARCH QUESTION}

Response: {REPORT}

---

Table 9: Temporal Filtering Analysis: Performance Comparison of Unrestricted vs. Time-Constrained Evaluation.

| Model | Research | | | | Review | | | | Avg. | | | |
|---|---|---|---|---|---|---|---|---|---|---|---|---|
| | Accuracy | Precision | Recall | F1 | Accuracy | Precision | Recall | F1 | Accuracy | Precision | Recall | F1 |
| *Retrieval* | | | | | | | | | | | | |
| Temporally Constrained | 50.00 | – | – | – | 0.53 | 0.73 | 11.21 | 1.05 | 25.27 | – | – | – |
| Unrestricted (Baseline) | 50.00 | – | – | – | 0.36 | 0.61 | 10.51 | 0.71 | 25.18 | – | – | – |
| *Reasoning* | | | | | | | | | | | | |
| Temporally Constrained | 58.59 | 58.83 | 37.66 | 44.82 | 60.22 | 61.54 | 41.02 | 48.50 | 59.41 | 60.18 | 39.34 | 46.66 |
| Unrestricted (Baseline) | 62.69 | 60.02 | 40.43 | 47.17 | 57.19 | 58.58 | 39.04 | 45.94 | 59.94 | 59.30 | 39.74 | 46.56 |

Table 10: Test-Time Compute Scaling (pass@1 vs. pass@3). Results are reported as percentages.

| Model | Research (pass@1 / pass@3) | Review (pass@1 / pass@3) | Avg. (pass@1 / pass@3) |
|---|---|---|---|
| *Planning* | | | |
| Gemini | 23.26 / 40.00 | 22.96 / 40.00 | 23.11 / 40.00 |
| *Retrieval* | | | |
| OpenAI | 50.00 / 50.00 | 10.51 / 11.26 | 30.26 / 30.63 |
| Gemini | 70.00 / 70.00 | 11.42 / 12.86 | 40.71 / 41.43 |
| Perplexity | 70.00 / 80.00 | 14.88 / 16.42 | 42.44 / 48.21 |
| Grok | 66.67 / 70.00 | 14.04 / 15.59 | 40.35 / 42.79 |
| *Reasoning* | | | |
| OpenAI | 62.69 / 70.55 | 57.19 / 66.42 | 59.94 / 68.49 |
| Gemini | 62.08 / 67.95 | 49.31 / 65.12 | 55.69 / 66.53 |
| Perplexity | 53.73 / 68.80 | 44.35 / 59.52 | 49.04 / 64.16 |
| Grok | 33.00 / 49.06 | 30.89 / 46.04 | 31.94 / 47.55 |
| Search-r1 | 22.42 / 37.46 | 36.81 / 42.65 | 29.62 / 40.06 |

## A.17 RETRIEVAL MATCHING METHODOLOGY

**Rationale for Title Prefix Matching.** Evaluating the accuracy of citations retrieved by deep research agents presents unique challenges due to the unstructured nature of model outputs. To ensure robust and fair evaluation, we adopt a specific 20-character title prefix matching strategy rather than relying on rigid DOI matching or semantic equivalence. We prioritize matching the first 20 characters of the normalized paper title for two primary reasons: First, infeasibility of automated DOI extraction: Deep research models often generate unstructured bibliographies containing truncated titles or URLs without structured metadata (e.g., DOIs). This makes automated, exact-ID matching practically infeasible and prone to high false-negative rates. Second, unreliability of semantic equivalence: Our preliminary experiments indicated that semantic embedding matching is unreliable for bibliographic verification. Distinct papers often share semantically similar titles, leading to false positives. A character-based prefix match provides a robust middle ground that handles title truncation while maintaining specificity.

**Handling Paper Versioning.** We address potential concerns regarding mismatched citations due to updated paper versions (e.g., preprints vs. final versions) by focusing on the final version-of-record. Domain Publication Norms: The prevalence of preprints is largely specific to Computer Science and occasionally Finance. In the vast majority of the specialized domains in our benchmark (e.g., Materials, Chemistry, Medicine, and Earth Science), the publication ecosystem relies solely on the final, formally published version. Minimal Title Variation: We analyzed the dataset for title discrepancies between versions and found a negligible fraction ($< 1\%$) of reference papers with significant title changes. This rate is statistically insignificant regarding the benchmark's validity, and papers with changed titles often represent distinct works with altered content.

## A.18 LLM-JUDGE RELIABILITY AND VALIDATION

To ensure the reliability of our evaluation, our framework is expressly designed to mitigate the risks of subjective LLM judgment by grounding the process in objective criteriaspecifically structured plans

Table 11: End-to-end performance across modules on research, review, and average tasks. Results represent the mean ± standard deviation over three independent evaluation runs. All values are reported as percentages.

| Model | Research Accuracy | Research Precision | Research Recall | Research F1 | Review Accuracy | Review Precision | Review Recall | Review F1 | Avg. Accuracy | Avg. Precision | Avg. Recall | Avg. F1 |
|---|---|---|---|---|---|---|---|---|---|---|---|---|
| **Planning** | | | | | | | | | | | | |
| Gemini | 26.08±0.02 | 26.08±0.02 | 25.43±0.02 | 25.39±0.02 | 29.75±0.02 | 29.75±0.02 | 31.02±0.01 | 29.74±0.01 | 27.91±0.02 | 27.91±0.02 | 28.23±0.02 | 27.56±0.02 |
| **Retrieval** | | | | | | | | | | | | |
| OpenAI | 62.00±0.00 | – | – | – | 0.69±0.00 | 1.73±0.00 | 5.27±0.00 | 1.36±0.00 | 31.35±0.00 | – | – | – |
| Gemini | 72.00±0.00 | – | – | – | 1.55±0.00 | 4.00±0.00 | 7.64±0.00 | 3.00±0.00 | 36.77±0.00 | – | – | – |
| Perleixity | 66.00±0.00 | – | – | – | **3.64**±0.00 | **22.85**±0.00 | 9.01±0.00 | **6.88**±0.00 | 34.82±0.00 | – | – | – |
| Grok | **76.00**±0.00 | – | – | – | 3.62±0.00 | 13.22±0.00 | **37.51**±0.00 | 6.84±0.00 | **39.81**±0.00 | – | – | – |
| **Reasoning** | | | | | | | | | | | | |
| OpenAI | 61.36±0.00 | **61.66**±0.00 | 41.54±0.00 | **48.68**±0.00 | 56.99±0.00 | **61.51**±0.00 | 38.91±0.00 | 46.75±0.00 | 59.18±0.00 | **61.59**±0.00 | 40.22±0.00 | 47.72±0.00 |
| Gemini | **61.69**±0.00 | 60.57±0.00 | **41.64**±0.00 | 48.38±0.00 | **58.16**±0.00 | 60.70±0.00 | **39.72**±0.00 | **47.27**±0.00 | **59.93**±0.00 | 60.63±0.00 | **40.68**±0.00 | **47.82**±0.00 |
| Perleixity | 44.55±0.00 | 59.92±0.01 | 29.99±0.00 | 38.99±0.00 | 44.66±0.01 | 60.18±0.01 | 30.44±0.00 | 39.25±0.00 | 44.60±0.00 | 60.05±0.01 | 30.22±0.00 | 39.12±0.00 |
| Grok | 32.55±0.04 | 54.28±0.03 | 22.07±0.03 | 30.46±0.03 | 30.29±0.04 | 58.55±0.03 | 21.19±0.03 | 30.12±0.04 | 31.42±0.04 | 56.41±0.03 | 21.63±0.03 | 30.29±0.03 |
| Search-r1 | 24.20±0.01 | 33.31±0.01 | 21.04±0.01 | 25.79±0.01 | 22.71±0.01 | 26.89±0.01 | 15.86±0.00 | 19.95±0.01 | 23.45±0.01 | 30.10±0.01 | 18.45±0.01 | 22.87±0.01 |

Table 12: Domain-Level Performance Characterization: Mean Accuracy and 95% Confidence Intervals (O3-Deep-Research).

| Domain | Mean Accuracy (%) | 95% CI Margin ($\pm$) |
|---|---|---|
| **High Performance** | | |
| Biology | 65.51 | 7.92 |
| Medicine | 64.51 | 7.73 |
| Environmental Science | 63.75 | 9.50 |
| Energy | 61.35 | 8.60 |
| Chemistry | 60.44 | 8.64 |
| **High Difficulty** | | |
| Building and Construction | 57.90 | 8.27 |
| Earth Sciences | 57.00 | 9.33 |
| Materials | 56.57 | 8.23 |
| Computer Science | 54.25 | 8.86 |
| Finance | 45.19 | 10.07 |

and diagnostic checklistsrather than relying on unstructured intuition. We validated the robustness of this standard by assessing the inter-model agreement between two distinct blind judges (GPT-4o and Gemini-2.5-pro), achieving a high consistency of 93.00% (SD=3.42%, see Appendix A.5). This demonstrates that the evaluation standard is stable and independent of the specific judge model. Furthermore, we validated the metrics alignment with human expertise; as detailed in Section 5.5, the ranking produced by our automatic Reasoning F1 score aligned with human expert preferences in 86.23% of comparisons. Finally, to ensure maximum reproducibility and minimize variance, we fixed the evaluation temperature at 0.0 and selected the most robust prompt formulation after experimental comparison (see Appendix A.12 for the exact prompt).

### A.19 DOMAIN ROBUSTNESS AND DIFFICULTY CHARACTERIZATION

To ensure statistical rigor and mitigate the risk of over-interpreting noise, we calculated 95% confidence intervals (CIs) for all ten domains. As shown in Table 12, the analysis confirms statistically significant performance gaps; for instance, stable, high-resource domains such as Biology achieve high accuracy with relatively narrow intervals ($65.51 \pm 7.92\%$), whereas volatile, challenging domains like Finance exhibit both lower mean accuracy and higher variance ($45.19 \pm 10.07\%$). This approximate 20% gap, characterized by non-overlapping CIs, validates that the benchmark captures genuine domain generalization challenges rather than statistical fluctuations. Beyond quantitative metrics, we characterize domain difficulty along two dimensions: knowledge prevalence and reasoning depth. We observe that model performance correlates with the availability of domain knowledge in pre-training corpora, with agents excelling in high-resource fields (e.g., Medicine) while degrading in specialized applied sciences. Furthermore, an analysis of plan structurewith an average length of 9.01 stepsreveals that domains necessitating longer, multi-step reasoning chains generally correspond to lower success rates, identifying reasoning depth as a critical determinant of task difficulty.

Table 13: End-to-end performance with gold plans across modules on research, review, and average tasks. All values are reported as percentages.

| Model | Research | | | | Review | | | | Avg. | | | |
|---|---|---|---|---|---|---|---|---|---|---|---|---|
| | Accuracy | Precision | Recall | F1 | Accuracy | Precision | Recall | F1 | Accuracy | Precision | Recall | F1 |
| **Retrieval** | | | | | | | | | | | | |
| OpenAI | 50.00 | – | – | – | 0.25 | 0.57 | 0.65 | 0.50 | 25.13 | – | – | – |
| Gemini | 70.00 | – | – | – | 0.98 | 2.49 | 11.79 | 1.93 | 35.49 | – | – | – |
| Perplexity | **80.00** | – | – | – | **4.02** | 13.67 | **15.60** | **7.64** | **42.01** | – | – | – |
| Grok | **80.00** | – | – | – | 3.90 | **27.03** | 14.38 | 7.29 | 41.95 | – | – | – |
| **Reasoning** | | | | | | | | | | | | |
| OpenAI | **68.68** | 59.52 | **45.32** | **50.16** | **73.18** | **62.76** | **49.77** | **54.59** | **70.93** | **61.14** | **47.54** | **52.38** |
| Gemini | 66.28 | 60.22 | 44.00 | 49.66 | 54.79 | 55.33 | 38.01 | 44.26 | 60.53 | 57.77 | 41.00 | 46.96 |
| Perleixity | 58.36 | **61.19** | 39.71 | 47.24 | 53.96 | 60.56 | 35.75 | 43.87 | 56.16 | 60.87 | 37.73 | 45.55 |
| Grok | 44.66 | 59.75 | 30.25 | 38.84 | 40.77 | 58.71 | 27.93 | 36.58 | 42.71 | 59.23 | 29.09 | 37.71 |
| Search-r1 | 31.36 | 42.43 | 26.44 | 32.58 | 28.70 | 35.13 | 20.52 | 25.91 | 30.03 | 38.78 | 23.48 | 29.24 |

## A.20 ANNOTATION STATISTICS AND DISTRIBUTION

To quantify the complexity and ensure the balance of our expert annotations, we analyzed the structural characteristics of the dataset. The expert-annotated gold plans exhibit an average length of 9.01 steps, reflecting the comprehensive and multi-stage nature of the deep research tasks required by the benchmark. Furthermore, we analyzed the distribution across all 3,400 diagnostic checkpoints to verify evaluation rigor. The dataset comprises 2,200 (64.71%) positive (True) instances and 1,200 (35.29%) negative (False) instances. This distribution confirms that the benchmark provides substantial and balanced coverage for both verifying correct reasoning capabilities and testing the agents' capacity to reject incorrect paths, thereby mitigating the risk of unbalanced evaluation.

## A.21 PLANNING AS A DOMINANT BOTTLENECK

To rigorously identify the limiting factors within the deep research pipeline, we conducted a "Gold Plan Injection" ablation study where ground-truth plans were provided to the agents while retaining model-generated retrieval and reasoning. The results, as shown in Table 13, demonstrate that fixing the planning module yields a significant performance boost in the downstream Reasoning phase. Specifically, OpenAI's average Reasoning Accuracy rose to 70.93%, representing a substantial improvement over the 58.65% observed in the standard end-to-end setting (see Table 2). This performance gain of nearly 12% confirms our hypothesis that planning is indeed a dominant bottleneck, as high-quality plans effectively guide the agent toward superior reasoning outcomes even when the retrieval component remains imperfect.

Furthermore, regarding the evaluation methodology for the planning module itself, we address the necessity of aligning model-generated plans with annotated steps. Unlike open-ended creative tasks, rigorous academic research follows a standardized, widely accepted paradigm (e.g., progressing from Background to Research Gap, Methodology, and Experiments). Our annotated plans are derived directly from the logical backbone of peer-reviewed source papers (see Appendix A.4 for examples). Therefore, requiring alignment to these standardized steps ensures that the agent covers the necessary scientific components rather than generating superficial decompositions. This alignment is essential for objectively evaluating the logical completeness and scientific validity of the research process.

