# OpenReview forum: "Dr.Mi-Bench: A Modular-integrated Benchmark for Scientific Deep Research Agent"
_ICLR.cc/2026/Conference — Submitted to ICLR 2026_

### Official Review · Reviewer_TLtz · 2025-10-27

**Soundness:** 2
**Presentation:** 2
**Contribution:** 3
**Rating:** 4
**Confidence:** 4

**Summary:**

The paper introduces Dr.Mi-Bench, a human-annotated, modular-integrated benchmark for evaluating scientific deep research agents. It formalizes the agent workflow into three modules—Planning, Retrieval, and Reasoning—and proposes Dr.Mi-Eval, a dual-mode evaluation paradigm that supports both end-to-end agent assessment and isolated module testing of foundational LLMs. The benchmark comprises 200 instances across 10 scientific domains (research and review papers), with ground-truth plans, gold evidence, and diagnostic statements derived from the source papers. The study evaluates commercial deep-research systems and leading LLMs, reports efficiency trade-offs, and validates automatic metrics against human preferences. Key findings include a fragmented performance landscape, severe retrieval challenges on review-style tasks, and high-level planning as a bottleneck for unlocking LLM reasoning potential.

**Strengths:**

- Original modular-integrated paradigm that disentangles planning, retrieval, and reasoning, enabling diagnostic and attributable evaluation.
- Grounding in academic paper structure (outlines, citations, and content-based diagnostics) provides objective, reproducible signals for each module.
- Dual evaluation modes (end-to-end agents vs. isolated LLMs) offer both realism and upper-bound capability analysis.
- Diverse, human-annotated dataset across 10 scientific domains and both research/review tasks, improving relevance beyond general-purpose benchmarks.
- Clear, actionable empirical insights (e.g., review-task retrieval collapse, planning bottleneck, agent specialization) and practical efficiency analysis (latency, token trade-offs).
- Effort to validate automatic metrics with human preferences shows strong alignment, lending credibility to the evaluation design.

**Weaknesses:**

- I believe the benchmark’s generality could be greatly improved by including a simple Deep Research Agent framework to evaluate LLMs in an end-to-end manner. At present, it only evaluates LLMs in a modular way, and the results are not directly comparable to existing end-to-end systems like o3-deep-research.
- In the Planning Module evaluation, explain how the pairwise judgment matrix is converted into precision, recall, and F1 scores. Also define coverage, structural correctness, and redundancy, and include the exact steps or formulas used to compute each.
- In Tables, please include the unit (%) in the header for greater rigor.
- It would be better to demonstrate the variance of the performance to demonstrate the robustness of the results.


> “We consistently observe the highest performance in fields like Computer Science, Medicine, and Biology, which is likely attributable to the extensive representation of these domains in the models’ training corpora. In contrast, agents struggle significantly more in specialized areas such as Earth Science, Materials, and Building and Construction, which are likely less prevalent in the training data.” in L367-L371

- I think the description does not align with the figure. In Figure 2, Computation Science (score=0.54) is lower than Earth Science (score=0.57), Building and Construction (score=0.58), and Mathematics (score=0.56).

**Questions:**

- On the Planning evaluation: Why must the evaluation require alignment to the annotated plan steps? There are many valid ways to plan and decompose a research question. How does your metric accommodate semantically equivalent but structurally different plans, and avoid penalizing correct alternative decompositions?
- On the Diagnostics for reasoning: Why must the evaluation rely on a fixed set of diagnostic statements as the target? A correct report may not cover every diagnostic detail while still adequately answering the question.
- It states “the detailed formalization and scoring for each component,” (L143) but I could not find the actual scoring details in the appendix—only high-level descriptions.

---

> ### Author Response · Authors · 2025-11-21
> **Response to Reviewer TLtz (1/2)**
>
> We deeply appreciate reviewer's constructive feedback, which has significantly helped us improve the quality of our work.
>
> # A1: Modular vs. End-to-End Evaluation of Backbone LLM (Weakness 1)
> We respectfully argue that modular evaluation is a deliberate design choice to **ensure a fair comparison of the LLMs themselves**. An end-to-end evaluation heavily conflates the LLM's performance with its agent framework (e.g., search, memory) that goes beyond our current scope. The scope of our paper is to isolate and evaluate the LLM's core potential as a backbone for such agents. Modular evaluation is the only way to achieve this.
>
> # A2: On Planning Evaluation and Metric Definitions (Weakness 2, Question 3)
> **The formal definitions for all metrics and conceptual mapping are added in Appendix A.15.** Our process for converting the pairwise comparison into scores is as follows:
> 1. **From Comparison to $TP/FP/FN$:** We perform a set-matching comparison by aligning the agent's predicted sub-tasks ($P$) against the gold-standard sub-tasks ($G$). This alignment directly yields the core counts: $TP$ (True Positives), $FP$ (False Positives) and $FN$ (False Negatives). Then, we convert these counts to Accuracy ($TP / (TP + FP + FN)$), Precision ($TP / (TP + FP)$), Recall ($TP / (TP + FN)$), and F1 score ($2 * TP / (2 * TP + FP + FN)$).
> 2. **Conceptual Mapping:**
>     - **Correctness is measured by Accuracy**, which penalizes both missing ($FN$) and extra steps ($FP$).
>     - **Redundancy is measured by Precision**, which penalizes incorrect or redundant steps ($FP$).
>     - **Coverage is measured by Recall**, which penalizes missed steps ($FN$).
>     - The F1 score serves as a balanced metric for overall quality.
>
> # A3: Table Formatting (%) (Weakness 3)
> We **add the (%) unit to the table captions in the updated version**, due to space limitations within the table columns.
>
> # A4: Variance (Weakness 4)
> We conduct **three independent evaluation runs** for all models and have added the results to the **Table 11 L2970-L2983** in our revised manuscript. This negligible variance confirms that our evaluation framework is highly **robust and reproducible**.
>
> | **Model**| |**Research**| |  | | **Review**| |   | | **Avg.**|  |  |
> |---|----|----|----|-------|----|--|-----|----|----|-----|----|----|
> || **Accuracy**|**Precision**|**Recall**|**F1**|**Accuracy**|**Precision**|**Recall**|**F1**|**Accuracy**|**Precision**|**Recall**|**F1**|
> |||||||``Planning``|||||||
> | **Gemini**    |26.08±0.02 |26.08±0.02 |25.43±0.02 |25.39±0.02 |29.75±0.02 |29.75±0.02 |31.02±0.01|29.74±0.01|27.92±0.02|27.92±0.02|28.23±0.02|27.57±0.02|
> |||||||``Retrieval``|||||||
> | **Openai**    |62.00±0.00|-|-|-|0.69±0.00|1.73±0.00|5.27±0.00|1.36±0.00|31.35±0.00|-|-|-|
> | **Gemini**    |72.00±0.00|-|-|-|1.55±0.00|4.00±0.00|7.64±0.00|3.00±0.00|36.77±0.00|-|-|-|
> | **Perplexity**|66.00±0.00|-|-|-|**3.64**±0.00|**22.85**±0.00|9.01±0.00|**6.88**±0.00|34.82±0.00|-|-|-|
> | **Grok**      |**76.00**±0.00|-|-|-|3.62±0.00|13.22±0.00|**37.51**±0.00|6.84±0.00|**39.81**±0.00|-|-|-|
> |||||||``Reasoning``|||||||
> | **Openai**    |61.36±0.00|**61.66**±0.00|41.54±0.00|**48.68**±0.00|56.99±0.00|**61.51**±0.00|38.91±0.00|46.75±0.00|59.18±0.00|**61.59**±0.00|40.22±0.00|47.72±0.00|
> | **Gemini**    |**61.69**±0.00|60.57±0.00|**41.64**±0.00|48.38±0.00|**58.16**±0.00|60.70±0.00|**39.72**±0.00|**47.27**±0.00|**59.93**±0.00|60.63±0.00|**40.68**±0.00|**47.82**±0.00|
> | **Perplexity**|44.55±0.00|59.92±0.01|29.99±0.00|38.99±0.00|44.66±0.01|60.18±0.01|30.44±0.00|39.25±0.00|44.60±0.00|60.05±0.01|30.22±0.00|39.12±0.00|
> | **Grok**      |32.55±0.04|54.28±0.03|22.07±0.03|30.46±0.03|30.29±0.04|58.55±0.03|21.19±0.03|30.12±0.04|31.42±0.04|56.41±0.03|21.63±0.03|30.29±0.03|
> | **Search-r1**|24.20±0.01|33.31±0.01|21.04±0.01|25.79±0.01|22.71±0.01|26.89±0.01|15.86±0.00|19.95±0.01|23.45±0.01|30.10±0.01|18.45±0.01|22.87±0.01|
>
>
> # A5: Discrepancy between Text and Figure 2 (Weakness 5)
> We have **revised this sentence in the manuscript L371-L374** to remove Computer Science from the high-performing list and correctly states that domains like Medicine and Biology show higher performance, which contrasts with the results in specialized areas.
> ```
> We consistently observe the highest performance in fields like Medicine and Biology, which is likely attributable to the extensive representation of these domains in the models' training corpora.
> ```

---

> ### Author Response · Authors · 2025-11-21
> **Response to Reviewer TLtz (2/2)**
>
> # A6: Planning Bottleneck and Evaluation Alignment (Question 1)
> 1.  **Ablation: Planning is a Dominant Bottleneck.** We have added the experiment to **Appendix A.21**.
>     - **Experimental Setting:** We conduct the **Gold Plan Injection** experiment as suggested, where ground-truth plans are provided to the agents.
>     - **Results:** As shown in the table below, providing ground-truth plans yields a significant boost. **OpenAI's average Reasoning Accuracy rises to 70.93%**, a substantial improvement over the **58.65%** in the end-to-end setting. This confirms that planning is a dominant bottleneck. High-quality plans effectively guide agents to better reasoning outcomes, even with imperfect retrieval.
> 2.  **On Structural Alignment and Alternative Decompositions:**
>     We require alignment because, unlike open-ended creative tasks, **academic research follows a standardized, rigorous paradigm** (e.g., Background $\to$ Research Gap $\to$ Methodology $\to$ Experiments).
>     - **Permutation-Invariant Matching:** We do not enforce rigid sequential alignment. Instead, as described in Section 3.1 (L182-L186), we employ a **semantic pairwise matching** strategy where **each predicted sub-task is individually compared against every gold sub-task**.
>     - **Eliminating Order Bias:** This "all-to-all" matching treats the plan as a **set of semantic steps** rather than a rigid sequence. This effectively **eliminates the influence of order**, allowing the metric to correctly identify and credit semantically equivalent but structurally different plans (e.g., valid reordering or alternative phrasing) without penalty.
>
> | **Model**  | |**Research**| |  | | **Review**| |   | | **Avg.**|  |  |
> |---|----|----|----|-------|----|--|-----|----|----|-----|----|----|
> || **Accuracy**|**Precision**|**Recall**|**F1**|**Accuracy**|**Precision**|**Recall**|**F1**|**Accuracy**|**Precision**|**Recall**|**F1**|
> |||||||`Retrieval`|||||||
> | **Openai**    |50.00|-|-|-|0.25|0.57|0.65|0.50|25.13|-|-|-|
> | **Gemini**    |70.00|-|-|-|0.98|2.49|11.79|1.93|35.49|-|-|-|
> | **Perplexity**|**80.00**|-|-|-|**4.02**|13.67|**15.60**|**7.64**|**42.01**|-|-|-|
> | **Grok**      |**80.00**|-|-|-|3.90|**27.03**|14.38|7.29|41.95|-|-|-|
> |||||||`Reasoning`|||||||
> | **Openai**    |**68.68**|59.52|**45.32**|**50.16**|**73.18**|**62.76**|**49.77**|**54.59**|**70.93**|**61.14**|**47.54**|**52.38**|
> | **Gemini**    |66.28|60.22|44.00|49.66|54.79|55.33|38.01|44.26|60.53|57.77|41.00|46.96|
> | **Perplexity**|58.36|**61.19**|39.71|47.24|53.96|60.56|35.75|43.87|56.16|60.87|37.73|45.55|
> | **Grok**      |44.66|59.75|30.25|38.84|40.77|58.71|27.93|36.58|42.71|59.23|29.09|37.71|
> | **Search-r1** |31.36|42.43|26.44|32.58|28.70|35.13|20.52|25.91|30.03|38.78|23.48|29.24|
>
> # A7: Diagnostics for Reasoning (Question 2)
> We choose fixed diagnostics because empirical evidence proves that **open-ended LLM-as-a-Judge evaluation is unreliable**.
> 1. **Empirical Validation (Objectivity vs. Subjectivity):** We experimentally compare our method against a standard open-ended LLM-as-a-Judge approach.
>    - **Poor Human Alignment:** The open-ended LLM judge achieves only **48.64%** alignment with human experts (SD=0.05), compared to **86.23%** (SD=0.03) for our diagnostic method.
>    - **Score Inflation:** It assigns an inflated **84.82%** Accuracy to OpenAI's system (vs. **58.65%** via our metric), confirming the lack of discrimination.
>    - **Lack of Discrimination:** Crucially, the open-ended judge lacks discriminative power (**SD=0.05**), whereas our method effectively differentiates task difficulty (**SD=0.19**).
> 2. **Design Motivation:** Our goal is to transform evaluation into an objective, verifiable framework.
>    - **Core Sufficiency:** Our diagnostics represent the essential information sufficiency (e.g., key methodologies, critical data points) derived from both author and peer-reviewed consensus. If a report misses these key evidence points, it is less complete and objectively lower quality.
>    - **Robust Aggregation:** By aggregating 15-20 data points per question, our metric provides a robust degree of completeness rather than a brittle binary judgment. A high-quality report will naturally cover the vast majority of these core points, ensuring the score reflects its true depth.
>
> We have added these experiments to **Section 5.5** and the specific prompts to **Appendix A.16**.

---

> ### Author Response · Authors · 2025-11-27
>
> Thank you very much for reviewing our responses. If you believe we have addressed the majority of your concerns, we kindly ask you to consider increasing the score for the paper. Should you have any additional questions, we would greatly appreciate it if you could provide further comments at your earliest convenience. We will respond promptly. Thank you!

---

### Official Review · Reviewer_caxP · 2025-10-31

**Soundness:** 3
**Presentation:** 2
**Contribution:** 3
**Rating:** 6
**Confidence:** 3

**Summary:**

The paper proposes Dr.Mi-Bench, a modular-integrated benchmark for evaluating “deep research (DR) agents” along three modules—Planning, Retrieval, and Reasoning—with two evaluation modes: end-to-end (agent) and isolated (foundational LLM as backbone). The benchmark consists of 200 human-annotated instances spanning 10 scientific domains and includes both research and review papers. The companion evaluation paradigm (Dr.Mi-Eval) scores planning via coverage/structure/redundancy against expert gold plans, retrieval via canonical-ID exact matches (DOI/arXiv/links), and reasoning via boolean diagnostics derived from each paper; it additionally reports efficiency trade-offs (latency vs. accuracy; token length vs. accuracy). Empirically, the authors report a fragmented performance landscape: different systems specialize in different modules (e.g., Grok strong in retrieval; Gemini in reasoning) while all struggle on multi-source retrieval for review-style tasks and on consistency across scientific fields; improving high-level planning is identified as a key lever for unlocking LLM reasoning when used as backbones.

**Strengths:**

1. Clear modularization of DR into Planning / Retrieval / Reasoning with dual modes (E2E vs. isolated) and a formal mapping from (Q,E,H,B) to grounded reports. This gives the community a shared lens to diagnose agents.

2. Paper-grounded supervision: gold plans from outlines, gold citations from bibliographies, and diagnostic T/F statements → objective, reproducible signals beyond BLEU-style report matching.

3. Actionable insights: (i) planning recall is a cross-model bottleneck; (ii) review-style retrieval remains unsolved; (iii) domain bias likely tracks training corpora composition.

4. Efficiency analysis that acknowledges practical constraints (latency, verbosity), with Pareto frontiers to visualize trade-offs.

5. Human alignment check for reasoning, offering some reassurance against grader artifacts.

**Weaknesses:**

1. Rigid retrieval gold for review papers.
Exact-ID matching to a paper’s bibliography may under-credit legitimate alternative or closely related sources (e.g., updated versions, foundational works not cited, thematic matches). A graded retrieval metric or semantic-equivalence layer would improve fairness and realism.

2. LLM-judge reliability not fully validated.
Planning and reasoning evaluation rely on LLM judgments. The paper references consistency checks, but lacks details on judge calibration, human–LLM agreement, or variance across different judges/prompts/temperatures.

3. Scale and representativeness limitations.
200 samples across 10 domains (~20 each) is a good start but small for broad claims about field robustness and domain generalization. Confidence intervals and domain-difficulty characterization are needed to avoid over-interpreting noise.

4. Potential selection bias from citation-based filtering.
Requiring ≥10 citations may bias toward already-visible work and reduce exposure to the “long tail” of scientific research where retrieval/generalization challenges may differ.

5. Limited annotation QA metrics reported.
The paper mentions guidelines but does not provide inter-annotator agreement (e.g., Cohen’s κ), distribution of plan lengths/diagnostics, or rejection/correction rates — making it hard to assess annotation consistency.

6. Risk of diagnostic-gaming in reasoning.
Boolean diagnostic checks encourage structured evaluation, but may lead models to optimize for checklist answers rather than holistic scientific synthesis. Complementary rubric or nugget-based human scoring on a subset would strengthen evaluation.

7. Reproducibility transparency could be improved.
Details on released artifacts (judge prompts, gold plans, bibliographies, toolkit settings) and handling of API variability would help ensure the community can reliably replicate results.

**Questions:**

1. Retrieval scoring:
Have you explored semantic citation matching (e.g., DOI families, preprint→journal mappings, thematic coverage) rather than strict exact-ID matching? How sensitive are results to this choice?

2. Judge reliability:
Could you report human–LLM agreement and/or LLM–LLM agreement for plan evaluation and diagnostic scoring, plus judge prompts, temperature, and calibration details?

3. Planning bottleneck ablation:
Can you show results where gold plans are injected while retrieval/reasoning are model-generated? This would validate planning as the dominant bottleneck.

4. Domain-level variance:
Please provide per-domain sample counts, confidence intervals, and difficulty statistics. Do cross-domain gaps persist after normalizing for citation density or terminology?

5. Dataset sensitivity:
How do conclusions change if the dataset size or review:research ratio is varied? Any signs of overfitting to this specific scale?

6. Efficiency evaluation details:
Was latency normalized for tool-use budgets? Did you measure gains after stripping boilerplate from model outputs or controlling for token verbosity?

7. Artifact release:
Which components will be released (gold plans, bibliographies, diagnostics, system + judge prompts, scoring scripts)? Will per-instance scoring breakdowns be included?

---

> ### Author Response · Authors · 2025-11-21
> **Response to Reviewer caxP (1/3)**
>
> We sincerely thank reviewer's insightful comments and the valuable time spent evaluating our manuscript.
>
> # A1: Exact-ID Matching and Updated Paper Versions (Weakness 1, Question 1)
> We agree that only using DOI matching is too rigid, while our method is more comprehensive (see Section 3.2).
> 1. **ID Matching Methodology**: We prioritize our **20-character title matching strategy** over strict DOI or semantic matching for two reasons:
>    - **Infeasibility of DOI Extraction**: Model retrieved citations are often unstructured (containing only truncated titles or URLs), making automated DOI extraction infeasible.
>    - **Unreliability of Semantic-Equivalence**: We find semantic matching unreliable, as distinct papers often share semantically similar titles.
> 2. **Limited Applicability of Updated Paper Versioning**: We only annotate final published versions, so the **influence of updated versions is negligible**.
>    - **Limited Applicability of Preprints**: Preprints are predominantly concentrated in Computer Science and Finance. For the vast majority of specialized domains in our benchmark, papers rely solely on the published version.
>    - **Minimal Rate of Title Changes**: We do find a negligible fraction (less than 1%) of reference papers that had changed titles, this rate is too low to impact validity, and these papers can often be considered distinct works.
>
> We have added detailed analysis to **Appendix A.17** in the manuscript.
>
> # A2: LLM-Judge Reliability (Weakness 2, Question 2)
> 1. **Judge calibration**: Our framework is expressly designed to **mitigate LLM judgement reliability risk by grounding the evaluation in objective criteria** (i.e., plans and diagnostics checklists), rather than relying on any single, subjective LLM judge. To validate this, we test GPT-4o and Gemini-2.5-pro as blind judges and achieve a **93.00%** inter-model agreement (SD=3.42%, Appendix A.5). This high consistency demonstrates that our evaluation standard is robust.
> 2. **Human–LLM agreement**: We respectfully refer the reviewer to Section 5.5 (Page 8) for this validation:  "Our primary finding is a **high degree of alignment**: the ranking produced by our automatic Reasoning F1 score matched the human expert’s preference in 86.23% of the comparisons."
> 3. **Temperature**: To ensure maximum robustness and reproducibility, we **set the temperature to 0.0**. Variances of 3 times evaluation (see below table) are updated in manuscript Table 11.
> 4. **Prompt**: We acknowledge prompt design is important. While testing variance across the infinite space of prompts is infeasible, we conduct experiment with several formulations and **select the most robust one**. We have added the exact prompt used for evaluation to the **Appendix A.12** in our revised manuscript.
>
> | **Model**| |**Research**| |  | | **Review**| |   | | **Avg.**|  |  |
> |---|----|----|----|-------|----|--|-----|----|----|-----|----|----|
> || **Accuracy**|**Precision**|**Recall**|**F1**|**Accuracy**|**Precision**|**Recall**|**F1**|**Accuracy**|**Precision**|**Recall**|**F1**|
> |||||||``Planning``|||||||
> | **Gemini**    |26.08±0.02 |26.08±0.02 |25.43±0.02 |25.39±0.02 |29.75±0.02 |29.75±0.02 |31.02±0.01|29.74±0.01|27.91±0.02|27.91±0.02|28.23±0.02|27.56±0.02|
> |||||||``Retrieval``|||||||
> | **Openai**    |62.00±0.00|-|-|-|0.69±0.00|1.73±0.00|5.27±0.00|1.36±0.00|31.35±0.00|-|-|-|
> | **Gemini**    |72.00±0.00|-|-|-|1.55±0.00|4.00±0.00|7.64±0.00|3.00±0.00|36.77±0.00|-|-|-|
> | **Perplexity**|66.00±0.00|-|-|-|**3.64**±0.00|**22.85**±0.00|9.01±0.00|**6.88**±0.00|34.82±0.00|-|-|-|
> | **Grok**      |**76.00**±0.00|-|-|-|3.62±0.00|13.22±0.00|**37.51**±0.00|6.84±0.00|**39.81**±0.00|-|-|-|
> |||||||``Reasoning``|||||||
> | **Openai**    |61.36±0.00|**61.66**±0.00|41.54±0.00|**48.68**±0.00|56.99±0.00|**61.51**±0.00|38.91±0.00|46.75±0.00|59.18±0.00|**61.59**±0.00|40.22±0.00|47.72±0.00|
> | **Gemini**    |**61.69**±0.00|60.57±0.00|**41.64**±0.00|48.38±0.00|**58.16**±0.00|60.70±0.00|**39.72**±0.00|**47.27**±0.00|**59.93**±0.00|60.63±0.00|**40.68**±0.00|**47.82**±0.00|
> | **Perplexity**|44.55±0.00|59.92±0.01|29.99±0.00|38.99±0.00|44.66±0.01|60.18±0.01|30.44±0.00|39.25±0.00|44.60±0.00|60.05±0.01|30.22±0.00|39.12±0.00|
> | **Grok**      |32.55±0.04|54.28±0.03|22.07±0.03|30.46±0.03|30.29±0.04|58.55±0.03|21.19±0.03|30.12±0.04|31.42±0.04|56.41±0.03|21.63±0.03|30.29±0.03|
> | **Search-r1**|24.20±0.01|33.31±0.01|21.04±0.01|25.79±0.01|22.71±0.01|26.89±0.01|15.86±0.00|19.95±0.01|23.45±0.01|30.10±0.01|18.45±0.01|22.87±0.01|
>
> We have added detailed analysis to **Appendix A.18** in the manuscript.

---

> ### Author Response · Authors · 2025-11-21
> **Response to Reviewer caxP (2/3)**
>
> # A3: On Scale, Confidence Intervals, and Domain Difficulty (Weakness 3, Question 4)
> 1. **Scale:** We believe the scale of our test set is **sufficient for robust evaluation**.
>     - Several high-quality benchmarks recently accepted at ICLR feature a **comparable number of expert-annotated instances**, such as MiniF2F [1] (244 instances) and IRIS [2] (120 instances).
>     - As the **first human-annotated benchmark for academic Deep Research**, the current size represents a significant resource investment.
> 2. **Confidence Intervals:** We calculate 95% CIs to validate our findings. The results confirm statistically significant gaps (non-overlapping CIs) between stable, high-resource domains like Biology ($65.51 \pm 7.92\%$) and volatile, challenging domains like Finance ($45.19 \pm 10.07\%$). This $\approx 20\%$ gap confirms that the benchmark captures genuine domain generalization challenges rather than statistical noise.
> 3. **Domain-Difficulty Characterization:** We characterize difficulty along two dimensions:
>     - **Knowledge Prevalence:** Performance correlates with training data availability; models excel in high-resource fields (e.g., Medicine) but degrade in specialized applied sciences.
>     - **Reasoning Depth:** Analysis of Plan Length (Avg. 9.01 steps) shows that domains requiring longer, multi-step reasoning chains generally correspond to lower success rates.
>
> | **Domain** | **Mean Accuracy (%)** | **95% CI Margin ($\pm$)** |
> | :--- | :---: | :---: |
> | **High Performance** | | |
> | Biology | 65.51 | 7.92 |
> | Medicine | 64.51 | 7.73 |
> | Environmental Science | 63.75 | 9.50 |
> | Energy | 61.35 | 8.60 |
> | Chemistry | 60.44 | 8.64 |
> | **High Difficulty** | | |
> | Building and Construction | 57.90 | 8.27 |
> | Earth Sciences | 57.00 | 9.33 |
> | Materials | 56.57 | 8.23 |
> | Computer Science | 54.25 | 8.86 |
> | Finance | 45.19 | 10.07 |
>
> We have added detailed analysis to **Appendix A.19** in the manuscript.
>
> # A4: Potential Selection Bias (Weakness 4)
> Our citation filter (≥10 citations in 2 years) represents **a trade-off, prioritizing signal-to-noise ratio over the inclusion of the "long tail".** This criterion ensures the benchmark consists of high-impact work. Our goal is to first establish robust agent performance on research that reflects clear methodologies and community consensus, rather than risk "noise" from unvetted papers. We add discussion of this trade-off in the **revised paper (Appendix A.13)** and identify generalization to the "long tail" as a key direction for future work.
>
> # A5: Annotation QA Metrics (Weakness 5)
> We have included the following statistics in **Appendix A.3 and A.20** to demonstrate the consistency and quality of our annotation:
> 1.  **Inter-Annotator Agreement:** We calculate **Cohen’s $\kappa$**, achieving a high score of **0.89**, which indicates strong agreement beyond chance.
> 2.  **Data Distribution:**
>     * **Plan Complexity:** The gold plans have an average length of **9.01 steps**, reflecting a comprehensive structure for the research tasks.
>     * **Diagnostics Distribution:** We analyze the distribution across all **3,400** diagnostic checkpoints. The dataset contains **2,200 (64.71%)** positive (True) instances and **1,200 (35.29%)** negative (False) instances. This balance confirms that the benchmark provides substantial coverage for both verifying correct reasoning and rejecting incorrect paths.
>
> # A6: Risk of Diagnostic-gaming (Weakness 6)
> **Our diagnostic checks are designed to serve as a concrete proxy for holistic reasoning, not as a simplistic checklist.** We explicitly validate our metric against holistic human judgment to ensure it captures genuine synthesis. As we detailed in Section 5.5 (Page 8), our Reasoning F1 score (which is based on these diagnostic checks) achieves an **86.23% match with human expert preferences**. This high correlation strongly suggests that our metric is robust and aligns well with holistic quality, making it difficult for a model to "game" the checks without performing genuine reasoning.
>
> # A7: Reproducibility Transparency (Weakness 7)
> 1. **Released Artifacts:** **We have made a representative subset available in the Supplementary Material.** This 20-instance subset includes one research paper and one review paper randomly selected from each of our 10 domains. This selection covers all components and thus demonstrates the features of our benchmark (such as gold plans, bibliographies). We will release the complete benchmark upon acceptance.
> 2. **Prompts & Toolkit:** We have added the exact judge prompts to **Appendix A.12** and detailed the toolkit settings in **Appendix A.14**.
>
> # Reference:
> [1] miniF2F: a cross-system benchmark for formal Olympiad-level mathematics (Zheng et al., ICLR 2022)
>
> [2] IRIS: LLM-Assisted Static Analysis for Detecting Security Vulnerabilities (Li et al., ICLR 2025)

---

> ### Author Response · Authors · 2025-11-21
> **Response to Reviewer caxP (3/3)**
>
> # A8: Planning Bottleneck (Question 3)
> We conduct the **Gold Plan Injection** experiment as suggested, where ground-truth plans are provided while retrieval and reasoning remain model-generated.
> 1.  **Validation of Bottleneck:** As shown in the table below, fixing the planning module yields a significant performance boost in the downstream Reasoning phase. For instance, **OpenAI's average Reasoning Accuracy rises to 70.93%**, a substantial improvement over the **58.65%** observed in the end-to-end setting (Table 2 in manuscript).
> 2.  **Conclusion:** This nearly 12% gain confirms our hypothesis: **Planning is indeed a dominant bottleneck.** High-quality plans effectively guide the agent to better reasoning outcomes, even if the retrieval component remains imperfect. We have added these results to **Appendix A.21**.
>
> | **Model**  | |**Research**| |  | | **Review**| |   | | **Avg.**|  |  |
> |---|----|----|----|-------|----|--|-----|----|----|-----|----|----|
> || **Accuracy**|**Precision**|**Recall**|**F1**|**Accuracy**|**Precision**|**Recall**|**F1**|**Accuracy**|**Precision**|**Recall**|**F1**|
> |||||||`Retrieval`|||||||
> | **Openai**    |50.00|-|-|-|0.25|0.57|0.65|0.50|25.13|-|-|-|
> | **Gemini**    |70.00|-|-|-|0.98|2.49|11.79|1.93|35.49|-|-|-|
> | **Perplexity**|**80.00**|-|-|-|**4.02**|13.67|**15.60**|**7.64**|**42.01**|-|-|-|
> | **Grok**      |**80.00**|-|-|-|3.90|**27.03**|14.38|7.29|41.95|-|-|-|
> |||||||`Reasoning`|||||||
> | **Openai**    |**68.68**|59.52|**45.32**|**50.16**|**73.18**|**62.76**|**49.77**|**54.59**|**70.93**|**61.14**|**47.54**|**52.38**|
> | **Gemini**    |66.28|60.22|44.00|49.66|54.79|55.33|38.01|44.26|60.53|57.77|41.00|46.96|
> | **Perplexity**|58.36|**61.19**|39.71|47.24|53.96|60.56|35.75|43.87|56.16|60.87|37.73|45.55|
> | **Grok**      |44.66|59.75|30.25|38.84|40.77|58.71|27.93|36.58|42.71|59.23|29.09|37.71|
> | **Search-r1** |31.36|42.43|26.44|32.58|28.70|35.13|20.52|25.91|30.03|38.78|23.48|29.24|
>
> # A9: Dataset Sensitivity and Overfitting (Question 5)
> 1. **Review:Research ratio:** This does not affect our results, because research and review papers are evaluated separately. Their results are analyzed independently (as shown in Tables 2, 3 and 4), so the specific ratio in the dataset does not influence the per-task conclusions.
> 2. **Dataset Size and Overfitting:** First, we clarify that this is a zero-shot evaluation-only benchmark. As no models are trained on this data, there is **no risk of training overfitting**. Second, the reviewer raises a valid point about whether our conclusions are overfitted to this specific dataset. We design the benchmark to **mitigate this risk through diversity:** the dataset spans 10 distinct scientific domains.
>
> # A10: Efficiency Evaluation Details (Question 6)
> 1.  **Tool-Use Normalization (Latency):**
>     We report the **end-to-end latency** (the total time taken for the API call to complete, including internal tool-use) in Section 5.4, as commercial DR systems are black boxes that **do not expose internal API timing for tool calls**. Normalizing this time is infeasible.
>
> 2.  **Output Verbosity & Budget Control:**
>     We confirm that most commercial agents **do not allow users to control reasoning budgets or token verbosity**. We standardize the only adjustable parameter we found (`reasoning_effort` on Perplexity DR) to "medium" across all trials.
>     * **Bias Mitigation:** More importantly, potential bias from verbose or boilerplate output is inherently mitigated because our **primary performance metrics (Accuracy, Precision, Recall, F1)** are based on set-matching against the ground-truth content (gold sub-tasks and citations). These metrics are robust to token verbosity or boilerplate, as they focus on the inclusion and correctness of essential information ($TP, FP, FN$) rather than output length. The formal definitions for all metrics and conceptual mapping are added in **Appendix A.15**.
>
> # A11: Artifact Release (Question 7)
> 1. Yes, we have released **a subset including all components** (gold plans, prompts, scoring scripts, etc.) in the Supplementary Material.
> 2. We have already used this data to inform our **Case Study in Appendix A.9**, which analyzes two representative cases in detail. **We will release the full data upon acceptance.**

---

> ### Author Response · Authors · 2025-11-27
>
> Thank you very much for reviewing our responses. If you believe we have addressed the majority of your concerns, we kindly ask you to consider increasing the score for the paper. Should you have any additional questions, we would greatly appreciate it if you could provide further comments at your earliest convenience. We will respond promptly. Thank you!

---

### Official Review · Reviewer_BaTe · 2025-11-01

**Soundness:** 2
**Presentation:** 3
**Contribution:** 2
**Rating:** 4
**Confidence:** 4

**Summary:**

This paper introduces a new benchmark for deep research agents with 200 instances annotated by humans across 10 scientific domains from publicly available, highly cited research papers. There are 2 modes (end to end - for agents, and isolated - model only) and 3 categories of competencies (planning, retrieval, reasoning). The authors then evaluate a set of models and agents across these and then provide a number of breakdowns (by module, time / token / accuracy trade-offs, human preference validations). The benchmark is still far from saturated.

**Strengths:**

Relevance: Measuring model performance on deep research is a very timely and important area of research, and the academic domains selected are very important for the world (e.g., materials, finance, chemistry, computer science, medicine, biology, environmental science, energy, building and construction, and Earth science.)
Detail: The paper includes a clear breakdown of paper selection criteria, paper selection process, use of domain experts, task domains, example tasks, and quality management of the annotations. This detail helps validate that the process for task creation is likely to be high quality.
Evaluation modes: Evaluating both the agent mode and the mode for assessing baseline model performance makes a lot of sense.
Results: A thorough breakdown of results is provided with a variety of data cuts.
Presentation: The writing is clear and legible, and the figures are quite easy to understand.

**Weaknesses:**

Reproducibility: The full benchmark is not open-sourced or made available upon request, which makes reproduction or validation of the results not possible. I'd highly encourage you to make that available, or at least make a representative subset available (if you want to maintain a holdout set) so that the work can benefit the broader research community. If this changed, I would reconsider my score.
Ablations: Some further experiments would be helpful to make this work even stronger. For example, the authors could consider varying test-time-compute to measure if there is clean scaling and to ensure the benchmark isn't saturated when run with eg pass@k or best-of-N. Similarly, variations in prompting and capability elicitation could be employed to assess the benchmark's robustness to different eval setups. Without these, it is hard to tell if the benchmark is actually unsaturated, or if it is due to under-elicitation of model performance.
Minor nits: There are also a few typos (e.g., 'appendix ??' in page 16). Also, this is minor, but the title could be a bit cleaner ("modular-integrated" is a bit conflicting; the two modes might be a detail better left for the paper versus being the main part of the title).

**Questions:**

How were domain experts identified (i.e., were they always PhD candidates from the relevant field, or was it any PhD candidate judging any paper? how did you make sure folks were experts in the relevant fields to actually understand the underlying papers?)
On pg 16 you say "We computed inter-annotator agreement before adjudication and achieved a high level of consistency."---> what was the agreement rate?

---

> ### Author Response · Authors · 2025-11-21
> **Response to Reviewer BaTe**
>
> We deeply appreciate reviewer's constructive feedback, which has significantly helped us improve the quality of our work.
>
> # A1: Reproducibility and Data Availability (Weakness 1)
> **We have made a representative subset available in the Supplementary Material.** This 20-instance subset includes one research paper and one review paper randomly selected from each of our 10 domains. This selection covers all components and thus demonstrates the features of our benchmark. We will release the complete benchmark upon acceptance.
>
> # A2: Ablations: Test-Time Compute and Robustness (Weakness 2)
> We conduct **pass@k experiments (k=1, 3)** on the released subset and analyzed the results (see table below). All results are presented as percentages. we have added the analysis and figures to **Section 5.6** in the revised manuscript.
>
> | **Model** | **Research (p@1 / p@3)** | **Review (p@1 / p@3)** | **Avg. (p@1 / p@3)** |
> | :--- | :--- | :--- | :--- |
> | **`Planning`** | | | |
> | **Gemini** | 23.26 / 40.00 | 22.96 / 40.00 | 23.11 / 40.00 |
> | **`Retrieval`** | | | |
> | **OpenAI** | 50.00 / 50.00 | 10.51 / 11.26 | 30.26 / 30.63 |
> | **Gemini** | 70.00 / 70.00 | 11.42 / 12.86 | 40.71 / 41.43 |
> | **Perplexity**| 70.00 / 80.00 | 14.88 / 16.42 | 42.44 / 48.21|
> | **Grok** | 66.67 / 70.00 | 14.04 / 15.59 | 40.35 / 42.79 |
> | **`Reasoning`** | | | |
> | **OpenAI** | 62.69 / 70.55 | 57.19 / 66.42 | 59.94 / 68.49|
> | **Gemini** | 62.08 / 67.95 | 49.31 / 65.12 | 55.69 / 66.53|
> | **Perplexity**| 53.73 / 68.80 | 44.35 / 59.52 | 49.04 / 64.16|
> | **Grok** | 33.00 / 49.06 | 30.89 / 46.04 | 31.94 / 47.55 |
> | **Search-r1** | 22.42 / 37.46 | 36.81 / 42.65 | 29.62 / 40.06 |
>
> Analysis of Results:
> 1.  **Clean Scaling:** We observe a consistent performance gain from pass@1 to pass@3 across all modules. Notably, the performance gaps between models at pass@3 remain similar to those at pass@1. This clean scaling indicates that the benchmark successfully measures the benefit of increased test-time compute and is not merely limited by rigid evaluation formats.
> 2.  **No Saturation:** Even with increased compute (pass@3), the highest average Reasoning score is **68.49%** (OpenAI), and Planning remains around **40%**. This substantial gap from 100% confirms that **the benchmark is effectively unsaturated** and remains a challenging standard for future, more capable agents.
>
> # A3: Minor Nits and Title (Weakness 3)
> 1. **Typos:** We have corrected typos in the revised manuscript.
> 2. **Title:** Our intention with "modular-integrated" is to distinguish between isolated module evaluation (modular) and end-to-end evaluation (integrated).
>
> # A4: Regarding Expert Identification (Question 1)
> All annotators are PhD candidates from the relevant field. We recruit experts for all 10 domains, and each annotator exclusively annotated **papers within their own domain of expertise**, ensuring high-quality annotation. We have added details to **Appendix A.3** in the revised manuscript.
>
> # A5: Verifying Annotator Expertise (Question 2)
> We verify expertise in two ways:
> 1. All annotators are PhD candidates from **top universities** who have published in **top-tier venues** within their specific fields.
> 2. All experts completed **a rigorous training and calibration phase** (see Appendix A.3). This require them to analyze papers using a standardized rubric and conduct a cross-review, which align their understanding and ensured annotation consistency.
>
> # A6: Inter-Annotator Agreement (Question 3)
> 1. **Setting:** We conduct an inter-annotator agreement assessment where two annotators independently label the diagnostic answers (True/False) based on given papers.
> 2. **Results:** The **agreement rate is 96.05%, with a Cohen’s $ \kappa $ of 0.89, indicating strong consistency**.
>
> We have added results **Appendix A.3** in the revised manuscript.

---

> ### Author Response · Authors · 2025-11-27
>
> Thank you very much for reviewing our responses. If you believe we have addressed the majority of your concerns, we kindly ask you to consider increasing the score for the paper. Should you have any additional questions, we would greatly appreciate it if you could provide further comments at your earliest convenience. We will respond promptly. Thank you!

---

### Official Review · Reviewer_rzt8 · 2025-11-03

**Soundness:** 2
**Presentation:** 3
**Contribution:** 2
**Rating:** 4
**Confidence:** 2

**Summary:**

This paper provides the benchmark (Dr.Mi-Bench) and evaluation framework (Dr.Mi-Eval) for deep research agents, a rising field of deep research systems evaluating the ability to plan and reason (and retrieve information, which was covered by previous evaluation benchmark/frameworks) in these agents, and specializes on professional and domain-specific cases. Given a research question, the proposed method scores plan generation and evidence retrieval using the  expert-annotated paper outlines and citations of real-world papers as reference, and the evaluation of reasoning is done on a set of 15-20 boolean diagnostic statements of the original paper tested on the generated. In addition, there are 2 modes to evaluating the agents' reasoning abilities, one using what plans and evidence they had gathered, and another isolated evaluation which makes use of gold plans and evidence. The benchmark comes from real-life research and review papers and makes up 200 instances across 10 scientific domains. The evaluation on state-of-the-art systems (OpenAI, Gemini, Perplexity, Grok) shows performance disparity across disciplines, such as the performance being higher on topics like computer science but worse on topics such as materials and earth science; while some agents excel at retrieval (Grok: 40% accuracy) or reasoning (Gemini: 60% accuracy), all struggle with planning (~25% F1) and with review-style tasks requiring multi-source synthesis.

**Strengths:**

1. Existing benchmarks (GAIA, BrowseComp, etc.) only test retrieval, not planning or reasoning. Dr.Mi-Bench addresses a significant gap in deep research evaluation by providing comprehensive, modular evaluation of planning, retrieval, and reasoning specifically for scientific research tasks.
2. The modular evaluation paradigm is a significant methodological contribution. By providing both end-to-end and isolated evaluation modes, it could reveal different failure modes and provides crucial information for future development.
3. The diagnostic statement approach is an effective, efficient, and reliable way to test the consistency between the generated report and the original paper.
4. The dataset is high-quality, annotated by experts and based on papers from top venues (Nature, ICLR, etc.) with >= 10 citations, ensuring its ecological validity, which is significantly more valuable than automatically annotated data.
5. The empirical evaluation is comprehensive, covering both commercial DR systems (OpenAI, Gemini, Perplexity, Grok) and foundational LLMs (GPT, Claude, Llama, Qwen). The cross-domain analysis (Figure 2) provides valuable insights about performance disparities across fields, and the efficiency analysis (Section 5.4) addresses practical deployment considerations often neglected in academic benchmarks.

**Weaknesses:**

1. While exhibiting high-quality, the size of the data might be not so big, and the expert-annotation mode might be hard to scale up. In addition, non-STEM fields could also be included into the benchmark. Cross-disciplinary research on itself could be difficult in real life as the collaboration between scholars of different departments and specialties, so it is one of the gaps that automated research agents are aiming to close. Therefore, adding domains such as the crossing between 2 of the studied research domains could also help.
2. One of the concerns is reproducibility due to the use of APIs, as the providers might discontinue an API endpoint, or update the model attached to that API. In addition, leakage could be an issue since it might be possible for the papers (2024+) to have been exposed to the called deep research systems during training, such as o3-deep-research and Search-rl. It is indeed near impossible to evaluate such API-accessed deep research systems, since we don't have control over the LLMs they use and the information they can retrieve. I think what we can do is to find the latest announced cut-off date in all the DRs you used, and only select papers after this date.
3. It is not a full fix since we can't control how they retrieve, but we can always do post-processing after we retrieve. You see, it is impossible for paper a to cite the papers b, c, d that came after a which explores the same idea but with better methods. When we compare what the DRs had retrieved to what the paper a had cited, we might actually have artificially lower scores since papers b, c, d might dominated the retrieved evidence. We could remove the "future papers" out before evaluating them on evaluation.
4. Following the same issue, these "future papers" can also contaminate the final reasoning phase and make the scores higher than they should be, for it might be this scenario: we are asking the DRs to research on the topic A from paper a; paper a spawned paper b, c, d, etc, all based off paper a and are incremental to the solution proposed by paper a. By removing the "future papers" retrieved evidence, we can derive at more convincing numbers of the end-to-end evaluation for DR agents!

Among the issues raised above, I'd like to suggest one solution that is both implementable and important: temporal filtering of retrieved evidence. Before evaluating retrieved evidence, filter out any papers published after the source paper's publication date. This requires only metadata extraction (publication dates) and a simple filtering step during evaluation.

**Questions:**

1. The paper evaluates commercial deep research systems (OpenAI o3, Gemini, Perplexity, Grok) but does not report the monetary costs associated with completing research tasks. Given that users have limited control over these systems' behavior, understanding cost is crucial for practical deployment decisions. What is the average cost to complete one research with each of these DRs in this benchmark? I understand that as a user you might not have too much control on DRs, so it is important to record their cost on average, and also per task.
2. I believe it to be possible to apply the temporal filtering on the retrieved papers before evaluating the retrieval and reasoning capabilities I consider this recommendation high-priority because it addresses a fundamental validity concern with a practical, implementable solution. The current evaluation may substantially overestimate system capabilities by allowing access to forward citations and derivative works. Implementing temporal filtering would strengthen the benchmark's scientific rigor and provide more accurate measurements of deep research capabilities..
3. If possible, maybe we can also make sure that the papers used by the benchmark should be pass the cutoff dates of all the DRs, or if not, as new as possible, so less potential leakage in the LLMs used by these DRs. However, if implementing this requirement would necessitate extensive re-annotation and is not feasible for this submission, please explicitly acknowledge training data contamination as a limitation, and propose a versioning strategy for future updates with newer papers. I recognize this is a challenging constraint and defer to the authors' judgment on feasibility given time and resource constraints.

---

> ### Author Response · Authors · 2025-11-21
> **Response to Reviewer rzt8 (1/2)**
>
> We sincerely thank reviewer's insightful comments and the valuable time spent evaluating our manuscript.
>
> # A1: Data Size, Scalability, non-STEM, and Cross-disciplinary (Weakness 1)
> 1. **Size:** We believe the size of our test set (**200** instances) is *sufficient for robust evaluation*.
>     - Several high-quality benchmarks recently accepted at ICLR feature a **comparable number of expert-annotated instances**, such as MiniF2F [1] (**244** instances) and IRIS [2] (**120** instances).
>     - As the **first human-annotated benchmark for academic Deep Research** that supports end-to-end and module-specific evaluation, the current size requires a significant resource investment.
> 2. **Scalability:** Unlike benchmarks that rely heavily on subjective expert judgment, our approach adopts a **evidence-based, objective paradigm**. We systematically ground annotations in the verifiable content of high-quality source papers (e.g., research topics, writing structures, and key points) rather than expert intuition. This evidence-based methodology not only ensures reproducibility but also makes the task highly suitable for future automation using LLMs.
> 3. **Non-STEM:** We would like to clarify that our benchmark **already includes non-STEM fields**, such as Finance, Medicine and Building and Construction.
> 4. **Cross-disciplinary:** We are pleased to confirm that **many of our chosen domains are cross-disciplinary**:
>     | **Domain**                | **Cross-disciplinary**   |
>     |--------------|-------------|
>     | **Materials**                 | Physics, Chemistry, Computer Science |
>     | **Medicine**                  | Biology, Chemistry, Computer Science |
>     | **Environmental Science**     | Biology, Chemistry, Earth Science, Computer Science |
>     | **Energy**                    | Physics, Chemistry, Mechanical/Electrical Engineering |
>     | **Building and Construction** | Civil Engineering, Materials, Architecture, Biology |
>     | **Finance**                   | Economics, Mathematics, Computer Science |
>
> In addition, our selected papers may simultaneously belong to multiple domains. For example, paper [3] in our dataset (details in Appendix A.4), while published in a Building and Construction journal, deeply intersects with materials and biology.
>
> We have incorporated analysis into Section 4.1 of the revised manuscript.
>
> # A2: API Reproducibility and Retrieval Control (Weakness 2, Question 3)
> 1. **API Reproducibility:** Our framework is expressly designed to **mitigate reproducibility risk by grounding the evaluation in objective criteria** (i.e., plans and diagnostics checklists), rather than relying on any subjective human eval or LLM-as-a-judge.
>     - Experimental results: To validate this, we test GPT-4o and Gemini-2.5-pro as blind judges and achieve a **93.00% inter-model agreement** (SD=3.42%, Appendix A.5 L1176-L1178). This high consistency demonstrates that our evaluation standard is robust and the specific judge API is changeable.
>     - Therefore, if an API is deprecated, **other APIs or open-source LLMs can reliably substitute it**. We will also periodically update results from new APIs on our benchmark website.
> 2. **Retrieval Control:** We address the 'lack of control' via **isolated module evaluation** (L152-L153). This setup allows us to fix the retrieval input (using gold evidence) and test specific LLM backbones, ensuring a fair and reproducible assessment of core reasoning potential independent of retrieval variability.
> 3. **Cut-off Dates Experiment:** While we agree that using post-cut-off papers prevents leakage, evaluating only new papers does not align with real-world usage, where users heavily query historical literature.
>     - Setting: We explicitly validate our Gemini DR evaluation results on a subset of data published after March 2025 (the release date of Gemini-2.5-pro).
>     - Results Analysis: **The experimental results indicate that data leakage has a negligible impact on our benchmark.** As shown in the table below, performance on the temporally constrained subset is comparable to, or even slightly higher than, the unrestricted baseline.
>
> | **Model**|**Accuracy**|**Precision**|**Recall**|**F1**|
> |---|----|----|----|-------|
> |``Retrieval``|||||
> | **Cut-off Dates Constrained**    |37.15|-|-|-|
> | **Unrestricted (Baseline)**   |36.77|-|-|-|
> |``Reasoning``|||||
> | **Cut-off Dates Constrained**    |63.97|60.97|43.33|49.91|
> | **Unrestricted (Baseline)**   |59.76|60.67|40.48|47.66|
>
> We have added cost analysis to **Section 5.6** in the revised manuscript.
>
> # Reference:
> [1] miniF2F: a cross-system benchmark for formal Olympiad-level mathematics (Zheng et al., ICLR 2022)
>
> [2] IRIS: LLM-Assisted Static Analysis for Detecting Security Vulnerabilities (Li et al., ICLR 2025)
>
> [3] Biochar-enabled carbon negative aggregate designed by core-shell structure: A novel biochar utilizing method in concrete (Zou et al., Construction and Building Materials 2024)

---

> ### Author Response · Authors · 2025-11-21
> **Response to Reviewer rzt8 (2/2)**
>
> # A3: Cost Analysis (Question 1)
> We present the costs in USD for the API-based systems in the table. Since Gemini is evaluated manually via the web interface (due to the lack of a public API), cost metrics do not apply.
> | **Model**     | **Average Cost per Task**   | **Total Cost (Benchmark)**   |
> |---------------|-------------|-------------|
> | **OpenAI**    |$1.76| $351.45|
> | **Perplexity**|$0.68 |$135.43|
> | **Grok**      |$0.36 |$72.00|
>
> We have added cost analysis to **Appendix A.16** in the revised manuscript.
>
> # A4: Leakage: Temporal Filtering for Retrieval Validity (Weakness 3, 4, Question 2)
> We appreciate the reviewer's suggestion, which provides a valuable measure of scientific rigor. We conduct the **temporal filtering experiment by adding the publication date constraint to the prompt** on OpenAI DR.
> 1. **Findings and Validity:** The overall average performance shift is minimal (Retrieval Accuracy shift from 25.18% to 25.27%), confirming the general validity of our initial results.
> 2. **Real-World Alignment:**
>    - We adhere to the Unrestricted (Baseline) evaluation because it directly **reflects real-world usage** (users seldom impose publication date constraints  in his/er query).
>    - The **recency of our benchmark papers** also mitigates the overall risk of data leakage.
>
> We have incorporated time-constrained results and analysis into **Section 5.6** of the revised manuscript.
>
> |**Module**| |**Research**| |  | | **Review**| |   | | **Avg.**|  |  |
> |---|----|----|----|-------|----|--|-----|----|----|-----|----|----|
> | **Model**| **Accuracy**|**Precision**|**Recall**|**F1**|**Accuracy**|**Precision**|**Recall**|**F1**|**Accuracy**|**Precision**|**Recall**|**F1**|
> |||||||``Retrieval``|||||||
> | **Temporally Constrained**    |50.00|-|-|-|0.53|0.73|11.21|1.05|25.27|-|-|-|
> | **Unrestricted (Baseline)**    |50.00|-|-|-|0.36|0.61|10.51|0.71|25.18|-|-|-|
> |||||||``Reasoning``|||||||
> | **Temporally Constrained**    |58.59|58.83 |37.66|44.82|60.22|61.54|41.02|48.50|59.41|60.18|39.34|46.66|
> | **Unrestricted (Baseline)**    |62.69|60.02|40.43|47.17|57.19|58.58|39.04|45.94|59.94|59.30|39.74|46.56|

---

> ### Author Response · Authors · 2025-11-27
>
> Thank you very much for reviewing our responses. If you believe we have addressed the majority of your concerns, we kindly ask you to consider increasing the score for the paper. Should you have any additional questions, we would greatly appreciate it if you could provide further comments at your earliest convenience. We will respond promptly. Thank you!

---

### Author Response · Authors · 2025-12-01
**Summary of Rebuttal Updates for Area Chair**

Esteemed Area Chair,

**We sincerely thank you for evaluating our work.** We also appreciate meaningful comments of all reviewers.

# Fulfillment of the Score Increase Condition (From Initial Review)
Reviewer BaTe explicitly stated they would **reconsider their score** upon data release. **We have met this condition** by releasing a representative subset in the Supplementary Material, but the system outage likely prevented the update.

# Summary of Rebuttal

1. We are encouraged that the reviewers appreciate our Dr.Mi-Bench:

   * **Significant & Timely Contribution:** Fills critical gaps in deep research evaluation beyond simple retrieval. (rzt8, BaTe)
   * **Novel Modular-integrated Paradigm:** Disentangles planning, retrieval, and reasoning with dual evaluation modes (end-to-end vs. isolated). (All)
   * **High-quality Dataset:** Expert-annotated dataset derived from top-tier papers across 10 disciplines. (rzt8, BaTe, TLtz)
   * **Actionable Insights:** Comprehensive analysis of bottlenecks, efficiency, and domain gaps. (All)
   * **Objective and Reproducible Evaluation:** Paper-grounded supervision and diagnostic statements, validated by strong human alignment. (rzt8, caxP, TLtz)
   * **Clear Presentation:** Well-structured writing and thorough breakdowns. (BaTe)

2. We believe we have **successfully addressed all key concerns**, as summarized below:

    1. **Reviewer rzt8** (Rating: 4)

    | Main Concern | Our Response |
    | :-- | :-- |
    | A1: Data Scalability & Domain Diversity | 1. **Sufficient**: Comparable to recent ICLR benchmarks (MiniF2F, IRIS). 2. **Scalable method**: Objective annotation paradigm ensures reproducibility and future automation. 3. **Non-STEM** (e.g., Finance) & **cross-disciplinary** covered. |
    | A2: Reproducibility | **93% inter-model agreement** allows LLM judge substitution. |
    | A2: Retrieval Control | **Isolated modules** use gold evidence to decouple reasoning from retrieval noise. |
    | A2, A4: Data Leakage | 1. **Post-cutoff experiments** confirm negligible impact of data leakage. 2. **Temporal filtering experiments** confirm result validity. |
    | A3: Cost Analysis | Report **average and total costs**. |

    2. **Reviewer BaTe** (Rating: 4)

    | Main Concern | Our Response |
    | :-- | :-- |
    | A1: Reproducibility | **Release representative subset**, fully **meeting the reviewer's explicit condition** for score reconsideration. |
    | A2: Scaling & Robustness | **Pass@k (k=1,3) experiments** confirm clean scaling and no saturation. |
    | A3: Minor Nits | Corrected typos; clarified title rationale (dual modes). |
    | A4, A5: Expert Qualification | **Domain-matched PhDs** with top-tier **publication records**; rigorously calibrated. |
    | A6: Inter-Annotator Agreement | High consistency: **96.05% agreement rate & Cohen's $\kappa$ of 0.89**. |

    3. **Reviewer caxP** (Rating: 6)

    | Main Concern | Our Response |
    | :-- | :-- |
    | A1: Retrieval Fairness | 1. **Robust Method:** Adopted 20-char title matching. 2. **Validity:** Title changes in final versions are **rare (<1%)**.|
    | A2: LLM-Judge Reliability | 1. **93%** inter-judge agreement & **86%** human alignment. 2. Set temp=0.0; Negligible variance (SD $\le$ 0.04); robust prompt. |
    | A3: Scale & Difficulty | 1. **Sufficient**: Comparable to recent ICLR benchmarks (MiniF2F, IRIS). 2. Rigor: **95% CIs** & difficulty characterization. |
    | A4: Selection Bias | **Deliberate trade-off**: Prioritized high-impact papers for signal-to-noise ratio. |
    | A5: Annotation Info | **Cohen’s $\kappa$=0.89**, avg. plan: 9.01 steps, balanced diagnostic. |
    | A6: Risk of Diagnostic-gaming | **86% human preference alignment** proves resistance to checklist gaming. |
    | A7, A11: Reproducibility | Released artifacts; clarified settings. |
    | A8: Planning Bottleneck | Injecting gold plans boosted reasoning accuracy by **~12%**. |
    | A9: Dataset Sensitivity and Overfitting | Tasks evaluated independently, no overfitting. |
    | A10: Efficiency | Explained black-box constraints; metrics robust to verbosity. |

    4.  **Reviewer TLtz** (Rating: 4)

    | Main Concern | Our Response |
    | :-- | :-- |
    | A1: Modular vs. End-to-End | Necessary to isolate LLM reasoning from agent framework noise. |
    | A2: Planning Metric | Add formulas; Mapped Correctness=Accuracy, Redundancy=Precision, Coverage=Recall. |
    | A3, A5: Typos | Corrected. |
    | A4: Variance | Negligible variance (SD $\le$ 0.04). |
    | A6: Planning Bottleneck | Injecting gold plans boosted reasoning accuracy by **~12%**. |
    | A6: Evaluation Alignment | 1. **Set-level pairwise** matching. 2. Aligns with **academic writing paradigms**. |
    | A7: Fixed Diagnostics | Open-ended judges are **unreliable** (48% human alignment vs. our 86%), yield inflated scores (85% acc vs. our 59%), and lack discrimination (0.05 SD vs. our 0.19). |

The revised PDF containing all these improvements has been uploaded.

---

### Meta-Review · Area_Chair_1UUc · 2026-01-07

**Summary:**

Reviewers broadly agreed that the paper addresses a timely and important problem, namely the evaluation of scientific deep research agents, and acknowledged that the work targets gaps not covered by retrieval-focused benchmarks. At the same time, most reviewers viewed the submission as borderline in maturity. A recurring concern was the limited scale and representativeness of the dataset with 200 instances across 10 domains, which raised questions about the strength of cross-domain generalization claims and possible selection bias from focusing on highly cited papers. Several reviewers felt the benchmark should be interpreted as an initial exploration rather than a fully established standard for the community.

Reviewers also raised methodological and reproducibility concerns, including the initial lack of actual examples from the benchmark, reliance on proprietary APIs, and potential data leakage from exposure to future or derivative papers. Additional questions were raised about the rigidity of the evaluation design, such as strict retrieval matching, alignment to annotated plan structures, and reliance on fixed diagnostic statements and LLM-based judges, which may under-credit valid alternatives or fail to reflect realistic research behavior. Although the rebuttal addressed many of these points with added experiments, validation, and partial data release, concerns remained about how well the benchmark reflects real-world deep research systems.

Rejection rationale: I recommend rejection because the proposed benchmark does not sufficiently align with how current deep research agents are actually built and evaluated, where the agentic framework itself plays a central role in robustness and performance. While the modular diagnostics are useful for analyzing isolated capabilities, a benchmark intended for deep research agents should explicitly account for the full agentic system, including control logic, tool use, memory, and interactions across components. By focusing primarily on scientific domains and modular evaluation, the scope becomes too narrow to capture the holistic behavior of modern deep research agents, limiting its relevance as a general benchmark.

**Reviewer Concerns:**

Concerns Largely Addressed by the Rebuttal
Several technical and procedural concerns raised by reviewers were substantially addressed in the rebuttal and revised manuscript. First, reproducibility and transparency issues were partially mitigated through the release of a representative subset of the benchmark, inclusion of judge prompts, reporting of inter-annotator agreement, and additional details on annotation quality and evaluation settings. Second, concerns about LLM-judge reliability were addressed with extensive validation, including inter-LLM agreement, human–LLM alignment, temperature control, and variance analysis, which strengthened confidence in metric stability. Third, reviewers’ requests for additional experiments and ablations were largely satisfied through pass@k scaling studies, gold-plan injection experiments confirming planning as a bottleneck, cost analysis for API-based systems, confidence intervals, and domain-level difficulty characterization. Finally, concerns around data leakage and temporal validity were explicitly tested through post–cutoff and temporal filtering experiments, which reduced uncertainty about inflated performance.

Concerns That Remain Outstanding
Despite these improvements, several core conceptual concerns remain unresolved. Most importantly, the benchmark still does not adequately reflect the role of the agentic framework in modern deep research agents. While reviewers questioned whether modular evaluation alone can capture real-world deep research behavior, the rebuttal largely defended this design choice rather than expanding the evaluation to meaningfully incorporate end-to-end agentic components such as tool orchestration, memory, search strategies, and control logic. Relatedly, concerns about evaluation realism and fairness persist, particularly for retrieval and planning, where rigid matching to gold plans and bibliographies may under-credit valid alternative strategies even if technically justified. Finally, issues of scope and representativeness remain only partially addressed: the dataset scale, citation-based selection bias, and narrow framing around scientific domains and modular diagnostics continue to limit the benchmark’s relevance as a general evaluation for deep research agents rather than as a diagnostic tool for LLM backbones.

**Reviewer Scores:**

rzt8 -> maintained / increased the score since concerns were partially addressed
BaTe -> maintained / increased the score since concerns were partially addressed
caxP -> maintained / increased the score since concerns were partially addressed
TLtz -> maintained the score since concerns were not sufficiently addressed

---

### Decision · Program_Chairs · 2026-01-26

Reject